# VL-ICL Bench: The Devil in the Details of Multimodal In-Context Learning

**Yongshuo Zong\*, Ondrej Bohdal\*, Timothy Hospedales**
University of Edinburgh    \* Co-first authors
{yongshuo.zong, ondrej.bohdal, t.hospedales}@ed.ac.uk

## Abstract

Large language models (LLMs) famously exhibit emergent in-context learning (ICL) – the ability to rapidly adapt to new tasks using few-shot examples provided as a prompt, without updating the model's weights. Built on top of LLMs, vision large language models (VLLMs) have advanced significantly in areas such as recognition, visual question answering (VQA), reasoning, and grounding. However, investigations into *multimodal ICL* have predominantly focused on few-shot VQA and image captioning, which we will show neither exploit the strengths of ICL, nor test its limitations. The broader capabilities and limitations of multimodal ICL remain under-explored. In this study, we introduce a comprehensive benchmark for multimodal in-context learning. Our *VL-ICL Bench* encompasses a broad spectrum of tasks that involve both images and text as inputs and outputs, and different types of challenges, from perception to reasoning and long context length. We evaluate the abilities of state-of-the-art VLLMs on this benchmark suite, revealing their diverse strengths and weaknesses, and showing that even the most advanced models, such as GPT-4, find the tasks challenging. By highlighting a range of new ICL tasks, and the associated strengths and limitations of existing models, we hope that our dataset will inspire future work on enhancing the in-context learning capabilities of VLLMs, as well as inspire new applications that leverage VLLM ICL. Project page: https://ys-zong.github.io/VL-ICL/.

## 1 Introduction

With the scaling of model size, large language models (LLMs) famously exhibit the emergent capability of in-context learning (ICL) (Brown et al., 2020; Dong et al., 2024). This refers to the ability to learn from *analogy* within a single feed-forward pass – thus, enabling the model to learn completely new tasks using a few input-output examples, without requiring any updates to the model parameters. This training-free nature of ICL has led to its rapid and broad application across a wide range of scenarios and applications, as illustrated by benchmarks such as Hendrycks et al. (2021); Zhong et al. (2024); Srivastava et al. (2023); Cobbe et al. (2021).

Vision large language models (VLLMs) are typically built on a base LLM, by augmenting it with a vision encoder and/or decoder connected by some stitching mechanism (Liu et al., 2024a; 2023; Bai et al., 2023; Zong et al., 2024; Alayrac et al., 2022; Ge et al., 2024). These models have rapidly advanced alongside LLMs, and attracted significant attention for their remarkable multi-modal capabilities in zero-shot recognition, reasoning, grounding, and visual question answering (VQA) among other capabilities. These capabilities have been thoroughly tested by a range of recent benchmark suites (Liu et al., 2024d; Fu et al., 2023; Li et al., 2024b; Yu et al., 2024). Meanwhile, VLLMs are also widely presumed to inherit in-context learning (ICL) capabilities from their base LLM, however, their abilities in this respect are poorly evaluated and poorly understood. Current VLLMs studies mainly report their zero-shot capabilities measured by the benchmarks above, while ICL is usually only evaluated qualitatively, or as a secondary consideration via few-shot visual question answering (VQA) or image captioning (Bai et al., 2023; Awadalla et al., 2023; Sun et al., 2024a; Laurençon et al., 2023), with a notable deficiency in quantitative assessment across a wider spectrum of ICL tasks. This is presumably due to the ready availability of VQA and captioning benchmarking infrastructure. However, we will show that captioning and VQA tasks are not ideal for ICL evalu-

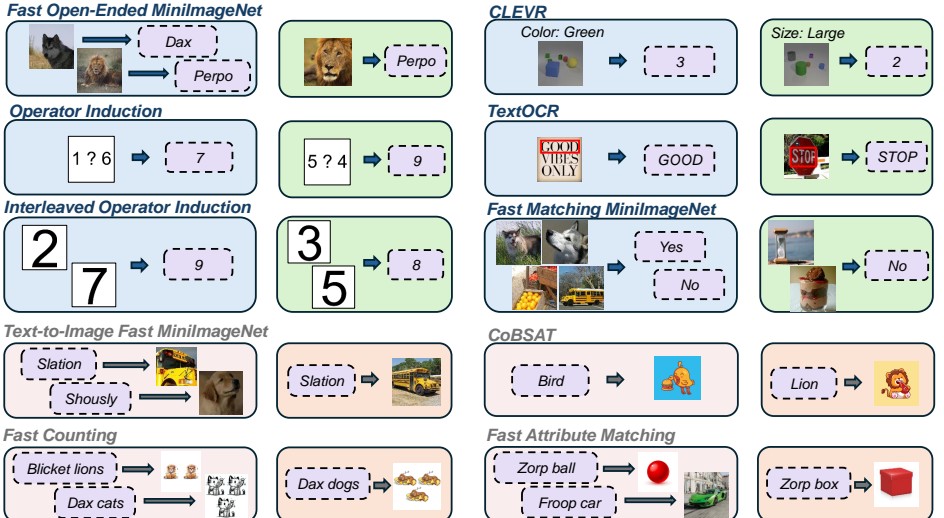

Figure 1: Illustration of the different tasks in VL-ICL Bench. Image-to-text tasks are in the first three rows, while text-to-image tasks are in the bottom two rows. Image-to-text tasks in the third row do reasoning on interleaved image-text inputs.

ation: They neither truly exploit the ability of ICL to improve performance from examples; nor do they test the limits of what ICL can do. Thus there is a lack of motivation for future VLLM research to better exploit and expose the underpinning LLM's ICL ability.

To enhance the understanding of multimodal ICL and assess the ICL capabilities of state-of-the-art VLLMs, we introduce a novel benchmark suite **VL-ICL Bench** (Figure 1), tailored for assessing VLLM in-context learning. Our benchmark suite incorporates both text-output and image-output tasks, and is designed to test various facets of VLLMs, including fine-grained perception, reasoning, rule induction, and context-length. We conduct comprehensive evaluations of state-of-the-art VLLMs that are capable of processing interleaved image-text as inputs on our benchmark. Results reveal that although certain models exhibit reasonable performance on specific tasks, none demonstrate uniform excellence across the entire spectrum of tasks, and some models perform near chance level on some tasks. We hope that this systematic study of different opportunities and challenges for multi-modal ICL will support practitioners to know what is currently possible and impossible in terms of training-free learning of new multi-modal tasks, and spur VLLM model developers to study how to expose as much as possible of the LLM's ICL ability to the multi-modal world.

To summarise our contributions: (1) We demonstrate the limitations inherent in the common practice of quantitatively evaluating VLLM ICL via VQA and captioning. (2) We introduce the first thorough and integrated benchmark suite of ICL tasks covering diverse challenges including perception, reasoning, rule-induction, long context-length and text-to-image/image-to-text. (3) We rigorously evaluate a range of state of the art VLLMs on our benchmark suite, and highlight their diverse strengths and weaknesses, as well the varying maturity of solutions to different ICL challenges.

## 2 BACKGROUND AND MOTIVATION

### 2.1 THE ICL PROBLEM SETTING

Given a pre-trained VLLM $\theta$, an optional text instruction $I$, a context set[1] $S = \{(x_i, y_i)\}$ of query example $x$ and labels $y$, and a test example $x^*$, ICL models estimate

$$p_\theta(y^*|x^*, I, S) \tag{1}$$

with a feed-forward pass. For LLMs, $x$ and $y$ are typically text. For VLLMs, $x$ can be text and/or images, and $y$ can be text (image-to-text ICL) or images (text-to-image ICL).

---

[1]We use context set and support set interchangeably in this paper.

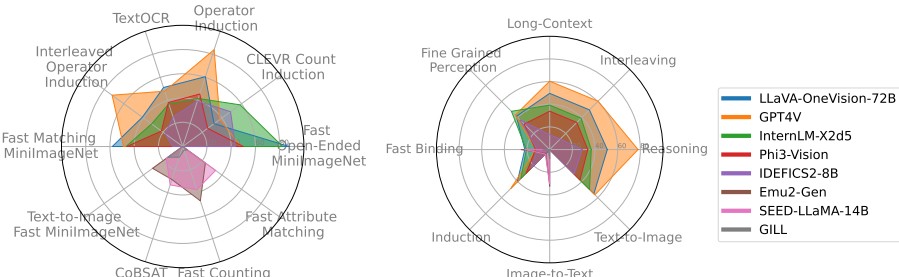

Figure 2: Performance of top-5 image-to-text models and top-3 text-to-image models on our benchmark. Left: By dataset separately. Right: By capability evaluated, averaging over datasets.

This ICL setting is in contrast to the simpler zero-shot scenario, where pre-trained models estimate $p_\theta(y^*|x^*, I)$ purely based on the pre-learned knowledge in $\theta$ with no additional training data provided in $S$. The zero-shot scenario has been rigorously evaluated by diverse benchmarks (Liu et al., 2024d; Fu et al., 2023; Li et al., 2024b; Yu et al., 2024), and in the following section we discuss the limitations of existing ICL evaluations that motivate our benchmark.

## 2.2 COMMON PRACTICE IN ICL EVALUATION

The benchmarks that have been most popular in prior attempts at quantitative evaluation of multi-modal ICL are VQA and image captioning. We focus our discussion in this section on image-to-text models (Alayrac et al., 2022; Bai et al., 2023; Li et al., 2023b; Laurençon et al., 2023; Awadalla et al., 2023), as in-context text-to-image models (Ge et al., 2024; Koh et al., 2023) are relatively less common and less mature, so there is no common evaluation practice yet. In the case of captioning, the context set $S$ contains examples of images $x$ and captions $y$; while for VQA the context $S$ contains image-question pairs $x$ and answers $y$.

Figure 3(a) plots the ICL performance of six popular VLLMs on three widely used benchmarks - MathVista VQA (Lu et al., 2024), VizWiz VQA (Gurari et al., 2018), and COCO Captioning (Lin et al., 2014) for varying numbers of training examples (shots). While the performance of the different models varies, the key observation is that most of lines/models show only limited improvement with ICL (shots > 0) compared to the zero-shot case (shots = 0). This is because, while the context set $S$ illustrates the notion of asking and answering a question or captioning images, the baseline VLLM $\theta$ is already quite good at VQA and captioning. The limiting factors in VLLM captioning and VQA are things like detailed perception, common sense knowledge, etc. – all of which are fundamental challenges to the VLLM, and not things that can reasonably be taught by a few-shot support set.

Given the discussion above, it is unclear why performance should improve with shots at all? We conjectured that this is largely due to the VLLM learning about each dataset's preferred *answer style*, rather than learning to better solve multi-modal inference tasks per-se. For example, in captioning zero-shot VLLMs tend to produce more verbose captions than COCO ground-truth, and they learn to be more concise through ICL. Meanwhile, for VQA, there is a standard practice of evaluating based on string match between the ground-truth answer and the model-provided answer. For example, VizWiz has unanswerable questions, which some VLLMs answer with `"I don't know"` which would not be string matched against a ground truth `"Unanswerable"`. Some models thus learn about answer-formatting (e.g., preferred terminology; avoid using any preface or postface that may throw of a string match) from the context set. This is indeed a kind of ICL, but perhaps not what one expects to be learning in VQA. To validate this conjecture, we repeat the previous evaluation, but using soft matching to eliminate the impact of answer format learning. For VQA, we use a pretrained LLM to determine whether the prediction has semantically equivalent to the ground-truth while for captioning, we use GPT-3.5 to score the quality of the generated caption on a scale of score 1-10 (details in appendix). Fig. 3(b) shows that the curves have almost fully flattened out, with zero-shot performance having improved. Fig. 3(c) quantifies this difference by showing the average rate of improvement with shots for exact match and LLM match. The change to LLM validation almost completely eliminates any benefit of ICL over zero-shot.

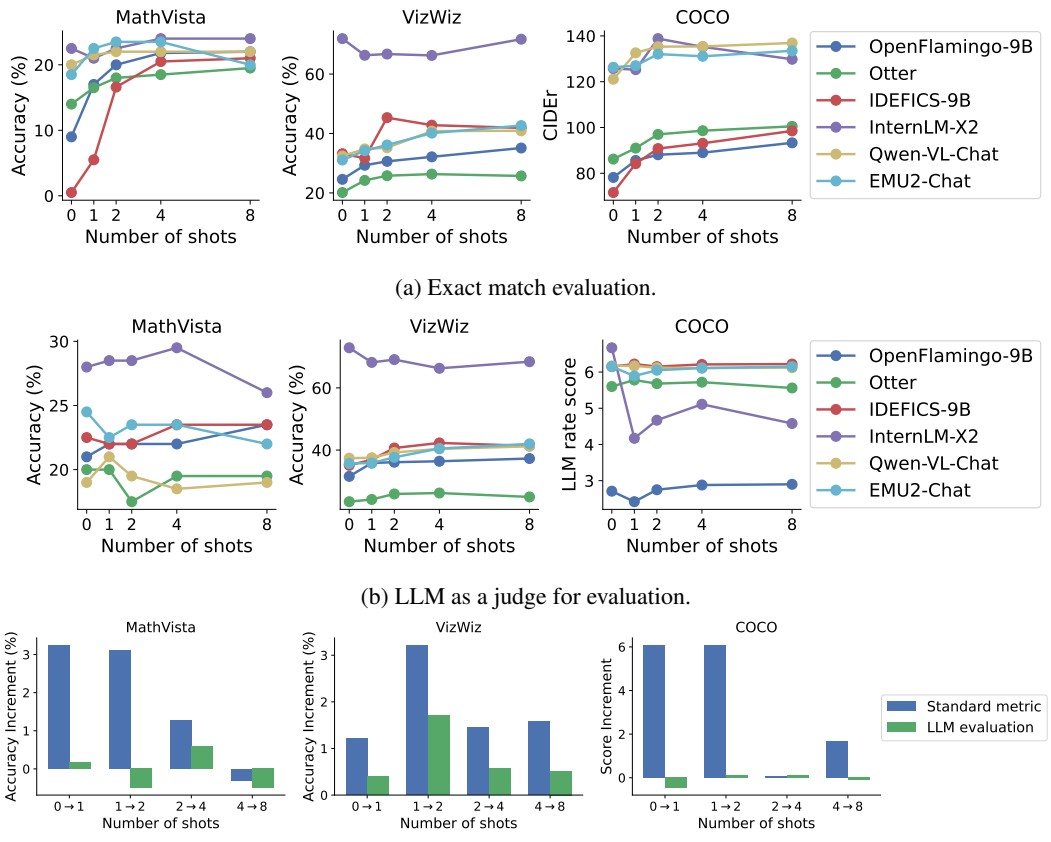

Figure 3: VQA and Captioning are poor benchmarks for image-to-text ICL. (a) Evaluating state-of-the-art VLLMs on representative examples of popular image-to-text ICL benchmarks – MathVista, VizWiz, and COCO – with standard exact match evaluation protocol. The amount of in-context learning is limited in many cases. (b) Re-evaluation of VLLMs with LLM-based evaluation almost eliminates improvement with context size. (c) The impact of ICL on performance goes from limited to negligible when moving from traditional to LLM-based evaluation. ICL on these benchmarks primarily learns answer style/format.

In contrast to the above, popular LLM ICL benchmarks in the language domain *do* usually exhibit non-trivial ICL learning (Brown et al., 2020; Dong et al., 2024). Figure 4 shows three state of the art VLLMs along with their corresponding base LLMs, evaluated on three popular NLP tasks (AG-News (Zhang et al., 2015), MIT Movies (Ushio & Camacho-Collados, 2021) and TREC (Voorhees & Tice, 2000)). We can see that in contrast to the VQA/Captioning benchmarks, models' zero-shot performance is often substantially improved by few-shot ICL. This result confirms that the LLM components in VLLMs do inherit the ICL ability of their base LLM. However, it raises the question of how we can meaningfully exploit and measure the ICL ability of VLLMs in the *multi-modal* context. In the next section, we introduce our benchmark VL-ICL Bench, which does exactly this.

## 3 VL-ICL BENCH

### 3.1 MAIN MULTIMODAL BENCHMARK

Our VL-ICL Bench covers a number of tasks, which includes diverse ICL capabilities spanning concept binding, reasoning or fine-grained perception. It covers both image-to-text and text-to-image generation. Our benchmark includes ten tasks detailed below, illustrated in Figure 1. Table 1 summarises the diverse capabilities tested by each VL-ICL Bench task, and demonstrates its com-

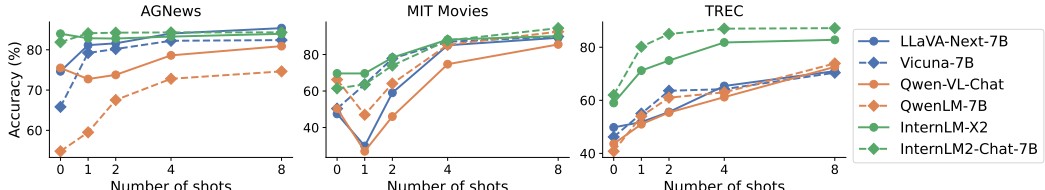

Figure 4: Evaluating state-of-the-art VLLM/LLM pairs on popular text-to-text ICL benchmarks. Few-shot ICL often substantially improves on zero-shot performance, indicating that meaningful in-context-learning is taking place, unlike for the popular image-to-text VLLM benchmarks in Fig. 3.

Table 1: VL-ICL Bench overview. It evaluates diverse capabilities and challenges of ICL with VLLMs, while being compact and easy to be use without prohibitive resource requirements.

| Dataset | Capabilities Tested | Train Set | Test Set | Size (GB) |
|---|---|---|---|---|
| Fast Open MiniImageNet | I2T, Fast Binding | 5,000 | 200 | 0.18 |
| CLEVR Count Induction | I2T, Fine Grained Perception, Induction | 800 | 200 | 0.18 |
| Operator Induction | I2T, Induction, Reasoning | 80 | 60 | 0.01 |
| Interleaved Operator Induction | I2T, Induction, Reasoning, Interleaving, Long-Context | 80 | 60 | 0.01 |
| TextOCR | I2T, Fine Grained Perception, Induction | 800 | 200 | 0.98 |
| Matching MiniImageNet | I2T, Induction, Interleaving, Long-Context | 1,600 | 400 | 0.11 |
| Text-to-image MiniImageNet | T2I, Fast Binding | 5,000 | 200 | 0.18 |
| CoBSAT | T2I, Induction | 800 | 200 | 0.07 |
| Fast Counting | T2I, Fast Binding | 800 | 40 | 0.03 |
| Fast Attribute Matching | T2I, Fast Binding | 300 | 200 | 0.07 |
| Total | T2I, I2T, Binding, Perception, Long-Context, Interleaving, Induction, Reasoning | 15,260 | 1,760 | 1.82 |

pactness. We follow the typical protocol of the ICL community (Dong et al., 2024; Tsimpoukelli et al., 2021; Min et al., 2022)[2] and split each dataset into train and test splits. Few-shot ICL is then performed/evaluated by sampling the support/context set from the training split, and the test/query examples from testing split. The final performance is the average of a number of such ICL episodes.

**Fast Open MiniImageNet** We use the variant of MiniImageNet few-shot object recognition (Vinyals et al., 2016) repurposed for ICL in Tsimpoukelli et al. (2021). In open-ended classification, VLLMs must name objects based on a few examples, rather than classifying them into predefined categories, making the chance rate effectively zero. Fast-binding tasks test models' ability to associate novel names or symbols with concepts, without prior knowledge. Tsimpoukelli et al. (2021) assign synthetic names (e.g., dax or perpo) to object categories, and the model must learn these associations to name test images. We use the two-way version of this task.

**CLEVR Count Induction** In this dataset, models must learn to solve tasks like `"How many red objects are there in the scene?"` from examples rather than explicit prompts. We input CLEVR scene images (Johnson et al., 2017) along with an *attribute: value* pair identifying a specific object type based on four attributes: size, shape, color, or material. The output $y$ is the count of objects matching the attribute. To succeed, the model must ground the attribute in the image, differentiate object types, and induce that the required operation is counting.[3].

**Operator Induction** In this image-to-text task, models must solve tasks of the type `2 ? 7 = 9` given training examples like `1 ? 3 = 4`. I.e., besides parsing an image $x$ to extract the numbers and operator, models need to induce that the role of the unknown operator is addition, and then conduct arithmetic reasoning on parsed test examples. Available mathematical operations are plus, minus and times, and we only consider single digit numbers. We generate our own images for this task.

**Interleaved Operator Induction** This task tests the ability of models to reason over multiple images within $x$ to produce a single answer $y$. In this variation of operator induction we give the model two query images as input containing each number in the expression, rather than a single image containing the whole expression, as above. While separating the images simplifies perception difficulty of parsing expressions, it also increases difficulty by requiring reasoning between two

---

[2]This is is different than the few-shot meta-learning community (Wang et al., 2020; Hospedales et al., 2021), which samples support/query sets from the same pool.

[3]This task could be performed zero-shot with a suitably detailed VQA prompt. But the goal is to test whether models can learn the task from a few examples by ICL.

images, whereas VLLMs are typically trained on single-image tasks. Additionally, multiple images increase the token count, challenging the VLLMs' ability to handle larger context lengths.

**TextOCR**    We repurpose the TextOCR dataset (Singh et al., 2021) to create a task where the model should learn to output the text shown in the red rectangle, as inspired by (Sun et al., 2024a). Images $x$ contain an image with a window of text highlighted, and outputs $y$ are the OCR text. This task could be achieved by a suitably detailed zero-shot prompt, but unlike Sun et al. (2024a), we focus on evaluating whether the task can be induced by way of example through ICL. Thus this task tests both fine-grained perception and induction capabilities.

**Fast Matching MiniImageNet**    This task is the simplest example of supervised learning of a relation between two images. For relation learning, inputs contain an image pair $x = \{x_1, x_2\}$, and output $y$ indicates whether a specific relation $r$ holds between them. VLLMs are required to learn the relation from examples where it does ($r(x_1, x_2) = true$) and does not ($r(x_1, x_2) = false$) hold, using artificial keywords to name the relation. We re-use MiniImageNet (Tsimpoukelli et al., 2021; Vinyals et al., 2016) and the relation to learn is whether the two images come from the same class or not (Sung et al., 2018). This task tests induction, long-context, and multiple interleaved images.

**Fast Text-to-Image MiniImageNet**    We introduce a further variation of MiniImageNet (Tsimpoukelli et al., 2021; Vinyals et al., 2016), which inputs synthetic category names $x$ (for fast binding), and outputs images $y$. The model should learn from the context set to associate synthetic names with distributions over images, and thus learn to generate a new image of the corresponding category when prompted with the artificial category name. This task tests image generation and fast binding.

**CoBSAT**    We also utilize a recent text-to-image CoBSAT (Zeng et al., 2024) benchmark as part of our larger VL-ICL Bench suite (Figure 1). This is a text-to-image task where the model must learn to synthesise images $y$ of a specified text concept $x$ (e.g., object category), but furthermore there is a latent variable common to the context set examples that must be induced and correctly rendered in novel testing images (e.g., common color of objects). This task tests image generation and latent variable induction. We also use selected CoBSAT images for Fast Counting and Fast Attribute tasks.

**Fast Counting**    In the fast counting task we associate artificial names with counts of objects in the image. The task is to generate an image that shows a given object in quantity associated with the keyword (e.g. *perpo dogs* where perpo means two). We use counts between one and four to make the task solvable, and ask the model to distinguish between two counts in a task.

**Fast Attribute Matching**    Similarly to the other tasks, we ask the model to learn to associate artificial names to a concept, in this task to attributes (e.g. color red). We then ask the model to generate an image of an object with the given attribute, when given the name of the object and the keyword associated with the attribute. We show the model two attributes in the support set.

**Capability Summary**    The VL-ICL Bench suite described above goes far beyond any individual existing ICL benchmark to test diverse capabilities of multi-modal ICL including (Table 1): Both *text-to-image* and *image-to-text* generation; *fast-binding* – the ability to rapidly ground new symbols to visual concepts and re-use those symbols in the context of new data; *fine-grained perception* – as required to count or read text; *interleaving* – the ability to reason over the content of multiple images when generating a single output; *rule induction* – inducing non-trivial concepts such as mathematical operators and latent variables from examples; simple *reasoning* such as arithmetic; and *long-context* – the ability of a VLLM to usefully exploit a large number of context tokens.

**Text Variation**    In order to compare the impact of multimodality we also include text-version alternatives for our tasks. For datasets such as open-ended MiniImageNet, instead of images we provide image captions and use those for reasoning. For example, in CLEVR we provide enumeration of the objects in the scene, including their attributes. Note that text versions are not practical for all of the tasks, in particular TextOCR is difficult to translate into a suitable text alternative.

## 4    RESULTS

### 4.1    EXPERIMENT SETUP

**Models**    We evaluate a diverse family of state-of-the-art models with various sizes (ranging from 0.5B to 80B) and different LLM backbones on our benchmark. Specifically, for image-to-text

VLLMs, we select Open Flamingo (9B) (Awadalla et al., 2023), IDEFICS (9/80B) (Laurençon et al., 2023), IDEFICS-v2 (8B) (Laurençon et al., 2024), Otter (9B) (Li et al., 2023b), InternLM-XComposer2 (7B) (Zhang et al., 2023a), LLaVA-Next (Vicuna-7B) (Liu et al., 2024b), Qwen-VL-Chat (9B) (Bai et al., 2023), Emu2-Chat (34B) (Sun et al., 2024a), VILA (7B) (Lin et al., 2024), Mantis (-Idefics2) (Jiang et al., 2024a), Phi-3-Vision (4B) (Abdin et al., 2024), LongVA (Zhang et al., 2024b), InternLM-XComposer2.5 (7B) (Zhang et al., 2024a), and LLaVA-OneVision (0.5/7/72B) (Li et al., 2024a). For Text-to-image VLLMs, we use GILL (7B) (Koh et al., 2023), SEED-LLaMA (8B, 14B) (Ge et al., 2024), Emu1 (14B) (Sun et al., 2024b), Emu2-Gen (34B) (Sun et al., 2024a). We also evaluate GPT4V (Achiam et al., 2023) on our benchmark. We use officially released model weights or GPT4 API and adopt greedy decoding for reproducibility. All experiments are conducted using three different random seeds and we report the average performance.

**Evaluation Metrics**    All our experiments evaluate test accuracy as a function of the number of shots (cf: Fig. 3 and Fig. 5). To summarise these curves for easy comparison we use three main metrics: zero-shot accuracy, peak (max.) accuracy over all shots and ICL efficiency (the area under the accuracy vs shots curve above the zero-shot starting point, normalized over the whole area). For text-to-image models, we employ LLaVA-Next-7B as the judge model to determine whether the generated images are correct (e.g., whether contain the target attribute). Implementation details are detailed in Appendix A.

**Prompt**    For consistency, we employ the following standard prompt format for in-context learning.

> [Task Description]
>
> **Support Set**: [Image][Question][Answer] (n-shot)
>
> **Query**: [Image][Question]
>
> **Prediction**: [Answer]

## 4.2 MAIN RESULTS

The main results for VL-ICL Bench are illustrated for selective subsets in Fig. 5 and summarised quantitatively in Tab. 2 and Tab. 3 for I2T and T2I benchmarks respectively. We make the following observations: (1) **VLLMs demonstrate non-trivial in-context learning on VL-ICL Bench tasks.** Unlike the common VQA and captioning benchmarks (Fig. 3), our tasks have low zero-shot performance and in every task at least one model shows a clear improvement in performance with number of shots. Thus, ICL capability is now indeed being demonstrated and exploited. (2) **VLLMs often struggle to make use of a larger number of ICL examples.** For several tasks and models performance increases with the first few shots; but the increase is not monotonic. Performance often decreases again as we move to a larger number of shots (e.g., GPT4V CLEVR Count Induction; InternLM-XComposer2 Operator induction in Fig. 5). More extremely, some models obtain negative impact from more shots, leading to negative ICL efficiency in Tab. 2. We attribute this to difficulty of dealing with the larger number of images and tokens confusing the model and overwhelming the value of additional training data. It is exacerbated by the difficulty of extrapolation over context length and number of input images, which for higher-shot ICL becomes greater than the context length and image number used for VLLM training. This shows an important limit of the current state-of-the-art in ICL: Future models must support longer contexts and more images to benefit from larger support sets. (3) **LLaVA-OneVision 72B is the best overall image-to-text model, closely followed by GPT4V**. (4) **Zero-Shot Performance is not strongly indicative of ICL ability.** For example LLaVA-Next-7B (Liu et al., 2024b) is perhaps one of the worst overall on VL-ICL Bench, which is surprising as it is a strong model in mainstream zero-shot benchmarks. This is due to point (2): Its training protocol uses one image at a time, and it uses a large number of tokens per image – thus ICL requires it to extrapolate substantially in input image number and token number, which it fails to do. (5) There is **No clear winner among text-to-image models.** However, text-to-image models have more consistent shot scaling than image-to-text models. This is due to training with more diverse interleaved datasets that provide multiple input images per instance, and using fewer tokens per image for better scaling.

We remark that our zero-shot and ICL efficiency summary metrics (Tabs 2, 3) can sum to one at most. These metrics are visualised together in Fig. 14, along with the best possible Pareto-front. The differing degree to which each model/dataset relies on ZS vs ICL performance is easily visible.

Table 2: Average zero-shot, peak and efficiency scores (%, ↑) of different models on VL-ICL Bench image-to-text datasets. LLaVA-OneVision-72B and GPT4V show the best ICL abilities.

| Model | Fast Open-Ended MiniImageNet | | | CLEVR Count Induction | | | Operator Induction | | | TextOCR | | | Interleaved Operator Induction | | | Fast Matching MiniImageNet | | | Avg. Rank | | |
|---|---|---|---|---|---|---|---|---|---|---|---|---|---|---|---|---|---|---|---|---|---|
| | Z.s. | Pk. | Eff. | Z.s. | Pk. | Eff. | Z.s. | Pk. | Eff. | Z.s. | Pk. | Eff. | Z.s. | Pk. | Eff. | Z.s. | Pk. | Eff. | Z.s. | Pk. | Eff. |
| OpenFlamingo-9B | 0.0 | 58.2 | 46.2 | 0.0 | 18.8 | 16.6 | 5.0 | 7.8 | -1.1 | 0.0 | 0.0 | 0.0 | 0.0 | 8.9 | 4.7 | 0.0 | 29.1 | 20.2 | 9.3 | 13.3 | 8.7 |
| IDEFICS-9B | 0.0 | 59.2 | 42.1 | 0.0 | 30.3 | 26.5 | 11.7 | 14.4 | -1.5 | 16.5 | 28.0 | 6.6 | 15.0 | 15.0 | -8.7 | 0.0 | 0.1 | 0.0 | 7.7 | 10.8 | 8.7 |
| IDEFICS-80B | 0.0 | 62.5 | 42.5 | 0.0 | 32.4 | 29.6 | 13.3 | 21.7 | 4.3 | 20.0 | 29.5 | 6.1 | 25.0 | 36.7 | 2.7 | 0.0 | 28.3 | 22.3 | 7.0 | 7.8 | 5.8 |
| IDEFICS2-8B | 0.0 | 61.5 | 37.3 | 2.0 | 51.5 | 43.0 | 36.7 | 47.2 | 4.6 | 23.0 | 30.3 | 3.5 | 38.3 | 38.3 | -18.1 | 0.0 | 25.2 | 10.3 | 3.0 | 5.5 | 8.5 |
| Otter | 0.0 | 28.5 | 20.6 | 0.0 | 8.3 | 5.3 | 21.7 | 21.7 | -10.0 | 0.0 | 0.8 | 0.5 | 8.3 | 9.4 | -1.0 | 0.0 | 0.0 | 0.0 | 8.0 | 14.5 | 13.0 |
| InternLM-X2 | 0.0 | 50.3 | 34.1 | 1.8 | 26.0 | 19.4 | 26.1 | 40.0 | 10.0 | 8.7 | 16.0 | 3.3 | 28.3 | 28.3 | -18.2 | 0.0 | 49.9 | 25.3 | 5.5 | 9.3 | 10.0 |
| Qwen-VL-Chat | 0.0 | 58.0 | 37.2 | 0.0 | 30.2 | 26.1 | 15.0 | 25.0 | 3.8 | 4.8 | 24.2 | 16.1 | 16.7 | 16.7 | -8.2 | 0.0 | 0.3 | 0.1 | 7.7 | 10.8 | 8.3 |
| LLaVA-Next-7B | 0.0 | 37.2 | 29.4 | 0.0 | 25.2 | 14.5 | 10.6 | 10.6 | -6.8 | 24.7 | 24.7 | -23.0 | 13.9 | 13.9 | -7.8 | 0.0 | 0.0 | 0.0 | 7.3 | 13.7 | 13.2 |
| Emu2-Chat | 0.0 | 29.3 | 21.6 | 5.3 | 17.7 | 9.1 | 28.6 | 28.6 | -6.7 | 25.8 | 36.5 | 5.7 | 26.7 | 26.7 | -13.2 | 0.0 | 40.2 | 32.9 | 4.0 | 10.2 | 11.0 |
| VILA-7B | 0.0 | 38.2 | 32.3 | 3.5 | 34.3 | 27.5 | 28.3 | 28.3 | -18.9 | 28.0 | 30.2 | -3.7 | 28.3 | 28.3 | -15.7 | 0.0 | 49.9 | 44.6 | 3.7 | 8.0 | 10.8 |
| Mantis-Idefics2 | 0.0 | 84.3 | 55.2 | 31.4 | 40.2 | -3.9 | 16.7 | 16.7 | -2.9 | 21.0 | 27.8 | 4.1 | 30.0 | 30.0 | -22.3 | 0.0 | 51.1 | 43.8 | 4.5 | 6.7 | 9.7 |
| Phi3-Vision | 0.0 | 50.0 | 33.3 | 0.0 | 34.5 | 20.4 | 0.0 | 54.4 | 46.8 | 8.0 | 41.5 | 28.9 | 0.0 | 27.8 | 18.9 | 0.0 | 50.8 | 42.4 | 9.2 | 6.3 | 5.3 |
| LongVA-7B | 0.0 | 52.7 | 39.9 | 0.0 | 29.2 | 21.4 | 18.3 | 18.3 | -11.9 | 26.0 | 26.0 | -11.0 | 10.0 | 10.0 | -5.3 | 0.0 | 50.0 | 45.0 | 6.3 | 10.5 | 9.3 |
| LLaVA-OneVision-72B | 0.0 | 98.7 | 68.4 | 0.5 | 42.3 | 32.7 | 33.3 | 75.6 | 29.8 | 48.5 | 51.7 | 2.4 | 38.3 | 47.8 | 2.8 | 0.0 | 70.3 | 50.8 | 2.5 | **1.7** | **3.8** |
| InternLM-X2d5 | 0.0 | 91.0 | 62.7 | 6.0 | 63.2 | 47.5 | 36.7 | 41.7 | 0.3 | 32.0 | 42.5 | 4.2 | 38.3 | 38.3 | -8.0 | 0.0 | 46.8 | 26.1 | **1.5** | 3.7 | 5.8 |
| GPT4V | 0.0 | 78.0 | 41.8 | 6.0 | 42.0 | 29.0 | 24.0 | 92.0 | 59.0 | 39.3 | 50.0 | 7.2 | 36.0 | 74.0 | 32.2 | 0.0 | 58.2 | 38.3 | 2.8 | 2.3 | **3.8** |

Table 3: Average zero-shot, peak and efficiency scores (%, ↑) of different models on VL-ICL Bench text-to-image datasets.

| Model | Text-to-Image Fast MiniImageNet | | | CoBSAT | | | Fast Counting | | | Fast Attribute Matching | | | Avg. Rank | | |
|---|---|---|---|---|---|---|---|---|---|---|---|---|---|---|---|
| | Z.s. | Pk. | Eff. | Z.s. | Pk. | Eff. | Z.s. | Pk. | Eff. | Z.s. | Pk. | Eff. | Z.s. | Pk. | Eff. |
| GILL | 0.0 | 16.0 | 13.6 | 2.7 | 12.3 | 7.1 | 34.2 | 34.2 | -26.3 | 20.5 | 20.5 | -13.2 | 3.5 | 4.8 | 4.8 |
| SEED-LLaMA-8B | 0.0 | 16.5 | 13.3 | 0.5 | 33.7 | 24.5 | 34.2 | 56.7 | 12.7 | 21.0 | 34.3 | 10.5 | 3.5 | 2.5 | **2.2** |
| SEED-LLaMA-14B | 0.8 | 21.2 | 16.3 | 5.5 | 43.8 | 30.7 | 41.3 | 51.6 | 1.9 | 22.4 | 35.6 | 8.4 | 2.0 | **2.0** | 2.5 |
| Emu1-Gen | 0.5 | 31.5 | 22.5 | 0.3 | 9.7 | 7.1 | 33.2 | 47.8 | 8.2 | 23.5 | 29.1 | 3.1 | 3.5 | 3.8 | 2.8 |
| Emu2-Gen | 0.0 | 37.0 | 28.6 | 8.7 | 28.7 | 15.6 | 44.2 | 59.2 | 5.2 | 26.5 | 34.0 | -0.2 | **1.5** | **2.0** | 2.8 |

## 4.3 ADDITIONAL ANALYSIS

We next use VL-ICL Bench to analyse several challenges and factors influencing ICL performance.

**Fast Concept Binding** In our open miniImageNet task, we follow Tsimpoukelli et al. (2021) to require fast-binding of synthetic concept names so as to purely test models' ICL ability, without confounding by VLLMs' zero-shot ability to associate visual concepts with names. Tab. 4 compares the fast and real-world miniImageNet recognition. Our fast-binding case is much more challenging.

**Direct comparison of multimodal and text ICL** We can disentangle the role of text versus image inputs for some image-to-text VL-ICL Bench tasks, where we can easily provide a semantically equivalent text input describing the image, in place of image tokens. Fig. 6 shows a comparison between image-input vs text-input for count induction, operator induction, and interleaved operator induction tasks. With text-input, performance grows much more sharply and consistently with number of shots. This is attributable to both (i) reduction of perception difficulty, and (ii) reduction in the total number of tokens compared to image input.

**Disentangling context length and in-context learning** One direct factor limiting VL ICL is the model's context length, as images translate to a large number of tokens (e.g., in LLaVA, one image translates to 576 tokens, causing an 8-shot setting to exceed a 4k context window). This makes it difficult to disentangle whether the limitation on VL-ICL performance arises from context length constraints or inherent challenges in the models' ability to perform ICL. To further understand this issue, we apply SelfExtend (Jin et al., 2024), a training-free position encoding extrapolation strategy that has been shown to effectively extend the context length of LLMs. We use SelfExtend to extend the context length of LLaVA and VILA from 4k to 16k. As shown in Tab. 5 (Fig. 15 in the Appendix), though it can help in some cases, performance often remains similar, suggesting

Table 4: Average zero-shot, peak and efficiency scores (%, ↑) on fast open-ended and real-world MiniImageNet dataset. Real-world version is much easier, with larger zero-shot and peak accuracies.

| Dataset | IDEFICS-80B | | | Otter | | | InternLM-X2 | | | Qwen-VL-Chat | | | LLaVA-Next-7B | | | GPT4V | | |
|---|---|---|---|---|---|---|---|---|---|---|---|---|---|---|---|---|---|---|
| | Z.s. | Pk. | Eff. | Z.s. | Pk. | Eff. | Z.s. | Pk. | Eff. | Z.s. | Pk. | Eff. | Z.s. | Pk. | Eff. | Z.s. | Pk. | Eff. |
| Fast Open-Ended | 0.00 | 62.50 | 42.54 | 0.00 | 28.50 | 20.62 | 0.00 | 50.33 | 34.10 | 0.00 | 58.00 | 37.22 | 0.00 | 33.67 | 14.57 | 0.00 | 78.00 | 41.80 |
| Real-World | 30.50 | 94.67 | 43.05 | 13.00 | 61.00 | 38.75 | 20.00 | 67.00 | 35.12 | 32.17 | 88.33 | 30.81 | 20.50 | 64.50 | 13.43 | 48.00 | 90.00 | 27.00 |

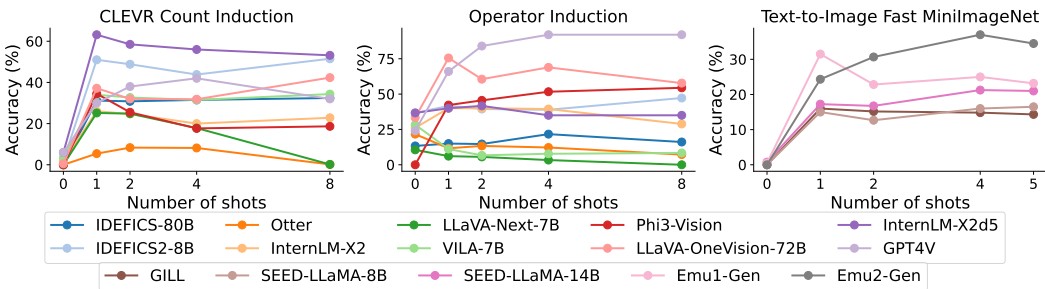

Figure 5: Illustrative VL-ICL Bench results. Our tasks better exploit and evaluate ICL compared to the mainstream in Fig. 3. They have lower zero-shot accuracy and more substantial gains with shots.

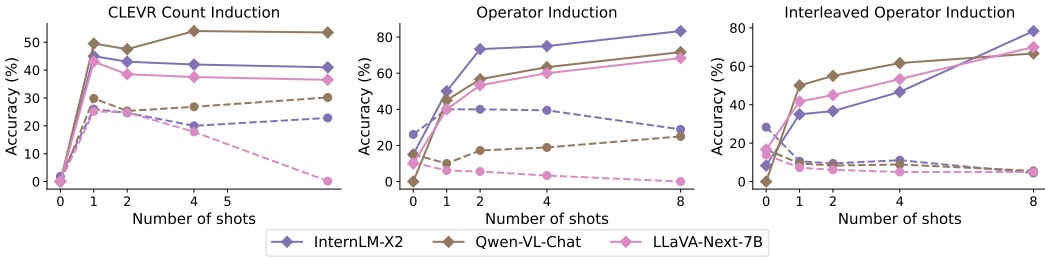

Figure 6: Comparison of multimodal (dashed line) and text (solid line). Performance increases more sharply and consistently with text inputs, highlighting the greater difficulty of multimodal ICL.

that *long-context capabilities are necessary but not sufficient for improving ICL.* Furthermore, even recent models with 128k context windows, like LLaVA-OneVision, do not always show improved performance with more shots as seen in Tab. 2. This highlights that VL-ICL is a more complex challenge, requiring an understanding of interleaved few-shot examples, which cannot be solved solely by increasing context length.

**Emergent threshold of multimodal ICL** We further investigate whether there is a threshold in model size at which emergent ICL abilities appear, as described by Wei et al. (2022a) for LLMs. To test this, we use the LLaVA-OneVision series (Li et al., 2024a), which share the same architecture and training data but differ in LLM size (0.5B, 7B, and 72B parameters). Notably, we find that the 72B model demonstrates emergent ICL abilities. As shown in Figure 7, the 72B model improves with more shots across all tasks, particularly in Interleaved Operator Induction and Fast Matching MiniImageNet, where the 0.5B and 7B models perform worse than chance as shots increase. This indicates that the 72B model understands the tasks, while the smaller models fail, highlighting the impact of model size on ICL and the presence of an emergent threshold.

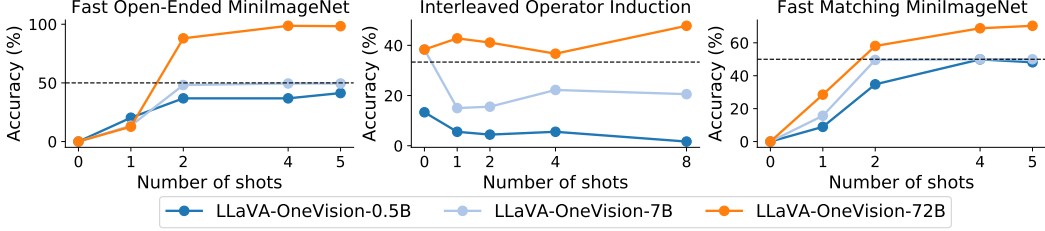

Figure 7: Comparison of different model sizes. Dashed line indicates random chance performance assuming available options (the models do open-ended generation so can be worse than random).

Additionally, we provide thorough analyses in the Appendix B and C on various factors, including scaling to many shots (up to 64), ICL efficiency versus zero-shot performance, chain-of-thought prompting, repeating the support set, varying levels of task descriptions, and qualitative results.

Table 5: Comparison of models with and without context extension strategy (SelfExtend). While it is helpful in some cases, context extension does not necessarily improve the performance of ICL.

| Dataset | Fast Open-Ended MiniImageNet | | | CLEVR Count Induction | | | Operator Induction | | | TextOCR | | |
|---|---|---|---|---|---|---|---|---|---|---|---|---|
| | Z.s. | Pk. | Eff. | Z.s. | Pk. | Eff. | Z.s. | Pk. | Eff. | Z.s. | Pk. | Eff. |
| LLaVA-Next-7B (w/o SelfExtend) | 0.0 | 37.2 | 29.4 | 0.0 | 25.2 | 19.3 | 10.6 | 10.6 | -6.8 | 24.7 | 24.7 | **-23.0** |
| LLaVA-Next-7B (w/ SelfExtend) | 0.0 | **51.0** | **38.9** | 0.0 | **29.0** | **25.4** | **11.7** | **11.7** | **-5.8** | **26.0** | **26.0** | -23.7 |
| VILA-7B (w/o SelfExtend) | 0.0 | 38.2 | 32.3 | 3.5 | 34.3 | **27.5** | **28.3** | **28.3** | **-18.9** | 28.0 | **30.2** | **-3.7** |
| VILA-7B (w/ SelfExtend) | 0.0 | **54.0** | **40.0** | **4.0** | **34.8** | **27.5** | **28.3** | **28.3** | -20.3 | 28.0 | 29.7 | -4.5 |

## 5 RELATED WORK

**VLLM Evaluation**   With the rapid development of VLLMs, researchers are creating evaluation benchmarks to thoroughly assess their capabilities from diverse perspectives. These evaluations range from zero-shot aggregated benchmarks like MME (Fu et al., 2023), MMbench (Liu et al., 2024d), and MM-VET (Yu et al., 2024) to datasets designed for fine-tuning on specific aspects, such as visual reasoning (Hudson & Manning, 2019) and knowledge-grounded QA (Lu et al., 2022). They predominantly focus on single-image, leaving in-context learning evaluation underexplored.

**In-Context Learning Evaluation**   The term "in-context" has been used in a few ways, including to describe scenarios with interleaved inputs, such as multiple video frames or multi-turn conversations (Li et al., 2023a;b; Zhao et al., 2024; Ge et al., 2024). Although the study of interleaved inputs presents an intriguing subject, it does not align with the core definition of in-context learning that we consider following (Brown et al., 2020; Dong et al., 2024; Min et al., 2022), which involves the emergent ability to learn a function from $x \rightarrow y$ from few-shot support input-output pairs. Prior evaluation of ICL (Awadalla et al., 2023; Laurençon et al., 2023; Sun et al., 2024a; Baldassini et al., 2024; Chen et al., 2024) in this sense is limited, and comes with serious drawbacks as discussed in Sec. 2.2. Concurrent to our work, CobSAT (Zeng et al., 2024) introduces a benchmark designed to evaluate in-context learning in text-to-image models, focusing particularly on latent variable induction capabilities. Our work spans a much wider range of tasks and capabilities, encompasses both image-to-text and text-to-image generation and subsumes CobSAT as one of our ten total benchmarks (Tab. 1). The concurrent work of Jiang et al. (2024b) studies ICL shot-scaling, but only for I2T recognition tasks - in contrast to our wide range of task types (Tab. 1), and without elucidating the limitations of mainstream VQA/captioning paradigm as we have in Sec. 2.2.

**Visual In-Context Learning**   The term "in-context" has also been used in pure vision models, which aim to perform diverse image-to-image tasks without task-specific prediction heads (Bar et al., 2022; Wang et al., 2023a;b), such as semantic segmentation, depth estimation, object detection, etc. However, these models are explicitly trained on paired in-context input-output data to be able to perform visual ICL during inference. In this paper, we focus on multimodal vision-language ICL, which is based on the emergent ability of LLMs.

## 6 DISCUSSION

We have introduced the first comprehensive benchmark suite *VL-ICL Bench* for multi-modal vision-and-language in-context learning. This benchmark suite avoids the issue with the existing mainstream but limited approach to evaluating image-to-text ICL – that ICL provides limited demonstrable benefit over zero-shot inference, and VLLMs learn answer formatting at best rather than any true multi-modal capability. In contrast, VL-ICL Bench tests a wide variety of multi-modal capabilities including both text-to-image and image-to-text generation, fine-grained perception, rule-induction, reasoning, image interleaving, fast concept binding, long context, and shot scaling. We hope this benchmark will inspire model developers to consider all these capabilities in VLLM development, and inform practitioners about the evolution of what VLLM ICL can and cannot do as the field develops. One limitation of this work is that we evaluate only a small number of text-to-image models, in contrast to the more comprehensive set of image-to-text models. This is due to the limited availability of VLLMs capable of handling interleaved inputs and generating images. Developing such models presents a promising direction for future research.

## 7 ACKNOWLEDGEMENT

Yongshuo Zong is supported by the United Kingdom Research and Innovation (grant EP/S02431X/1), UKRI Centre for Doctoral Training in Biomedical AI at the University of Edinburgh, School of Informatics. For the purpose of open access, the author has applied a creative commons attribution (CC BY) licence to any author accepted manuscript version arising.

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

APPENDIX

**Table of Content**

## A    IMPLEMENTATION AND EVALUATION DETAILS

**VL-ICL Bench Evaluation Metrics**    We use accuracy as the metric across all subsets in our benchmark. For text-to-image generation tasks, we utilize the state-of-the-art VLLM LLaVA-Next (Liu et al., 2024b) as the judge model to decide whether the generated images contain the required object or attribute.

**Models Configurations**    We additionally provide a summary of the configurations for the models benchmarked in our paper in Table 6, with a particular focus on the number of tokens per image and the context length. This information helps elucidate why some models exhibit poor scalability with increasing shots, as the total lengths exceed the maximum context window.

Table 6: Detailed configurations of the models used in our benchmark.

| Model | Connection Module | Image Tokens | Context Length (Train) | Context Length (Test) |
|---|---|---|---|---|
| OpenFlamingo-9B | Perceiver | 64 | 2048 | 2048 |
| IDEFICS-9B | Perceiver | 64 | 2048 | 2048 |
| Otter | Perceiver | 64 | 2048 | 2048 |
| InternLM-XComposer2 | Perceiver | 64 | 2048 | 4096 |
| Qwen-VL-Chat | Cross-Attention | 256 | 2048 | 8192 |
| LLaVA-Next | MLP | 576 | 2048 | 4096 |
| Emu1 | C-Former | 512 | 2048 | 2048 |
| Phi3-Vision | MLP | AnyRes | - | 128K |
| LongVA | MLP | UniRes | - | 200K+ |
| IDEFICS2 | MLP | 64 | - | 32K |
| LLaVA-OneVision | MLP | AnyRes | - | 128K |
| InternLM-X2d5 | Partial-LoRA | AnyRes | 24K | 96K |
| Emu2 | Linear layers | 64 | 2048 | 2048 |
| GILL | Linear layers | 4 | 2048 | 2048 |
| SEED-LLaMA | Q-Former | 32 | 2048 | 4096 |

**Prompts**    We list specific prompts below that we use for specific experiments.

> **Prompt to judge image generation for Fast MiniImageNet and CobSAT dataset**
>
> **User:** Decide whether the image contains the following concept: {*GT*}. Answer with 'yes' or 'no'.

> **Prompt to judge the answer for Vizwiz VQA.**
>
> **User:** Based on the image and question, decide whether the predicted answer has the same meaning as the ground truth. Answer with 'yes' or 'no'. Question: {*Question*} Predicted answer: {*Prediction*} Ground Truth: {*GT*}

**Prompt to rate the quality of COCO captioning (Main text, section 2.2)**

**User:** Given the following image, you are to evaluate the provided generated caption based on its relevance, accuracy, completeness, and creativity in describing the image. Rate the caption on a scale from 1 to 10, where 10 represents an exceptional description that accurately and completely reflects the image's content, and 1 represents a poor description that does not accurately describe the image.

Generated Caption: {*Prediction*}

Ground Truth Caption: {*GT*}

Consider the following criteria for your rating:

1 (Very Poor): The caption does not correspond to the image's content, providing incorrect information or irrelevant descriptions. It misses essential elements and may introduce non-existent aspects.

3 (Poor): The caption only slightly relates to the image, missing significant details or containing inaccuracies. It acknowledges some elements of the image but overlooks key aspects.

5 (Fair): The caption provides a basic description of the image but lacks depth and detail. It captures main elements but misses subtleties and may lack creativity or precision.

7 (Good): The caption accurately describes the main elements of the image, with some attention to detail and creativity. Minor inaccuracies or omissions may be present, but the overall description is sound.

8 (Very Good): The caption provides a detailed and accurate description of the image, with good creativity and insight. It captures both essential and minor elements, offering a well-rounded depiction.

9 (Excellent): The caption delivers an accurate, detailed, and insightful description, demonstrating high creativity and a deep understanding of the image. It covers all relevant details, enhancing the viewer's perception.

10 (Exceptional): The caption offers a flawless description, providing comprehensive, accurate, and highly creative insights. It perfectly aligns with the image's content, capturing nuances and offering an enhanced perspective.

Please provide your rating. You should ONLY output the score number.

**Details on Dataset Sampling and Filtering**    We include additional details on how we performed dataset sampling and filtering when creating our benchmark:

- Fast Open-Ended MiniImageNet: we take the first 200 query examples provided in the original open-ended MiniImageNet, together with their associated support examples.

- CLEVR: we randomly sample 800 examples from the original CLEVR dataset training scenes to use as support examples, and we select 200 examples randomly from the validation split as query examples.

- Operator Induction: we generate this dataset ourselves and we consider all single digit combinations. We use randomly selected 80 combinations of digits as the support set and 20 combinations of digits as the query set. Each combination is used with 3 different operators.

- TextOCR: we use 800 randomly selected examples from the original TextOCR training set as support examples and 200 randomly selected examples from the validation set as query examples. Before selecting these we did filtering to ensure we use only valid texts that are not marked as rotated. To make the task manageable, we select the largest text in the image for OCR.

- Interleaved Operator Induction: same as for the operator induction task.

- Fast Matching MiniImageNet: this is derived from our Fast Open-Ended MiniImageNet, using the same underlying data.

- Text-to-Image Fast MiniImageNet: we repurpose our Fast Open-Ended MiniImageNet for this task, so we use the same underlying data.
- CoBSAT: take the first 80% of the scenarios from the original CoBSAT dataset for the support set and the remaining 20% for the query set.
- Fast Counting: take action and colour categories from CoBSAT, using half of animals and objects for the support examples and another half for the query examples. Create composed images by repeating the image one, two, three or four times.
- Fast Attribute Matching: take all attribute-object pairs from CoBSAT and use 30 of them as support and 20 as query sets, with no overlaps across objects between the sets.

## B  FURTHER ANALYSIS

### B.1  FULL VL-ICL BENCH RESULTS

We show the main results across all datasets in Fig. 8.

### B.2  SCALING TO MORE SHOTS

To examine the maximum number of shots the models can handle and whether the model can still benefit from more shots, we further increase the support set size to 16, 32, and 64 shots. We choose three models for this experiment: OpenFlamingo 9B (Awadalla et al., 2023), IDEFICS-9B-Instruct (Laurençon et al., 2023), InternLM-XComposer2 (Zhang et al., 2023a), LongVA (Zhang et al., 2024b), and Mantis-Idefics2 (Jiang et al., 2024a). These models were selected because each image they process translates to fewer tokens (Table 6) or they have a longer context window, ensuring that they do not exceed the maximum context length when evaluated with 64 shots. IDEFICS-9B-Instruct demonstrates a better scaling capability compared to other models in most of the datasets. Besides, while InternLM-XComposer2 has strong performance in a low-shot regime, the performance quickly decreases with many shots. This may be due to the mismatch between training (4096) and testing (2048) context length (Table 6) where the extrapolation of context length has been a well-known challenging task (Press et al., 2022; Liu et al., 2024c).

### B.3  CHAIN-OF-THOUGHT PROMPTING

To investigate whether there is any strategy that can enhance in-context learning, one straightforward method is Chain-of-Thought (CoT) prompting (Wei et al., 2022b). CoT prompts the model to articulate its reasoning process concerning latent variables from the support set, potentially improving its learning and inference capabilities. We experiment with Qwen-VL-Chat (Bai et al., 2023) and InternLM-XComposer2 (Zhang et al., 2023a) that have state-of-the-art LLMs with strong reasoning ability. Below is the specific prompt we use.

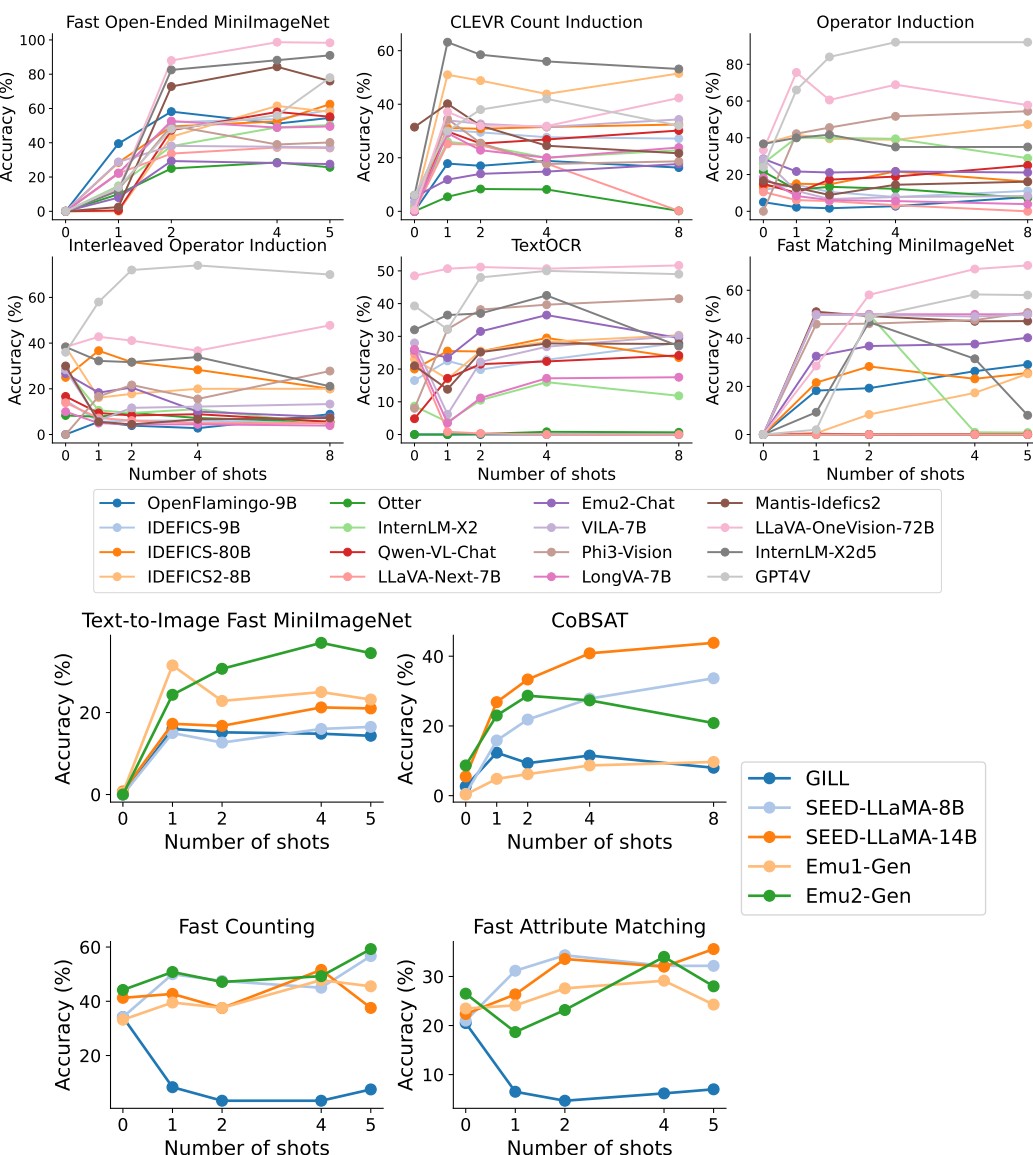

Figure 8: VL-ICL Bench results. Top: Image-to-Text. Bottom: Text-to-Image tasks.

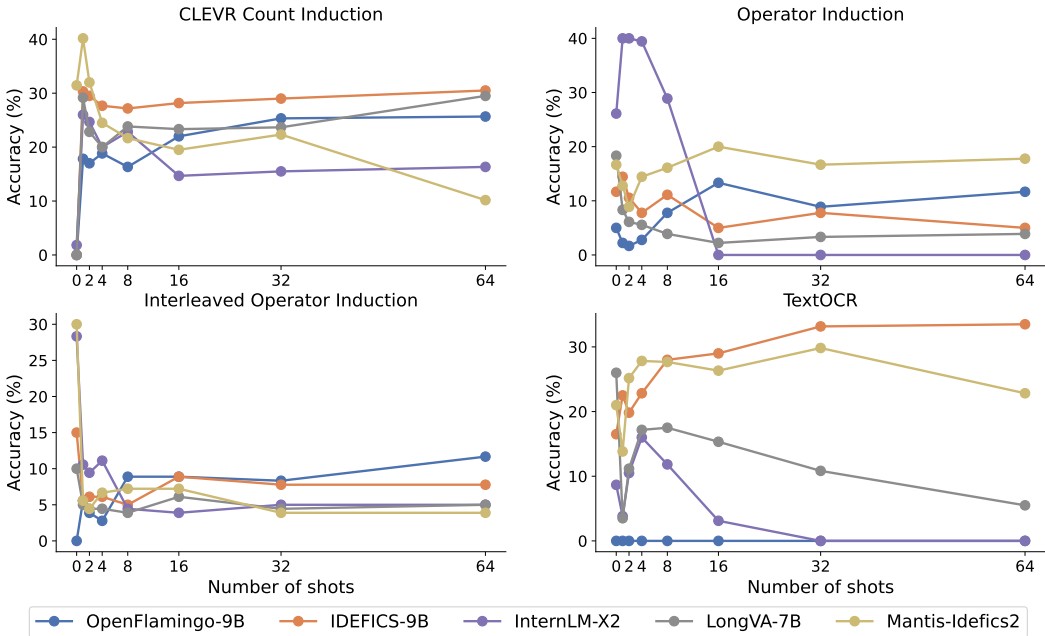

Figure 9: Results of scaling to many shots (Max 64). IDEFICS-9B-Instruct exhibits strong scaling capabilities across most datasets compared to other models. Additionally, while InternLM-XComposer2 shows strong performance in low-shot scenarios, its performance diminishes rapidly as the number of shots increases.

> [CoT Prompt]: Let's first think step by step and analyze the relationship between the given few-shot question-answer pairs. Give reasoning rationales.
>
> **User**: [Task Description][Support Set][Query][CoT Prompt]
>
> **VLLMs**: [Generated rationals]
>
> **User**: [Task Description][Support Set][Query][Generated rationals]
>
> **VLLMs**: Prediction

We do not observe a consistent improvement with chain-of-thought prompting: it benefits performance on some datasets while detracting from it on others. These findings underscore the complexity of in-context learning tasks, suggesting that fundamental advancements in model development are necessary. Such tasks cannot be readily addressed with simple prompting techniques like CoT.

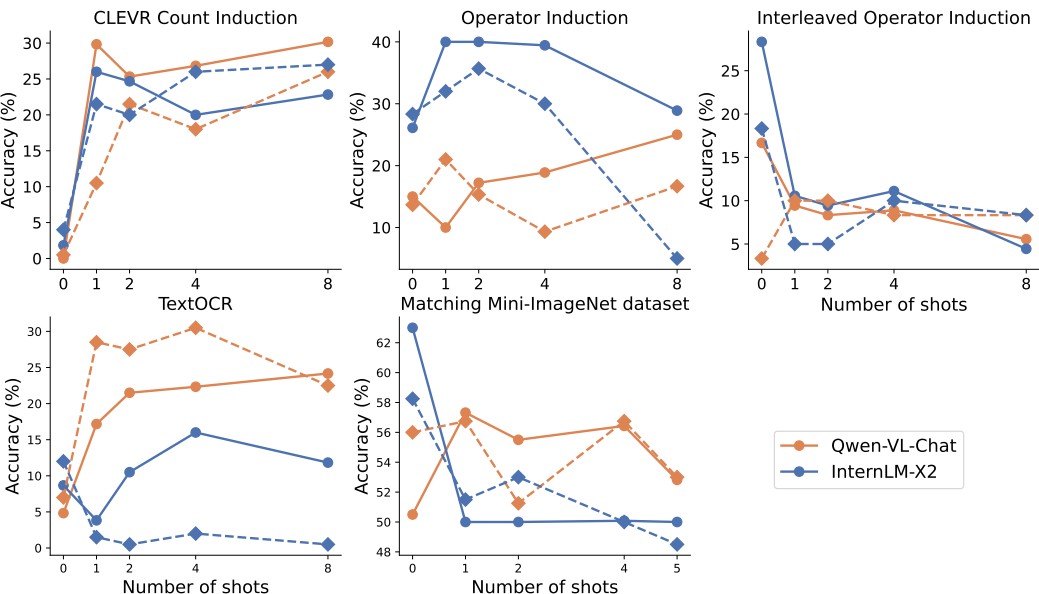

Figure 10: Comparison of Chain-of-Thought prompting (dashed line, diamond markers) with baseline results (solid line, circle markers) across a selection of datasets and models. Chain-of-thought prompting does not consistently improve performance across datasets, highlighting the complexity of in-context learning tasks and the need for fundamental model development beyond simple prompting techniques.

## B.4 REPEATING SUPPORT SET

In this subsection, we experiment with an interesting setting: we duplicate the same support example multiple times to assess whether repetition enhances performance. We employ the Qwen-VL-Chat model for these experiments, with the results presented in Figure 11. We found that duplicating shots is particularly beneficial in the 1-shot scenario for Fast Open-Ended MiniImageNet, although this is not consistently observed across other datasets. The likely reason is that Fast Open-Ended MiniImageNet gains from the reinforcement of binding the concept through repeated examples, whereas for tasks like operator induction, diverse examples are necessary to facilitate the learning process.

## B.5 INFLUENCE OF INSTRUCTION FINE-TUNING

We investigate how instruction-following fine-tuning affects in-context learning capabilities. We compare two model families, each with a pre-trained version and an instruction-following fine-tuned version: Qwen-VL versus Qwen-VL-Chat (Bai et al., 2023) and IDEFICS-9B versus IDEFICS-

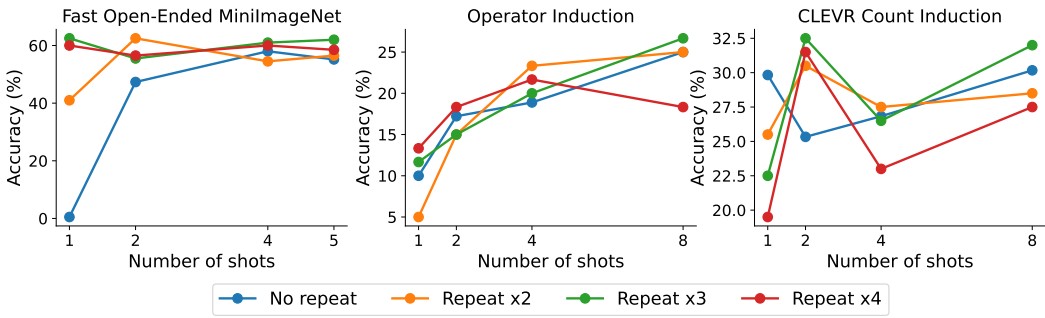

Figure 11: Investigation of the impact of repeating the in-context examples across a selection of datasets, using Qwen-VL-Chat model. The X-axis represents the number of unique shots, not the total number of shots. For example, *1-shot Repeat x2* means there is one unique shot and it is repeated twice.

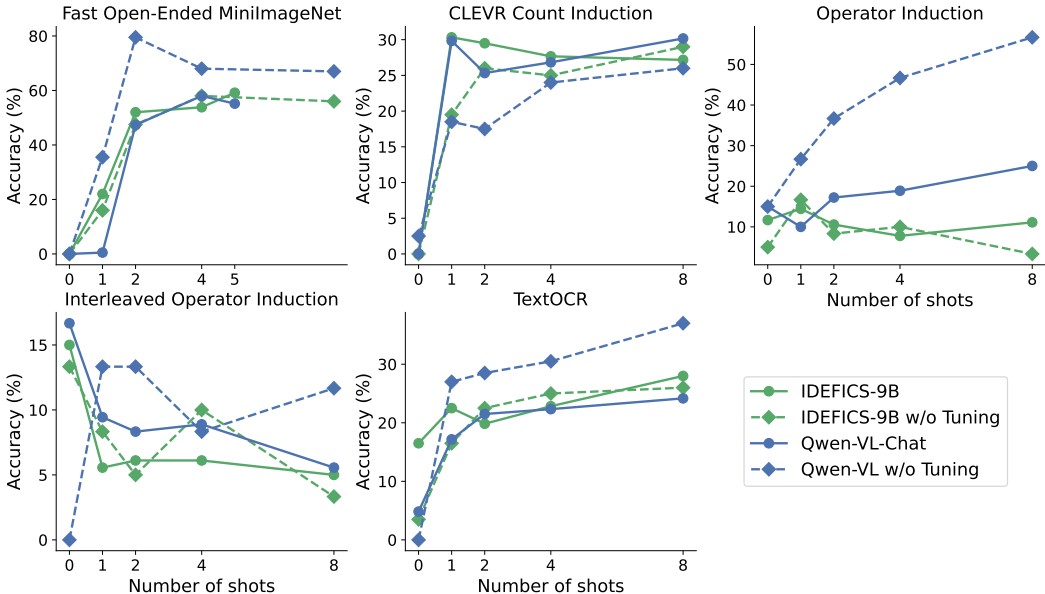

Figure 12: Comparison of using (solid line) and not using instruction tuning (dashed line). Although the outcomes vary, models not fine-tuned with instructions exhibit marginally better scalability with respect to the number of shots, as evidenced in datasets like TextOCR.

9B-Instruct (Laurençon et al., 2023). Their performance differences are illustrated in Figure 12. Although the outcomes vary, models not fine-tuned with instructions exhibit marginally better scalability concerning the number of shots, as seen with the TextOCR dataset. Further studies are needed to understand whether instruction-following fine-tuning harm the in-context ability.

## B.6 DIFFERENT LEVELS OF TASK DESCRIPTION DETAILS

We show the impact of different levels of details in the prompt description in Figure 13. The results show that generally the best results are obtained with the most detailed descriptions, but this is not necessarily the case in all settings and in some cases, even no descriptions can be better. The performance is often similar across different levels of details, but in some cases, it can be significantly worse, e.g. for TextOCR. We also provide tables with the full results. In our main experiments, we adopt detailed task descriptions for all datasets.

The task descriptions that we use for the different datasets are as follows:

**Fast Open-Ended MiniImageNet**

**Detailed**: Induce the concept from the in-context examples. Answer the question with a single word or phase.

**Concise**: Answer the question with a single word or phase.

**CLEVR Count Induction**

**Detailed**: The image contains objects of different shapes, colors, sizes and materials. The question describes the attribute and its value. You need to find all objects within the image that satisfy the condition. You should induce what operation to use according to the results of the in-context examples and then calculate the result.

**Concise**: Find objects of the given type, induce what operation to use and calculate the result.

**Operator Induction**

**Detailed**: The image contains two digit numbers and a ? representing the mathematical operator. Induce the mathematical operator (addition, multiplication, minus) according to the results of the in-context examples and calculate the result.

**Concise**: Induce the mathematical operator and calculate the result.

**TextOCR**

**Detailed**: An image will be provided where a red box is drawn around the text of interest. Answer with the text inside the red box. Ensure that the transcription is precise, reflecting the exact characters, including letters, numbers, symbols.

**Concise**: Answer with the text inside the red box.

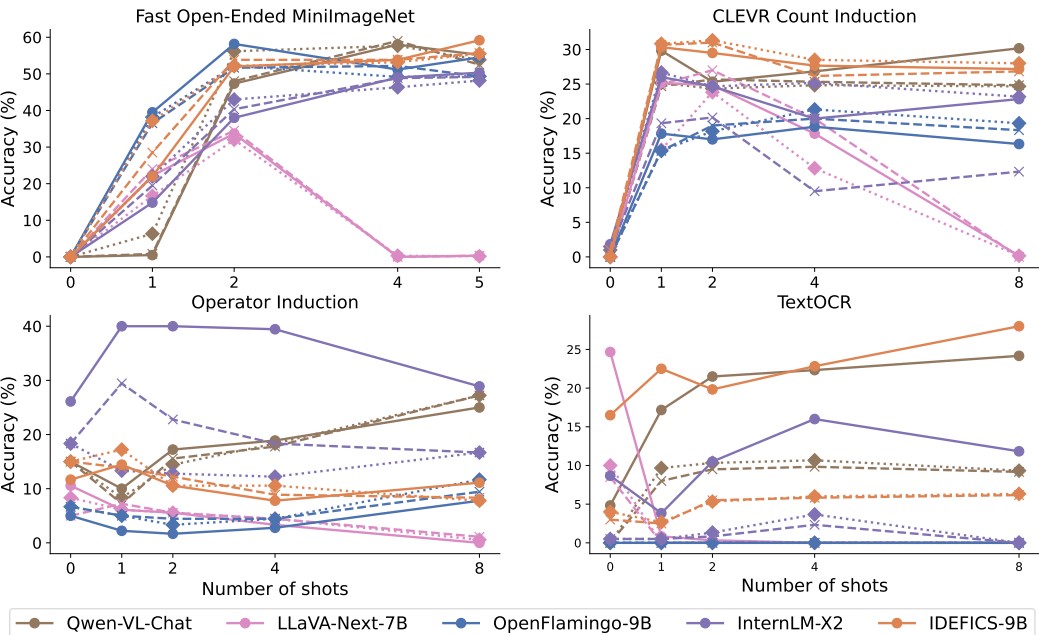

Figure 13: Comparison of detailed task description (solid line, circle markers), concise task description (dashed line, x markers) and no task description (dotted line, diamond markers) across a selection of datasets and models.

## B.7 ICL EFFICIENCY VS ZERO-SHOT PERFORMANCE

We analyse the in-context learning efficiency and compare it against the zero-shot performance. We define ICL efficiency as the ability to improve the performance after seeing a few examples. We calculate its value as a proportion of the area between the zero-shot curve and the few-shot curve, over the total available area (0-100% accuracy, max. number of shots). The results are shown in Fig. 14, where we only show models with non-negative efficiency. We also show the pareto front line that describes the constraint on the best possible models: zero-shot performance + ICL efficiency = 100%. The results indicate that there is a good variability between different models in how efficiently they can learn from the provided support examples.

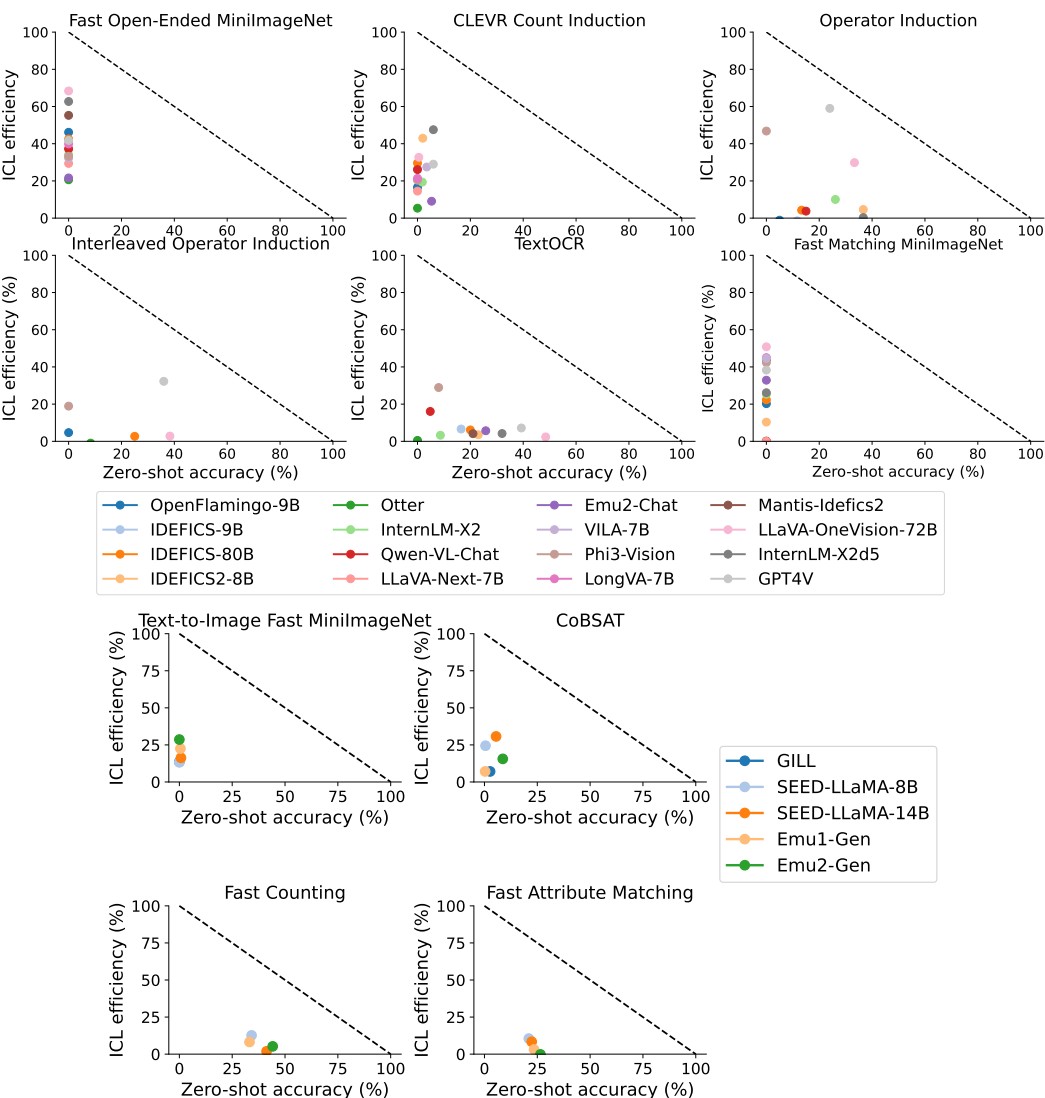

Figure 14: ICL efficiency vs zero-shot performance results. Top: Image-to-Text. Bottom: Text-to-Image tasks.

## B.8 SELF-EXTEND ANALYSIS

We analyse the impact of using self-extend mechanism to improve the ability to handle longer context. The results are shown in Figure 15 and were described earlier in the main text.

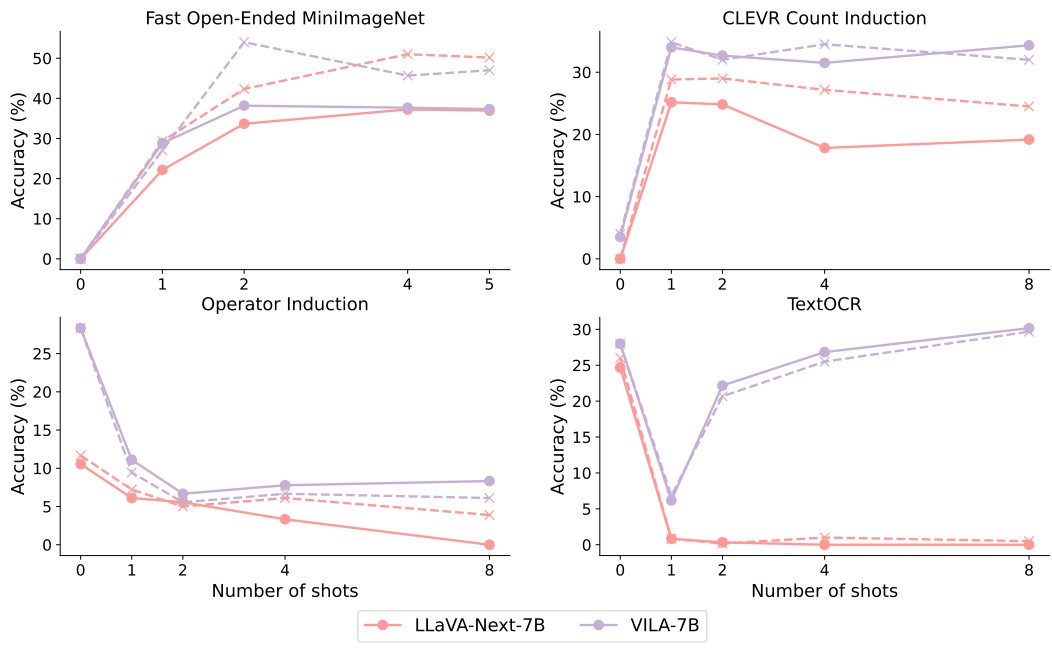

Figure 15: Comparison of models without self-extend (solid line, circle markers) and with self-extend (dashed line, x markers).

## B.9 AVERAGE PERFORMANCE OVER TIME

We analyse in Figure 16 how the average accuracy and ICL efficiency of image-to-text and text-to-image models evolved over time on our tasks. We can see that the performance as well as ICL efficiency have been continuously improving even though there are some outliers (e.g. GPT4V) that were released earlier and obtained some of the stronger performances. This shows that our benchmark can effectively assess the ICL capability of the VLMs.

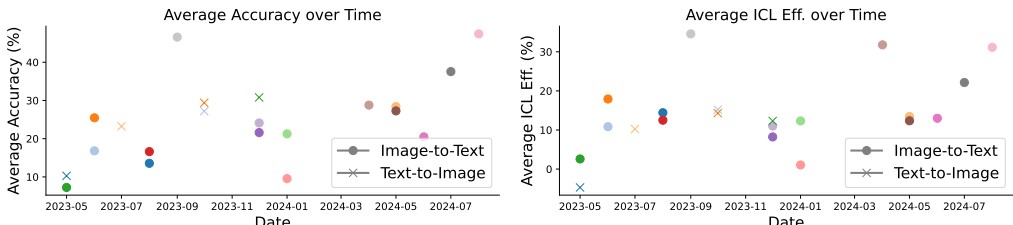

Figure 16: Average accuracy and ICL eff. in terms of when the model was released.

## B.10 IMPACT OF IN-CONTEXT DEMONSTRATION SELECTION

As shown in previous work (Zhang et al., 2023b; Li et al., 2024c; Chen et al., 2024; Yang et al., 2023), different in-context demonstration selection strategies can have a large impact on the final performance. To further understand this, we employ a similarity-based selection strategy (Zhang et al., 2023b; Li et al., 2024c; Chen et al., 2024; Yang et al., 2023). Specifically, for image-to-text tasks (CLEVR and TextOCR), we use CLIP image embeddings to retrieve the top-k most similar images and their corresponding text. Conversely, for text-to-image tasks (CoBSAT), we use CLIP text embeddings to retrieve the top-k most similar texts and their associated images. Additionally, we experiment with three different arrangements of the selected top-k demonstrations: (1) from the most similar one to the least similar one, followed by the query; (2) from the least similar one to the most similar one, followed by the query; and (3) a random order of the top-k demonstrations. These are compared to the original setup which used random selection.

We select several top-performing models for our experiments, with the results presented in Table 7 to 9. In each cell, the first three numbers correspond to the three arrangements of the selected top-k demonstrations (1), (2), and (3), while the final value represents the baseline performance with random selection. For both text-to-image and image-to-text tasks, similarity-based retrieval consistently enhances performance. Specifically, similarity-based selection methods have a ~3% improvement in peak accuracy and a ~3% boost in ICL efficiency compared to the baseline. These findings demonstrate that selection strategies can effectively enhance ICL tasks, highlighting a promising direction for future work aimed at improving performance across various tasks. Detailed analysis of the changes in peak accuracy and ICL efficiency is in Table 10.

Table 7: {Similarity-based Selection on CLEVR (Accuracy %).

| Model | 1-shot | 2-shot | 4-shot | 8-shot |
|---|---|---|---|---|
| LLaVA-OneVision -7B | 48.0 / 48.0 / 48.0 / 38.2 | 42.5 / 37.0 / 39.7 / 33.8 | 33.5 / 34.5 / 35.0 / 31.5 | 33.0 / 33.5 / 34.0 / 28.3 |
| Phi3-Vision | 42.0 / 42.0 / 42.0 / 34.5 | 31.5 / 29.5 / 27.5 / 25.5 | 29.5 / 28.5 / 32.5 / 17.7 | 36.5 / 36.5 / 33.7 / 18.7 |
| Mantis-Idefics2 | 36.0 / 36.0 / 36.0 / 40.2 | 36.5 / 35.5 / 35.3 / 32.0 | 28.5 / 29.5 / 28.5 / 24.5 | 22.5 / 21.5 / 26.3 / 21.7 |
| InternLM-X2D5 | 62.0 / 62.0 / 62.0 / 63.2 | 62.0 / 59.5 / 61.3 / 58.5 | 57.5 / 55.5 / 55.8 / 56.0 | 53.0 / 51.5 / 51.7 / 53.2 |

Table 8: Similarity-based Selection on TextOCR (Accuracy %).

| Model | 1-shot | 2-shot | 4-shot | 8-shot |
|---|---|---|---|---|
| LLaVA-OneVision -7B | 40.0 / 40.0 / 40.0 / 35.7 | 42.0 / 42.0 / 42.2 / 42.2 | 43.0 / 44.0 / 43.5 / 42.2 | 45.0 / 46.0 / 44.8 / 44.7 |
| Phi3-Vision | 32.5 / 32.5 / 32.5 / 32.2 | 40.0 / 39.5 / 39.0 / 38.2 | 42.5 / 42.0 / 42.7 / 39.7 | 45.0 / 43.0 / 44.7 / 41.5 |
| Mantis-Idefics2 | 12.5 / 12.5 / 12.5 / 13.8 | 24.5 / 23.5 / 24.7 / 25.2 | 27.5 / 28.0 / 29.7 / 27.8 | 29.0 / 28.0 / 27.8 / 27.7 |
| InternLM-X2D5 | 38.0 / 38.0 / 38.0 / 36.5 | 38.0 / 42.0 / 39.8 / 37.0 | 42.0 / 42.5 / 42.5 / 42.5 | 27.5 / 26.5 / 27.5 / 27.0 |

Table 9: Similarity-based Selection on CoBSAT (Accuracy %).

| Model | 1-shot | 2-shot | 4-shot | 8-shot |
|---|---|---|---|---|
| Seed-LLama-8B | 24.0 / 24.0 / 23.7 / 15.8 | 35.0 / 34.0 / 33.5 / 21.8 | 36.0 / 35.5 / 37.2 / 27.8 | 34.5 / 34.0 / 33.7 / 33.7 |
| Seed-LLama-14B | 27.0 / 27.0 / 27.0 / 23.0 | 37.0 / 34.0 / 36.5 / 33.3 | 43.5 / 41.5 / 42.2 / 40.8 | 42.0 / 43.0 / 47.8 / 43.8 |
| Emu2-Gen | 27.5 / 27.5 / 27.5 / 23.0 | 39.0 / 34.0 / 36.0 / 28.7 | 28.5 / 38.5 / 29.5 / 27.3 | 23.0 / 38.5 / 21.5 / 20.8 |

## B.11 IMPACT OF DIFFERENT MODEL ARCHITECTURES

We analyse the impact of different model architectures in Figure 17. More specifically we split the models into three categories: cross-attention based, LLaVA-like and GPT4V for which the architecture is unknown. To give more context, we include time in our analysis. The results indicate that the best accuracies and ICL efficiencies are obtained by LLaVA-like models (LLaVA-OneVision-72B, InternLM-X2d5, Phi3-Vision). However, more broadly there is no clear trend between cross-attention and LLaVA-like models, and both categories can obtain comparable performance. The first models were typically cross-attention based, but researchers have continued developing them also more recently.

## C QUALITATIVE ANALYSIS

We also include a qualitative analysis on selected cases, where we analyse the impact of using more support examples on the quality of the output. We analyse text-to-image tasks in Fig. 18, using Emu2-Gen model. For text-to-image MiniImageNet the model should learn from the support examples that the artificial names *slation* and *shously* correspond to a lion and a school bus, respectively. Emu2-Gen is able to do it to a certain extent, but may get confused by additional support examples as more support examples are not necessarily helpful. In CoBSAT, the support set induces that the animal should have glacier and desert background. With no support examples the model only displays the animal, but with more support examples it learns that it should use glacier background. In the second example, the model is able to capture that it should use desert background, but is less successful in showing the required animal – zebra. The quality of the generated images is not necessarily better with more support examples.

Table 10: Zero-shot accuracy, peak accuracy and ICL efficiency scores (%, ↑) of different models for the three arrangements of similarity-based selection as well as the original random selection.

| | Z.s. | Pk. | Eff. |
|---|---|---|---|
| **CLEVR Count Induction** | | | |
| LLaVA-OneVision-7B | 5.5 | **48.0** / **48.0** / **48.0** / 38.2 | 29.6 / 29.1 / **29.9** / 24.8 |
| Phi3-Vision | 0.0 | **42.0** / **42.0** / **42.0** / 34.5 | **31.3** / 30.6 / 31.0 / 20.4 |
| Mantis-Idefics2 | 31.5 | 36.5 / 36.0 / 36.0 / **40.2** | -1.9 / -1.9 / **-1.2** / -3.9 |
| InternLM-X2d5 | 6.0 | 62.0 / 62.0 / 62.0 / **63.2** | **48.6** / 47.0 / 47.5 / 47.5 |
| **TextOCR** | | | |
| LLaVA-OneVision-7B | 39.0 | 45.0 / **46.0** / 44.8 / 44.7 | 3.7 / **4.3** / 3.9 / 2.8 |
| Phi3-Vision | 8.0 | **45.0** / 43.0 / 44.7 / 41.5 | **31.2** / 30.5 / 31.1 / 29.0 |
| Mantis-Idefics2 | 21.0 | 29.0 / 28.0 / **29.7** / 27.8 | 4.0 / 3.8 / **4.6** / 4.1 |
| InternLM-X2d5 | 32.0 | 42.0 / **42.5** / **42.5** / **42.5** | 4.5 / **5.2** / 5.0 / 4.2 |
| **CoBSAT** | | | |
| SEED-LLaMA-8B | 0.5 | 36.0 / 35.5 / **37.2** / 33.7 | **31.2** / 30.7 / 31.1 / 24.4 |
| SEED-LLaMA-14B | 5.5 | 43.5 / 43.0 / **47.8** / 43.8 | 32.0 / 30.9 / **32.8** / 30.2 |
| Emu2-Gen | 8.7 | **39.0** / 38.5 / 36.0 / 28.7 | 19.0 / **25.7** / 18.5 / 15.5 |

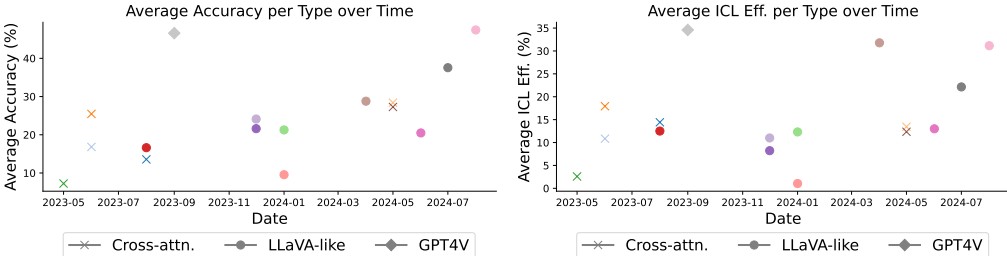

Figure 17: Average accuracy and ICL eff. per model architecture, with information on when the model was released. The best models are LLaVA-like, but otherwise the performance is comparable.

For qualitative analysis of image-to-text tasks, we discuss some of the common mistakes that the models make for each task.

**Open-Ended MiniImageNet** It is relatively common for the models to predict the real-world class, even if it is asked to use the artificial names from the support set. With more support examples such mistakes are less likely to occur as the model learns to use the artificial names.

**CLEVR** In many cases the model rephrases the question, while in others it talks that e.g. the described object is present. Such behaviour is more common with fewer or no support examples. With more support examples the model learns to predict a count but gets incorrect answer. It can be because some objects are more difficult to recognize, e.g. if one partially covers another.

**Operator Induction** A very common mistake is to use a different operator than what would be induced from the support examples. For example, the model may guess it should add two numbers instead of multiplying them and vice versa.

**Interleaved Operator Induction** The model sometimes predicts the first displayed number or a direct combination of them, e.g. if the two numbers are 1 and 2, it returns 12. It is also relatively common to use an incorrect operator between the numbers.

**TextOCR** In many cases the model returns more words than are highlighted in the red box, but includes the highlighted word as one of them. It is also common that the model misses a letter in the text or returns a word that is similar but different from the correct answer. In some cases though the answer may be very different from what is highlighted, possibly returning a different word in the image.

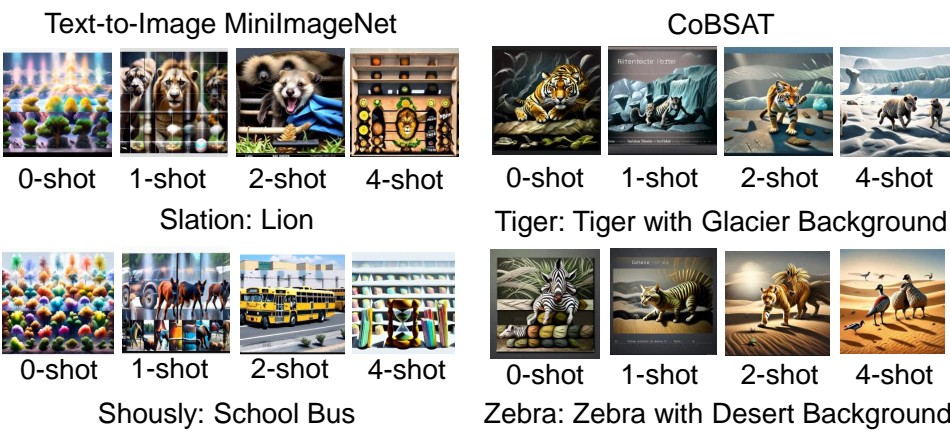

Figure 18: Qualitative analysis: Images generated by Emu2-Gen show the ability to learn the concept induced by the support examples.

# D    COMPLETE RESULTS

In this section, we show the raw results of the figures in the main text in Section D.1 to D.3, and results of supplementary materials in Section D.4.

## D.1    INITIAL ANALYSIS OF VQA AND IMAGE CAPTIONING

Table 11: Results of MathVista using string match.

| Model | 0-Shot | 1-Shot | 2-Shot | 4-Shot | 8-Shot |
|---|---|---|---|---|---|
| OpenFlamingo-9B | 9.00 | 17.00 | 20.00 | 21.80 | 22.00 |
| Otter | 14.00 | 16.50 | 18.00 | 18.50 | 19.50 |
| IDEFICS-9B | 0.50 | 5.50 | 16.60 | 20.50 | 21.00 |
| InternLM-XComposer2 | 22.50 | 21.00 | 22.50 | 24.00 | 24.00 |
| Qwen-VL-Chat | 20.00 | 21.50 | 22.00 | 22.00 | 22.00 |
| LLaVA-Next-Vicuna-7B | 23.00 | 22.00 | 15.00 | 10.00 | 9.50 |
| Emu2-Chat | 18.50 | 22.50 | 23.50 | 23.50 | 20.00 |

Table 12: Results of MathVista using LLM for answer extraction.

| Model | 0-Shot | 1-Shot | 2-Shot | 4-Shot | 8-Shot |
|---|---|---|---|---|---|
| OpenFlamingo-9B | 21.00 | 22.00 | 22.00 | 22.00 | 23.50 |
| Otter | 20.00 | 20.00 | 17.50 | 19.50 | 19.50 |
| IDEFICS-9B | 22.50 | 22.00 | 22.00 | 23.50 | 23.50 |
| InternLM-XComposer2 | 28.00 | 28.50 | 28.50 | 29.50 | 26.00 |
| Qwen-VL-Chat | 19.00 | 21.00 | 19.50 | 18.50 | 19.00 |
| LLaVA-Next-Vicuna-7B | 23.00 | 25.00 | 18.00 | 15.00 | 10.50 |
| Emu2-Chat | 24.50 | 22.50 | 23.50 | 23.50 | 22.00 |

Table 13: Results of VizWiz using exact match.

| Model | 0-Shot | 1-Shot | 2-Shot | 4-Shot | 8-Shot |
|---|---|---|---|---|---|
| OpenFlamingo-9B | 24.57 | 29.31 | 30.62 | 32.14 | 35.11 |
| Otter | 20.13 | 24.21 | 25.78 | 26.33 | 25.71 |
| IDEFICS-9B | 33.20 | 31.67 | 45.33 | 42.80 | 41.87 |
| InternLM-X2 | 71.93 | 66.33 | 66.73 | 66.27 | 71.73 |
| Qwen-VL-Chat | 32.40 | 34.80 | 35.20 | 40.80 | 40.90 |
| LLaVA-Next-Vicuna-7B | 54.12 | 28.13 | 10.20 | 6.60 | 0.40 |
| Emu2-Chat | 31.06 | 34.20 | 36.13 | 40.12 | 42.66 |

Table 14: Results of VizWiz using LLM as the judge.

| Model | 0-Shot | 1-Shot | 2-Shot | 4-Shot | 8-Shot |
|---|---|---|---|---|---|
| OpenFlamingo-9B | 31.54 | 35.85 | 36.10 | 36.4 | 37.28 |
| Otter | 23.40 | 24.12 | 25.88 | 26.19 | 24.93 |
| IDEFICS-9B | 35.10 | 36.90 | 40.66 | 42.30 | 41.35 |
| InternLM-XComposer2 | 72.90 | 68.20 | 69.10 | 66.30 | 68.42 |
| Qwen-VL-Chat | 37.40 | 37.53 | 39.22 | 40.38 | 41.20 |
| LLaVA-Next-Vicuna-7B | 55.26 | 26.08 | 11.38 | 8.72 | 2.50 |
| Emu2-Chat | 35.68 | 35.83 | 37.67 | 40.51 | 41.99 |

Table 15: Results of COCO captions (CIDEr).

| Model | 0-Shot | 1-Shot | 2-Shot | 4-Shot | 8-Shot |
|---|---|---|---|---|---|
| OpenFlamingo-9B | 78.2 | 85.6 | 88.1 | 89.0 | 93.3 |
| Otter | 86.2 | 91.0 | 97.0 | 98.6 | 100.5 |
| IDEFICS-9B | 71.6 | 84.20 | 90.8 | 93.1 | 98.5 |
| InternLM-XComposer2 | 125.74 | 125.26 | 138.82 | 135.22 | 129.8 |
| Qwen-VL-Chat | 121.10 | 132.6 | 135.3 | 135.4 | 136.9 |
| LLaVA-Next-Vicuna-7B | 131.24 | 81.75 | 40.49 | 34.47 | 26.26 |
| Emu2-Chat | 126.3 | 127.0 | 132.06 | 131.10 | 133.5 |

Table 16: Results of COCO captions. Scores are rated by LLaVA-Next from 1-10 (higher the better).

| Model | 0-Shot | 1-Shot | 2-Shot | 4-Shot | 8-Shot |
|---|---|---|---|---|---|
| OpenFlamingo-9B | 2.71 | 2.42 | 2.75 | 2.88 | 2.90 |
| Otter | 5.60 | 5.78 | 5.68 | 5.72 | 5.56 |
| IDEFICS-9B | 6.15 | 6.22 | 6.15 | 6.21 | 6.22 |
| InternLM-XComposer2 | 6.67 | 4.17 | 4.67 | 5.11 | 4.58 |
| Qwen-VL-Chat | 6.17 | 6.17 | 6.12 | 6.11 | 6.12 |
| LLaVA-Next-Vicuna-7B | 6.34 | 3.54 | 3.75 | 3.43 | 3.57 |
| Emu2-Chat | 6.15 | 5.89 | 6.05 | 6.11 | 6.15 |

Table 17: Comparisons of VLLMs and LLMs for text ICL on AGNews dataset (Accuracy %).

| Model | 0-Shot | 1-Shot | 2-Shot | 4-Shot | 8-Shot |
|---|---|---|---|---|---|
| LLaVA-Next-Vicuna-7B | 74.66 | 81.16 | 81.61 | 84.05 | 85.38 |
| Vicuna-7B | 65.83 | 79.22 | 80.20 | 82.24 | 82.41 |
| Qwen-VL-Chat | 75.49 | 72.74 | 73.78 | 78.62 | 80.91 |
| QwenLM-7B | 54.80 | 59.51 | 67.53 | 72.80 | 74.64 |
| InternLM-XComposer2 | 83.99 | 82.87 | 82.80 | 83.28 | 83.97 |
| InternLM2-Chat-7B | 81.89 | 84.11 | 84.25 | 84.32 | 84.33 |

Table 18: Comparisons of VLLMs and LLMs for text ICL on MIT Movies dataset (Accuracy %).

| Model | 0-Shot | 1-Shot | 2-Shot | 4-Shot | 8-Shot |
|---|---|---|---|---|---|
| LLaVA-Next-Vicuna-7B | 47.47 | 29.88 | 59.04 | 85.06 | 89.16 |
| Vicuna-7B | 50.36 | 63.61 | 77.83 | 86.99 | 89.88 |
| Qwen-VL-Chat | 50.36 | 26.99 | 46.02 | 74.70 | 85.54 |
| QwenLM-7B | 66.27 | 46.99 | 64.10 | 85.30 | 92.53 |
| InternLM-XComposer2 | 69.64 | 69.64 | 78.31 | 88.19 | 90.12 |
| InternLM2-Chat-7B | 61.45 | 63.61 | 73.98 | 87.71 | 94.46 |

Table 19: Comparisons of VLLMs and LLMs for text ICL on TREC dataset (Accuracy %).

| Model | 0-Shot | 1-Shot | 2-Shot | 4-Shot | 8-Shot |
|---|---|---|---|---|---|
| LLaVA-Next-Vicuna-7B | 49.80 | 51.80 | 55.60 | 65.40 | 71.00 |
| Vicuna-7B | 46.20 | 55.00 | 63.60 | 64.20 | 70.40 |
| Qwen-VL-Chat | 43.60 | 51.00 | 55.40 | 61.20 | 72.60 |
| QwenLM-7B | 40.80 | 54.00 | 61.00 | 63.00 | 73.90 |
| InternLM-XComposer2 | 59.00 | 71.20 | 75.00 | 81.80 | 82.80 |
| InternLM2-Chat-7B | 62.00 | 80.20 | 85.00 | 87.00 | 87.20 |

## D.2    MAIN RESULTS

Table 20: Results of different models on Fast Open-Ended Mini-ImageNet (Accuracy %).

| Model | 0-Shot | 1-Shot | 2-Shot | 4-Shot | 5-Shot |
|---|---|---|---|---|---|
| OpenFlamingo-9B | 0.00 ± 0.00 | 39.50 ± 1.22 | 58.17 ± 3.57 | 51.17 ± 0.85 | 54.50 ± 5.66 |
| IDEFICS-9B | 0.00 ± 0.00 | 22.00 ± 0.41 | 52.00 ± 2.94 | 53.83 ± 0.94 | 59.17 ± 6.20 |
| IDEFICS-80B | 0.00 ± 0.00 | 28.50 ± 0.27 | 49.50 ± 1.28 | 52.47 ± 3.25 | 62.50 ± 2.00 |
| IDEFICS2-8B | 0.00 ± 0.00 | 0.00 ± 0.00 | 43.50 ± 1.47 | 61.50 ± 1.41 | 58.00 ± 0.71 |
| Otter | 0.00 ± 0.00 | 10.00 ± 0.71 | 25.00 ± 1.22 | 28.50 ± 2.86 | 25.67 ± 2.25 |
| InternLM-X2 | 0.00 ± 0.00 | 14.83 ± 1.03 | 38.00 ± 1.78 | 49.00 ± 1.78 | 50.33 ± 3.86 |
| Qwen-VL-Chat | 0.00 ± 0.00 | 0.50 ± 0.41 | 47.33 ± 2.49 | 58.00 ± 2.83 | 55.17 ± 2.25 |
| LLaVA-Next-7B | 0.00 ± 0.00 | 22.17 ± 4.03 | 33.67 ± 2.25 | 37.24 ± 1.02 | 36.95 ± 0.24 |
| Emu2-Chat | 0.00 ± 0.00 | 8.00 ± 1.87 | 29.33 ± 1.84 | 28.18 ± 4.26 | 27.54± 5.12 |
| VILA-2.7B | 0.00 ± 0.00 | 5.67 ± 0.24 | 50.33 ± 1.18 | 45.00 ± 1.08 | 42.00 ± 1.78 |
| VILA-7B | 0.00 ± 0.00 | 28.83 ± 1.43 | 38.17 ± 2.72 | 37.67 ± 3.68 | 37.33 ± 1.18 |
| Mantis-Idefics2 | 0.00 ± 0.00 | 2.50 ± 1.08 | 72.83 ± 2.39 | 84.33 ± 2.09 | 76.00 ± 0.41 |
| Phi3-Vision | 0.00 ± 0.00 | 12.67 ± 0.85 | 50.00 ± 1.08 | 39.00 ± 2.48 | 40.17 ± 1.25 |
| LongVA-7B | 0.00 ± 0.00 | 22.33 ± 0.24 | 52.67 ± 3.09 | 48.83 ± 1.43 | 49.50 ± 1.08 |
| LLaVA-OneVision-72B | 0.00 ± 0.00 | 12.67 ± 0.94 | 88.00 ± 0.82 | 98.67 ± 0.47 | 98.33 ± 0.62 |
| InternLM-X2d5 | 0.00 ± 0.00 | 12.00 ± 0.00 | 82.50 ± 1.22 | 88.17 ± 1.70 | 91.00 ± 1.47 |
| GPT4V | 0.00 | 14.00 | 48.00 | 56.00 | 78.00 |

Table 21: Results of different models on Real-name Mini-ImageNet (Accuracy %).

| Model | 0-Shot | 1-Shot | 2-Shot | 4-Shot | 5-Shot |
|---|---|---|---|---|---|
| OpenFlamingo-9B | 0.00 ± 0.00 | 26.00 ± 2.86 | 53.33 ± 3.27 | 52.83 ± 0.94 | 49.50 ± 1.22 |
| IDEFICS-9B | 26.50 ± 0.00 | 41.83 ± 2.25 | 74.50 ± 2.27 | 89.00 ± 0.41 | 91.17 ± 1.89 |
| IDEFICS-80B | 30.50 ± 0.00 | 41.83 ± 1.18 | 82.00 ± 2.68 | 94.67 ± 0.62 | 91.33 ± 1.43 |
| Otter | 13.00 ± 0.00 | 51.00 ± 2.16 | 57.33 ± 3.09 | 56.50 ± 1.08 | 61.00 ± 1.87 |
| InternLM-X2 | 20.00 ± 0.00 | 31.50 ± 1.63 | 67.00 ± 1.47 | 66.83 ± 0.24 | 66.67 ± 1.89 |
| Qwen-VL-Chat | 32.17 ± 0.24 | 40.67 ± 1.03 | 58.00 ± 0.71 | 84.67 ± 1.03 | 88.33 ± 2.05 |
| LLaVA-Next-7B | 20.50 ± 0.00 | 64.50 ± 0.82 | 52.83 ± 1.25 | 7.83 ± 1.65 | 7.83 ± 1.55 |
| Emu2-Chat | 29.89 ± 0.00 | 50.17 ± 1.44 | 51.43 ± 1.52 | 59.38 ± 2.03 | 57.25 ± 3.06 |
| Mantis-Idefics2 | 18.00 ± 0.00 | 27.00 ± 1.08 | 49.33 ± 1.25 | 16.67 ± 1.70 | 17.83 ± 2.05 |
| IDEFICS2-8B | 25.50 ± 0.00 | 24.17 ± 1.25 | 57.83 ± 1.31 | 79.00 ± 2.55 | 79.83 ± 1.25 |
| InternLM-X2d5 | 18.50 ± 0.00 | 41.67 ± 1.31 | 69.83 ± 0.24 | 83.50 ± 0.82 | 84.50 ± 0.71 |
| LLaVA-OneVision-7B | 24.50 ± 0.00 | 54.17 ± 0.85 | 68.17 ± 1.31 | 67.00 ± 1.08 | 67.17 ± 3.42 |
| GPT4V | 48.00 | 56.00 | 78.00 | 90.00 | 86.00 |

## D.3    ADDITIONAL ANALYSIS

Table 22: Results of different models on Operator Induction dataset (Accuracy %).

| Model | 0-Shot | 1-Shot | 2-Shot | 4-Shot | 8-Shot |
|---|---|---|---|---|---|
| OpenFlamingo-9B | $5.00 \pm 0.00$ | $2.22 \pm 3.14$ | $1.67 \pm 1.36$ | $2.78 \pm 0.79$ | $7.78 \pm 2.08$ |
| IDEFICS-9B | $11.67 \pm 0.00$ | $14.44 \pm 0.79$ | $10.56 \pm 2.08$ | $7.78 \pm 2.08$ | $11.11 \pm 1.57$ |
| IDEFICS-80B | $13.33 \pm 0.00$ | $15.00 \pm 2.72$ | $14.67 \pm 2.36$ | $21.67 \pm 1.36$ | $16.11 \pm 2.08$ |
| IDEFICS2-8B | $36.67 \pm 0.00$ | $41.67 \pm 4.08$ | $39.45 \pm 6.14$ | $38.89 \pm 3.14$ | $47.22 \pm 4.78$ |
| Otter | $21.67 \pm 0.00$ | $11.67 \pm 2.36$ | $13.33 \pm 1.36$ | $12.22 \pm 1.57$ | $7.22 \pm 1.57$ |
| InternLM-X2 | $26.11 \pm 3.14$ | $40.00 \pm 10.80$ | $40.00 \pm 4.91$ | $39.44 \pm 7.49$ | $28.89 \pm 19.83$ |
| Qwen-VL-Chat | $15.00 \pm 0.00$ | $10.00 \pm 1.36$ | $17.22 \pm 3.14$ | $18.89 \pm 1.57$ | $25.00 \pm 2.72$ |
| LLaVA-Next-7B | $10.56 \pm 1.57$ | $6.11 \pm 1.57$ | $5.56 \pm 2.08$ | $3.33 \pm 2.72$ | $0.00 \pm 0.00$ |
| Emu2-Chat | $28.56 \pm 1.57$ | $21.67 \pm 5.93$ | $21.11 \pm 1.57$ | $21.67 \pm 0.00$ | $21.11 \pm 5.50$ |
| VILA-2.7B | $16.67 \pm 0.00$ | $12.78 \pm 2.08$ | $11.11 \pm 3.14$ | $11.67 \pm 1.36$ | $11.67 \pm 2.72$ |
| VILA-7B | $28.33 \pm 0.00$ | $11.11 \pm 4.37$ | $6.67 \pm 3.60$ | $7.78 \pm 0.78$ | $8.33 \pm 2.72$ |
| Phi3-Vision | $0.00 \pm 0.00$ | $42.22 \pm 1.57$ | $45.56 \pm 0.79$ | $51.67 \pm 1.36$ | $54.44 \pm 2.83$ |
| LongVA-7B | $18.33 \pm 0.00$ | $8.33 \pm 3.60$ | $6.11 \pm 2.83$ | $5.56 \pm 2.83$ | $3.89 \pm 2.83$ |
| Mantis-Idefics2 | $16.67 \pm 0.00$ | $12.78 \pm 2.08$ | $8.89 \pm 2.08$ | $14.44 \pm 4.16$ | $16.11 \pm 0.79$ |
| LLaVA-OneVision-72B | $33.33 \pm 0.00$ | $75.56 \pm 3.42$ | $60.56 \pm 6.71$ | $68.89 \pm 1.57$ | $57.78 \pm 2.08$ |
| InternLM-X2d5 | $36.67 \pm 0.00$ | $40.00 \pm 3.60$ | $41.67 \pm 2.72$ | $35.00 \pm 2.36$ | $35.00 \pm 4.08$ |
| GPT4V | 24.00 | 66.00 | 84.00 | 92.00 | 92.00 |

Table 23: Results of different models on TextOCR dataset (Accuracy %).

| Model | 0-Shot | 1-Shot | 2-Shot | 4-Shot | 8-Shot |
|---|---|---|---|---|---|
| OpenFlamingo-9B | $0.00 \pm 0.00$ | $0.00 \pm 0.00$ | $0.00 \pm 0.00$ | $0.00 \pm 0.00$ | $0.00 \pm 0.00$ |
| IDEFICS-9B | $16.50 \pm 0.00$ | $22.50 \pm 1.08$ | $19.83 \pm 0.62$ | $22.83 \pm 1.31$ | $28.00 \pm 1.63$ |
| IDEFICS-80B | $20.00 \pm 0.00$ | $25.50 \pm 2.18$ | $25.38 \pm 2.78$ | $29.50 \pm 2.89$ | $23.50 \pm 3.47$ |
| IDEFICS2-8B | $23.00 \pm 0.00$ | $17.00 \pm 1.41$ | $25.50 \pm 0.82$ | $28.33 \pm 1.25$ | $30.33 \pm 0.85$ |
| Otter | $0.00 \pm 0.00$ | $0.00 \pm 0.00$ | $0.17 \pm 0.24$ | $0.83 \pm 0.47$ | $0.67 \pm 0.24$ |
| InternLM-X2 | $8.67 \pm 4.01$ | $3.83 \pm 0.62$ | $10.50 \pm 0.71$ | $16.00 \pm 2.48$ | $11.83 \pm 2.95$ |
| Qwen-VL-Chat | $4.83 \pm 6.84$ | $17.17 \pm 1.43$ | $21.50 \pm 1.08$ | $22.33 \pm 1.31$ | $24.17 \pm 0.24$ |
| LLaVA-Next-7B | $24.67 \pm 2.25$ | $0.83 \pm 0.24$ | $0.33 \pm 0.24$ | $0.00 \pm 0.00$ | $0.00 \pm 0.00$ |
| Emu2-Chat | $25.83 \pm 0.24$ | $23.50 \pm 1.47$ | $31.50 \pm 1.87$ | $36.50 \pm 1.87$ | $29.50 \pm 1.78$ |
| VILA-2.7B | $0.50 \pm 0.00$ | $6.33 \pm 1.18$ | $14.83 \pm 2.39$ | $16.67 \pm 0.62$ | $18.50 \pm 1.22$ |
| VILA-7B | $28.00 \pm 0.00$ | $6.17 \pm 1.03$ | $22.17 \pm 0.24$ | $26.83 \pm 0.47$ | $30.17 \pm 1.03$ |
| Phi3-Vision | $8.00 \pm 0.00$ | $32.17 \pm 2.25$ | $38.17 \pm 0.85$ | $39.67 \pm 0.24$ | $41.50 \pm 1.78$ |
| LongVA-7B | $26.00 \pm 0.00$ | $3.50 \pm 0.82$ | $11.17 \pm 0.85$ | $17.17 \pm 1.65$ | $17.50 \pm 2.12$ |
| Mantis-Idefics2 | $21.00 \pm 0.00$ | $13.83 \pm 1.18$ | $25.17 \pm 0.62$ | $27.83 \pm 1.25$ | $27.67 \pm 0.47$ |
| LLaVA-OneVision-72B | $48.50 \pm 0.00$ | $50.67 \pm 1.70$ | $51.17 \pm 0.24$ | $50.67 \pm 2.09$ | $51.67 \pm 0.62$ |
| InternLM-X2d5 | $32.00 \pm 0.00$ | $36.50 \pm 1.41$ | $37.00 \pm 0.41$ | $42.50 \pm 0.71$ | $27.02 \pm 0.24$ |
| GPT4V | 39.29 | 32.14 | 48.00 | 50.00 | 49.00 |

Table 24: Results of different models on CLEVR dataset (Accuracy %).

| Model | 0-Shot | 1-Shot | 2-Shot | 4-Shot | 8-Shot |
|---|---|---|---|---|---|
| OpenFlamingo-9B | $0.00 \pm 0.00$ | $17.83 \pm 2.25$ | $17.00 \pm 2.27$ | $18.83 \pm 1.03$ | $16.33 \pm 1.43$ |
| IDEFICS-9B | $0.00 \pm 0.00$ | $30.33 \pm 2.25$ | $29.50 \pm 1.47$ | $27.67 \pm 2.05$ | $27.17 \pm 2.87$ |
| IDEFICS-80B | $0.00 \pm 0.00$ | $31.16 \pm 2.10$ | $30.82 \pm 1.59$ | $31.50 \pm 1.00$ | $32.43 \pm 3.62$ |
| IDEFICS2-8B | $2.00 \pm 0.00$ | $51.00 \pm 0.71$ | $48.83 \pm 4.09$ | $43.83 \pm 0.62$ | $51.50 \pm 1.87$ |
| Otter | $0.00 \pm 0.00$ | $5.42 \pm 1.06$ | $8.33 \pm 2.24$ | $8.17 \pm 1.44$ | $0.17 \pm 0.24$ |
| InternLM-X2 | $1.83 \pm 0.24$ | $26.00 \pm 1.63$ | $24.67 \pm 5.25$ | $20.00 \pm 2.94$ | $22.83 \pm 0.85$ |
| Qwen-VL-Chat | $0.00 \pm 0.00$ | $29.83 \pm 4.55$ | $25.33 \pm 3.47$ | $26.83 \pm 3.06$ | $30.17 \pm 2.95$ |
| LLaVA-Next-7B | $0.00 \pm 0.00$ | $25.17 \pm 6.64$ | $24.83 \pm 4.90$ | $17.83 \pm 4.59$ | $0.17 \pm 0.24$ |
| Emu2-Chat | $5.33 \pm 0.24$ | $11.83 \pm 2.72$ | $14.00 \pm 3.49$ | $14.83 \pm 1.89$ | $17.67 \pm 1.03$ |
| VILA-2.7B | $0.00 \pm 0.00$ | $30.67 \pm 2.09$ | $29.17 \pm 0.85$ | $29.17 \pm 1.03$ | $29.83 \pm 0.24$ |
| VILA-7B | $3.50 \pm 0.00$ | $34.00 \pm 2.86$ | $32.67 \pm 1.43$ | $31.50 \pm 2.16$ | $34.33 \pm 2.39$ |
| Phi3-Vision | $0.00 \pm 0.00$ | $34.50 \pm 2.55$ | $25.50 \pm 2.45$ | $17.67 \pm 2.09$ | $18.67 \pm 0.85$ |
| LongVA-7B | $0.00 \pm 0.00$ | $29.17 \pm 1.31$ | $22.83 \pm 2.39$ | $20.00 \pm 1.08$ | $23.83 \pm 3.47$ |
| Mantis-Idefics2 | $31.45 \pm 2.18$ | $40.17 \pm 3.12$ | $32.00 \pm 0.71$ | $24.50 \pm 1.08$ | $21.67 \pm 2.05$ |
| LLaVA-OneVision-72B | $0.50 \pm 0.00$ | $37.17 \pm 2.01$ | $32.00 \pm 0.71$ | $31.83 \pm 1.84$ | $42.33 \pm 0.85$ |
| InternLM-X2d5 | $6.00 \pm 0.00$ | $63.17 \pm 1.25$ | $58.50 \pm 2.86$ | $56.00 \pm 2.68$ | $53.17 \pm 2.49$ |
| GPT4V | 6.00 | 30.00 | 38.00 | 42.00 | 32.00 |

Table 25: Results of different models on Fast Matching MiniImageNet dataset (Accuracy %).

| Model | 0-Shot | 1-Shot | 2-Shot | 4-Shot | 5-Shot |
|---|---|---|---|---|---|
| OpenFlamingo-9B | 0.00 ± 0.00 | 18.25 ± 1.32 | 19.28 ± 5.17 | 26.33 ± 3.02 | 29.11 ± 1.30 |
| IDEFICS-9B | 0.00 ± 0.00 | 0.08 ± 0.12 | 0.00 ± 0.00 | 0.00 ± 0.00 | 0.00 ± 0.00 |
| IDEFICS-80B | 0.00 ± 0.00 | 21.58 ± 3.70 | 28.30 ± 1.94 | 23.16 ± 1.45 | 25.58 ± 2.13 |
| Otter | 0.00 ± 0.00 | 0.00 ± 0.00 | 0.00 ± 0.00 | 0.00 ± 0.00 | 0.00 ± 0.00 |
| IDEFICS2-8B | 0.00 ± 0.00 | 0.58 ± 0.12 | 8.33 ± 1.16 | 17.33 ± 0.92 | 25.25 ± 1.74 |
| Mantis-Idefics2 | 0.00 ± 0.00 | 51.08 ± 3.44 | 49.25 ± 1.59 | 47.08 ± 0.72 | 47.17 ± 0.47 |
| InternLM-X2 | 0.00 ± 0.00 | 49.92 ± 0.12 | 49.92 ± 0.12 | 0.92 ± 0.31 | 0.83 ± 0.24 |
| Qwen-VL-Chat | 0.00 ± 0.00 | 0.33 ± 0.24 | 0.00 ± 0.00 | 0.08 ± 0.12 | 0.08 ± 0.12 |
| Phi3-Vision | 0.00 ± 0.00 | 45.92 ± 0.85 | 46.08 ± 1.48 | 47.58 ± 1.23 | 50.75 ± 1.43 |
| LLaVA-Next-7B | 0.00 ± 0.00 | 0.00 ± 0.00 | 0.00 ± 0.00 | 0.00 ± 0.00 | 0.00 ± 0.00 |
| LongVA-7B | 0.00 ± 0.00 | 50.00 ± 0.00 | 50.00 ± 0.00 | 50.00 ± 0.00 | 50.00 ± 0.00 |
| VILA-7B | 0.00 ± 0.00 | 49.67 ± 0.31 | 49.75 ± 0.35 | 49.08 ± 0.77 | 49.92 ± 0.24 |
| Emu2-Chat | 0.00 ± 0.00 | 32.59 ± 0.58 | 36.80 ± 1.80 | 37.62 ± 0.93 | 40.25 ± 2.43 |
| InternLM-X2d5 | 0.00 ± 0.00 | 9.25 ± 0.00 | 46.75 ± 0.00 | 31.50 ± 0.00 | 8.00 ± 0.00 |
| LLaVA-OneVision-72B | 0.00 ± 0.00 | 28.50 ± 2.07 | 58.08 ± 0.92 | 68.83 ± 0.31 | 70.33 ± 0.66 |
| GPT4V | 0.00 | 2.00 | 48.75 | 58.25 | 58.00 |

Table 26: Results of different models on Interleaved Operator induction (Accuracy %).

| Model | 0-Shot | 1-Shot | 2-Shot | 4-Shot | 8-Shot |
|---|---|---|---|---|---|
| OpenFlamingo-9B | 0.00 ± 0.00 | 5.56 ± 1.57 | 3.89 ± 2.83 | 2.78 ± 0.79 | 8.89 ± 3.42 |
| IDEFICS-9B | 15.00 ± 0.00 | 5.56 ± 2.08 | 6.11 ± 0.79 | 6.11 ± 1.57 | 5.00 ± 2.36 |
| IDEFICS-80B | 25.00 ± 0.00 | 36.67 ± 1.21 | 31.67 ± 2.46 | 28.33 ± 3.13 | 20.00 ± 2.77 |
| IDEFICS2-8B | 38.33 ± 0.00 | 16.11 ± 0.79 | 17.78 ± 3.14 | 20.00 ± 1.36 | 20.00 ± 4.71 |
| Otter | 8.33 ± 0.00 | 7.78 ± 1.57 | 9.44 ± 3.14 | 7.22 ± 2.83 | 5.56 ± 2.83 |
| InternLM-X2 | 28.33 ± 0.00 | 10.56 ± 2.83 | 9.44 ± 2.83 | 11.11 ± 3.93 | 4.44 ± 2.83 |
| Qwen-VL-Chat | 16.67 ± 0.00 | 9.44 ± 0.79 | 8.33 ± 1.36 | 8.89 ± 2.83 | 5.56 ± 0.79 |
| LLaVA-Next-7B | 13.89 ± 1.57 | 7.22 ± 2.83 | 6.11 ± 3.14 | 5.00 ± 0.00 | 5.00 ± 2.72 |
| Emu2-Chat | 26.67 ± 0.00 | 18.33 ± 2.72 | 20.56 ± 3.42 | 10.00 ± 0.00 | 7.62 ± 1.83 |
| VILA-2.7B | 13.33 ± 0.00 | 10.56 ± 0.79 | 10.56 ± 0.79 | 11.11 ± 2.08 | 11.67 ± 1.36 |
| VILA-7B | 28.33 ± 0.00 | 6.11 ± 2.08 | 11.67 ± 3.60 | 12.22 ± 2.83 | 13.33 ± 2.36 |
| Phi3-Vision | 0.00 ± 0.00 | 16.67 ± 2.72 | 21.67 ± 1.36 | 15.56 ± 1.57 | 27.78 ± 3.14 |
| LongVA-7B | 10.00 ± 0.00 | 5.00 ± 1.36 | 4.44 ± 0.79 | 4.45 ± 3.14 | 3.89 ± 4.37 |
| Mantis-Idefics2 | 30.00 ± 0.00 | 5.56 ± 3.42 | 4.44 ± 0.79 | 6.67 ± 1.36 | 7.22 ± 0.78 |
| LLaVA-OneVision-72B | 38.33 ± 0.00 | 42.78 ± 2.08 | 41.11 ± 2.08 | 36.67 ± 2.72 | 47.78 ± 10.03 |
| InternLM-X2d5 | 38.33 ± 0.00 | 32.22 ± 3.43 | 31.67 ± 3.60 | 33.89 ± 2.08 | 21.11 ± 2.08 |
| GPT4V | 36.00 | 58.00 | 72.00 | 74.00 | 70.00 |

Table 27: Results of different models on CobSAT: Total accuracies (Accuracy %).

| Model | 0-Shot | 1-Shot | 2-Shot | 4-Shot | 8-Shot |
|---|---|---|---|---|---|
| GILL | 2.67 ± 0.24 | 12.33 ± 1.31 | 9.33 ± 0.24 | 11.50 ± 1.47 | 8.00 ± 1.63 |
| SEED-LLaMA-8B | 0.50 ± 0.41 | 15.83 ± 1.65 | 21.83 ± 1.65 | 27.83 ± 2.36 | 33.67 ± 2.32 |
| SEED-LLaMA-14B | 5.50 ± 0.71 | 26.83 ± 1.65 | 33.33 ± 3.32 | 40.83 ± 1.65 | 43.83 ± 2.87 |
| Emu1-Gen | 0.33 ± 0.47 | 4.83 ± 0.47 | 6.17 ± 2.72 | 8.67 ± 1.18 | 9.67 ± 0.24 |
| Emu2-Gen | 8.67 ± 0.62 | 23.00 ± 3.24 | 28.67 ± 2.01 | 27.33 ± 2.72 | 20.83 ± 0.85 |

Table 28: Results of different models on CobSAT: Latent accuracies (Accuracy %).

| Model | 0-Shot | 1-Shot | 2-Shot | 4-Shot | 8-Shot |
|---|---|---|---|---|---|
| GILL | 7.67 ± 0.24 | 47.67 ± 1.43 | 53.17 ± 1.65 | 67.33 ± 1.03 | 72.33 ± 0.85 |
| SEED-LLaMA-8B | 8.00 ± 0.41 | 38.50 ± 1.47 | 44.33 ± 0.62 | 49.50 ± 0.82 | 56.00 ± 1.41 |
| SEED-LLaMA-14B | 15.17 ± 0.62 | 41.50 ± 1.41 | 52.17 ± 2.09 | 53.67 ± 1.93 | 57.17 ± 1.84 |
| Emu1-Gen | 8.00 ± 0.41 | 55.50 ± 2.55 | 71.00 ± 0.71 | 77.00 ± 0.82 | 82.00 ± 0.00 |
| Emu2-Gen | 18.00 ± 1.63 | 43.83 ± 4.33 | 72.33 ± 1.25 | 81.50 ± 0.41 | 78.33 ± 1.25 |

Table 29: Results of different models on CobSAT: Non-latent accuracies (Accuracy %).

| Model | 0-Shot | 1-Shot | 2-Shot | 4-Shot | 8-Shot |
|---|---|---|---|---|---|
| GILL | $19.33 \pm 0.24$ | $21.33 \pm 1.65$ | $16.17 \pm 1.65$ | $19.83 \pm 2.01$ | $15.83 \pm 2.78$ |
| SEED-LLaMA-8B | $12.33 \pm 1.03$ | $47.33 \pm 4.03$ | $52.00 \pm 1.87$ | $58.83 \pm 1.93$ | $63.33 \pm 2.01$ |
| SEED-LLaMA-14B | $82.67 \pm 0.24$ | $76.33 \pm 0.94$ | $75.83 \pm 1.70$ | $78.33 \pm 0.85$ | $80.83 \pm 0.85$ |
| Emu1-Gen | $26.50 \pm 0.41$ | $11.17 \pm 0.47$ | $13.33 \pm 2.25$ | $16.00 \pm 0.71$ | $17.00 \pm 0.71$ |
| Emu2-Gen | $62.00 \pm 0.41$ | $49.17 \pm 4.29$ | $42.33 \pm 2.62$ | $35.67 \pm 2.05$ | $29.33 \pm 1.43$ |

Table 30: Results of different models on Text-to-Image Fast Mini-ImageNet (Accuracy %)

| Model | 0-Shot | 1-Shot | 2-Shot | 4-Shot | 5-Shot |
|---|---|---|---|---|---|
| GILL | $0.00 \pm 0.00$ | $16.00 \pm 2.27$ | $15.17 \pm 2.72$ | $14.83 \pm 0.24$ | $14.33 \pm 2.25$ |
| SEED-LLaMA-8B | $0.00 \pm 0.00$ | $15.00 \pm 3.27$ | $12.67 \pm 1.18$ | $16.00 \pm 2.12$ | $16.50 \pm 1.87$ |
| SEED-LLaMA-14B | $0.75 \pm 0.25$ | $17.25 \pm 2.75$ | $16.75 \pm 1.75$ | $21.25 \pm 1.75$ | $21.00 \pm 3.00$ |
| Emu1-Gen | $0.50 \pm 0.41$ | $31.50 \pm 1.87$ | $22.83 \pm 2.72$ | $25.00 \pm 0.71$ | $23.17 \pm 1.03$ |
| Emu2-Gen | $0.00 \pm 0.00$ | $24.33 \pm 3.30$ | $30.67 \pm 1.31$ | $37.00 \pm 1.22$ | $34.50 \pm 0.00$ |
| Anole-7B | $0.50 \pm 0.41$ | $11.00 \pm 2.86$ | $7.00 \pm 0.71$ | $0.17 \pm 0.24$ | $0.17 \pm 0.24$ |

Table 31: Results of different models on Fast Counting (Accuracy %).

| Model | 0-Shot | 1-Shot | 2-Shot | 4-Shot | 5-Shot |
|---|---|---|---|---|---|
| GILL | $34.17 \pm 2.36$ | $8.33 \pm 4.25$ | $3.33 \pm 2.36$ | $3.33 \pm 1.18$ | $7.50 \pm 2.04$ |
| SEED-LLaMA-8B | $34.17 \pm 2.36$ | $50.00 \pm 6.12$ | $47.50 \pm 4.08$ | $45.00 \pm 2.04$ | $56.67 \pm 6.24$ |
| SEED-LLaMA-14B | $41.26 \pm 1.25$ | $42.72 \pm 5.14$ | $37.50 \pm 2.36$ | $51.57 \pm 3.25$ | $37.55 \pm 5.54$ |
| Emu1-Gen | $33.19 \pm 4.27$ | $39.55 \pm 4.51$ | $37.56 \pm 3.17$ | $47.81 \pm 3.94$ | $45.52 \pm 3.57$ |
| Emu2-Gen | $44.17 \pm 4.32$ | $50.80 \pm 2.12$ | $47.13 \pm 3.98$ | $49.25 \pm 4.04$ | $59.25 \pm 5.23$ |

Table 32: Results of different models on Fast Attribute Matching (Accuracy %).

| Model | 0-Shot | 1-Shot | 2-Shot | 4-Shot | 5-Shot |
|---|---|---|---|---|---|
| GILL | $20.50 \pm 1.08$ | $6.50 \pm 0.41$ | $4.67 \pm 1.43$ | $6.17 \pm 1.31$ | $7.00 \pm 1.47$ |
| SEED-LLaMA-8B | $21.00 \pm 0.41$ | $31.17 \pm 1.43$ | $34.33 \pm 0.62$ | $32.17 \pm 1.03$ | $32.17 \pm 1.03$ |
| SEED-LLaMA-14B | $22.36 \pm 0.28$ | $26.35 \pm 1.93$ | $33.57 \pm 2.01$ | $32.00 \pm 1.94$ | $35.59 \pm 3.09$ |
| Emu1-Gen | $23.47 \pm 3.98$ | $24.11 \pm 6.26$ | $27.58 \pm 2.47$ | $29.14 \pm 5.98$ | $24.27 \pm 1.86$ |
| emu2-gen | $26.50 \pm 1.78$ | $18.67 \pm 1.03$ | $23.17 \pm 1.43$ | $34.00 \pm 2.27$ | $28.00 \pm 1.08$ |

Table 33: Results of different models on the Text version of Operator Induction (Accuracy %).

| Model | 0-Shot | 1-Shot | 2-Shot | 4-Shot | 8-Shot |
|---|---|---|---|---|---|
| InternLM-XComposer2 | 15.00 | 50.00 | 73.33 | 75.00 | 83.33 |
| Qwen-VL-Chat | 0.00 | 45.00 | 56.67 | 63.33 | 71.67 |
| LLaVA-Next-Vicuna-7B | 10.00 | 40.00 | 53.33 | 60.00 | 68.33 |

Table 34: Results of different models on the text version of Interleaved Operator Induction (Accuracy %).

| Model | 0-Shot | 1-Shot | 2-Shot | 4-Shot | 8-Shot |
|---|---|---|---|---|---|
| InternLM-XComposer2 | 8.33 | 35.00 | 36.67 | 46.67 | 78.33 |
| Qwen-VL-Chat | 0.00 | 50.00 | 55.00 | 61.67 | 66.67 |
| LLaVA-Next-Vicuna-7B | 16.67 | 41.67 | 45.00 | 53.33 | 70.00 |

Table 35: Results of different models on the text version of CLEVR dataset (Accuracy %).

| Model | 0-Shot | 1-Shot | 2-Shot | 4-Shot | 8-Shot |
|---|---|---|---|---|---|
| InternLM-XComposer2 | 0.00 | 45.00 | 43.00 | 42.00 | 41.00 |
| Qwen-VL-Chat | 0.00 | 49.50 | 47.50 | 54.00 | 53.50 |
| LLaVA-Next-Vicuna-7B | 0.00 | 43.00 | 38.50 | 37.50 | 36.50 |

Table 36: Results of different models on the text version of Text-to-Image Fast Mini-ImageNet (Accuracy %).

| Model | 0-Shot | 1-Shot | 2-Shot | 4-Shot | 5-Shot |
|---|---|---|---|---|---|
| GILL | 0.00 | 18.50 | 20.00 | 20.50 | 18.50 |
| SEED-LLaMA-8B | 0.00 | 16.30 | 15.20 | 16.50 | 14.20 |
| SEED-LLaMA-14B | 1.50 | 23.00 | 20.00 | 22.50 | 15.50 |
| Emu1-Gen | 0.50 | 28.60 | 29.10 | 24.20 | 20.00 |
| Emu2-Gen | 0.20 | 32.40 | 38.80 | 40.50 | 42.10 |

Table 37: Results of different models on the text version of CobSAT: Total accuracies (%).

| Model | 0-Shot | 1-Shot | 2-Shot | 4-Shot | 8-Shot |
|---|---|---|---|---|---|
| GILL | 6.00 | 13.00 | 20.50 | 22.50 | 23.50 |
| SEED-LLaMA-8B | 0.50 | 14.50 | 15.50 | 30.50 | 32.00 |
| SEED-LLaMA-14B | 6.00 | 13.50 | 28.00 | 34.00 | 40.50 |
| Emu1-Gen | 2.50 | 11.00 | 19.50 | 23.50 | 20.00 |
| Emu2-Gen | 7.50 | 19.50 | 32.50 | 46.50 | 45.00 |

Table 38: Results of different models on the text version of CobSAT: Latent accuracies (%).

| Model | 0-Shot | 1-Shot | 2-Shot | 4-Shot | 8-Shot |
|---|---|---|---|---|---|
| GILL | 6.50 | 33.00 | 39.00 | 37.50 | 38.00 |
| SEED-LLaMA-8B | 4.00 | 17.50 | 18.50 | 35.00 | 46.00 |
| SEED-LLaMA-14B | 6.50 | 60.00 | 55.50 | 60.00 | 66.00 |
| Emu1-Gen | 6.00 | 24.00 | 31.50 | 43.50 | 42.00 |
| Emu2-Gen | 12.00 | 74.00 | 86.00 | 92.50 | 88.50 |

Table 39: Results of different models on the text version of CobSAT: Non-latent accuracies (%).

| Model | 0-Shot | 1-Shot | 2-Shot | 4-Shot | 8-Shot |
|---|---|---|---|---|---|
| GILL | 86.00 | 44.50 | 62.50 | 67.00 | 71.50 |
| SEED-LLaMA-8B | 21.00 | 80.00 | 83.50 | 80.50 | 74.50 |
| SEED-LLaMA-14B | 90.00 | 21.50 | 56.50 | 63.00 | 67.50 |
| Emu1-Gen | 30.00 | 33.50 | 52.00 | 48.50 | 45.50 |
| Emu2-Gen | 68.50 | 22.50 | 37.50 | 50.50 | 49.00 |

Table 40: Comparisons of in-context learning ability with and without instruction-following fine-tuning on Fast Open-Ended MiniImageNet Dataset.

| Model | 0-Shot | 1-Shot | 2-Shot | 4-Shot | 5-Shot |
|---|---|---|---|---|---|
| IDEFICS-9B | 0.00 | 16.00 | 47.50 | 58.00 | 56.00 |
| IDEFICS-9B-Instruct | 0.00 | 22.00 | 52.00 | 53.83 | 59.17 |
| Qwen-VL | 0.00 | 35.50 | 79.50 | 68.00 | 67.00 |
| Qwen-VL-Chat | 0.00 | 0.50 | 47.33 | 58.00 | 55.17 |

Table 41: Comparisons of in-context learning ability with and without instruction-following fine-tuning on TextOCR Dataset (Accuracy %).

| Model | 0-Shot | 1-Shot | 2-Shot | 4-Shot | 8-Shot |
|---|---|---|---|---|---|
| IDEFICS-9B | 3.50 | 16.50 | 22.50 | 25.00 | 26.00 |
| IDEFICS-9B-Instruct | 16.50 | 22.50 | 19.83 | 22.83 | 28.00 |
| Qwen-VL | 0.00 | 27.00 | 28.50 | 30.50 | 37.00 |
| Qwen-VL-Chat | 4.83 | 17.17 | 21.50 | 22.33 | 24.17 |

Table 42: Comparisons of in-context learning ability with and without instruction-following fine-tuning on CLEVR Dataset (Accuracy %).

| Model | 0-Shot | 1-Shot | 2-Shot | 4-Shot | 8-Shot |
|---|---|---|---|---|---|
| IDEFICS-9B | 0.00 | 19.50 | 26.00 | 25.00 | 29.00 |
| IDEFICS-9B-Instruct | 0.00 | 30.33 | 29.50 | 27.67 | 27.17 |
| Qwen-VL | 2.50 | 18.50 | 17.50 | 24.00 | 26.00 |
| Qwen-VL-Chat | 0.00 | 29.83 | 25.33 | 26.83 | 30.17 |

Table 43: Comparisons of in-context learning ability with and without instruction-following fine-tuning on Operator Induction Dataset (Accuracy %).

| Model | 0-Shot | 1-Shot | 2-Shot | 4-Shot | 8-Shot |
|---|---|---|---|---|---|
| IDEFICS-9B | 5.00 | 16.67 | 8.33 | 10.00 | 3.33 |
| IDEFICS-9B-Instruct | 11.67 | 14.44 | 10.56 | 7.78 | 11.11 |
| Qwen-VL | 15.00 | 26.67 | 36.67 | 46.67 | 56.67 |
| Qwen-VL-Chat | 15.00 | 10.00 | 17.22 | 18.89 | 25.00 |

Table 44: Comparisons of in-context learning ability with and without instruction-following fine-tuning on Interleaved Operator Induction Dataset (Accuracy %).

| Model | 0-Shot | 1-Shot | 2-Shot | 4-Shot | 8-Shot |
|---|---|---|---|---|---|
| IDEFICS-9B | 13.33 | 8.33 | 5.00 | 10.00 | 3.33 |
| IDEFICS-9B-Instruct | 15.00 | 5.56 | 6.11 | 6.11 | 5.00 |
| Qwen-VL | 0.00 | 13.33 | 13.33 | 8.33 | 11.67 |
| Qwen-VL-Chat | 16.67 | 9.44 | 8.33 | 8.89 | 5.56 |

## D.4 SUPPLEMENTARY RESULTS

### D.4.1 SCALING TO MANY SHOTS

Table 45 to 48

### D.4.2 CHAIN-OF-THOUGHT PROMPTING

Table 49 to 53

### D.4.3 REPEATING SUPPORT SET

Table 54 to 56.

### D.4.4 DIFFERENT LEVELS OF TASK DESCRIPTIONS

Table 57 to 60.

### D.4.5 EMERGENT BEHAVIOR ANALYSIS - LLaVA-ONEVISION WITH DIFFERENT MODEL SIZES

Table 61 to 66.

### D.4.6 ANALYSIS ON CONTEXT EXTENSION

Table 67 to 70.

Table 45: Results of many shots on CLEVR dataset.

| Model | 16-Shot | 32-Shot | 64-Shot |
|---|---|---|---|
| OpenFlamingo-9B | $22.00 \pm 1.47$ | $25.33 \pm 1.65$ | $25.67 \pm 2.39$ |
| IDEFICS-9B-Instruct | $28.17 \pm 2.66$ | $29.00 \pm 1.08$ | $30.50 \pm 1.78$ |
| InternLM-X2 | $14.67 \pm 1.70$ | $15.50 \pm 1.08$ | $16.33 \pm 1.03$ |
| LongVA-7B | $23.33 \pm 1.03$ | $23.67 \pm 1.65$ | $29.50 \pm 3.63$ |
| Mantis-Idefics2 | $19.50 \pm 0.82$ | $22.33 \pm 1.31$ | $10.17 \pm 2.25$ |

Table 46: Results of many shots on Operator Induction dataset.

| Model | 16-Shot | 32-Shot | 64-Shot |
|---|---|---|---|
| OpenFlamingo-9B | $13.33 \pm 3.60$ | $8.89 \pm 1.57$ | $11.67 \pm 1.36$ |
| IDEFICS-9B-Instruct | $5.00 \pm 3.60$ | $7.78 \pm 1.57$ | $5.00 \pm 1.36$ |
| InternLM-X2 | $0.00 \pm 0.00$ | $0.00 \pm 0.00$ | $0.00 \pm 0.00$ |
| LongVA-7B | $2.22 \pm 0.78$ | $3.33 \pm 0.00$ | $3.89 \pm 0.79$ |
| Mantis-Idefics2 | $20.00 \pm 4.91$ | $16.67 \pm 1.36$ | $17.78 \pm 0.78$ |

Table 47: Results of many shots on Interleaved Operator Induction dataset.

| Model | 16-Shot | 32-Shot | 64-Shot |
|---|---|---|---|
| OpenFlamingo-9B | $8.89 \pm 3.42$ | $8.33 \pm 3.60$ | $11.67 \pm 3.60$ |
| IDEFICS-9B-Instruct | $8.89 \pm 2.08$ | $7.78 \pm 2.08$ | $7.78 \pm 2.83$ |
| InternLM-X2 | $3.89 \pm 0.79$ | $5.00 \pm 1.36$ | $5.00 \pm 1.36$ |
| LongVA-7B | $6.11 \pm 1.57$ | $4.44 \pm 0.79$ | $5.00 \pm 0.00$ |
| Mantis-Idefics2 | $7.22 \pm 2.08$ | $3.89 \pm 2.08$ | $3.89 \pm 2.08$ |

Table 48: Results of many shots on TextOCR dataset.

| Model | 16-Shot | 32-Shot | 64-Shot |
|---|---|---|---|
| OpenFlamingo-9B | $0.00 \pm 0.00$ | $0.00 \pm 0.00$ | $0.00 \pm 0.00$ |
| IDEFICS-9B-Instruct | $29.00 \pm 1.22$ | $33.17 \pm 0.85$ | $33.50 \pm 1.47$ |
| InternLM-X2 | $3.11 \pm 0.58$ | $0.00 \pm 0.00$ | $0.00 \pm 0.00$ |
| LongVA-7B | $15.33 \pm 1.55$ | $10.83 \pm 0.47$ | $5.50 \pm 0.71$ |
| Mantis-Idefics2 | $26.33 \pm 1.43$ | $29.83 \pm 1.65$ | $22.83 \pm 1.70$ |

Table 49: Results with Chain-of-Thought prompting on Operator Induction dataset.

| Model | 0-shot | 1-shot | 2-Shot | 4-Shot | 8-Shot |
|---|---|---|---|---|---|
| Qwen-VL-Chat | 13.67 | 21.00 | 15.33 | 9.33 | 16.67 |
| InternLM-X2 | 28.33 | 32.00 | 35.67 | 30.00 | 5.00 |

Table 50: Results with Chain-of-Thought prompting on Interleaved Operator Induction dataset.

| Model | 0-shot | 1-shot | 2-Shot | 4-Shot | 8-Shot |
|---|---|---|---|---|---|
| Qwen-VL-Chat | 3.33 | 10.00 | 10.00 | 8.33 | 8.33 |
| InternLM-X2 | 18.33 | 5.00 | 5.00 | 10.00 | 8.33 |

Table 51: Results with Chain-of-Thought prompting on TextOCR dataset.

| Model | 0-shot | 1-shot | 2-Shot | 4-Shot | 8-Shot |
|---|---|---|---|---|---|
| Qwen-VL-Chat | 7.00 | 28.50 | 27.50 | 30.50 | 22.50 |
| InternLM-X2 | 12.00 | 1.50 | 0.50 | 2.00 | 0.50 |

Table 52: Results with Chain-of-Thought prompting on CLEVR dataset.

| Model | 0-shot | 1-shot | 2-Shot | 4-Shot | 8-Shot |
|---|---|---|---|---|---|
| Qwen-VL-Chat | 0.50 | 10.50 | 21.50 | 18.00 | 26.00 |
| InternLM-X2 | 4.00 | 21.50 | 20.00 | 26.00 | 27.00 |

Table 53: Results with Chain-of-Thought prompting on Matching Mini-ImageNet dataset.

| Model | 0-shot | 1-shot | 2-Shot | 4-Shot | 5-Shot |
|---|---|---|---|---|---|
| Qwen-VL-Chat | 56.00 | 56.75 | 51.25 | 56.75 | 53.00 |
| InternLM-X2 | 58.25 | 51.50 | 53.00 | 50.00 | 48.50 |

Table 54: Result of Qwen-VL-Chat on Fast Open-Ended MiniImageNet dataset with repeated in-context examples.

| Model | 1-shot | 2-Shot | 4-Shot | 5-Shot |
|---|---|---|---|---|
| No Repeat | 0.50 | 47.33 | 58.00 | 55.17 |
| Repeat x2 | 41.00 | 62.50 | 54.50 | 56.50 |
| Repeat x3 | 62.50 | 55.50 | 61.00 | 62.00 |
| Repeat x4 | 60.00 | 56.50 | 60.00 | 58.50 |

Table 55: Result of Qwen-VL-Chat on Operator Induction dataset with repeated in-context examples.

| Model | 1-shot | 2-Shot | 4-Shot | 8-Shot |
|---|---|---|---|---|
| No repeat | 10.00 | 17.22 | 18.89 | 25.00 |
| Repeat x2 | 5.00 | 15.00 | 23.33 | 25.00 |
| Repeat x3 | 11.67 | 15.00 | 20.00 | 26.67 |
| Repeat x4 | 13.33 | 18.33 | 21.67 | 18.33 |

Table 56: Result of Qwen-VL-Chat on CLEVR dataset with repeated in-context examples.

| Model | 1-shot | 2-Shot | 4-Shot | 8-Shot |
|---|---|---|---|---|
| No Repeat | 29.83 | 25.33 | 26.83 | 30.17 |
| Repeat x2 | 25.50 | 30.50 | 27.50 | 28.50 |
| Repeat x3 | 22.50 | 32.50 | 26.50 | 32.00 |
| Repeat x4 | 19.50 | 31.50 | 23.00 | 27.50 |

Table 57: Results of different models on Fast Open-Ended MiniImageNet dataset (Accuracy %).

| Model | 0-Shot | 1-Shot | 2-Shot | 4-Shot | 5-Shot |
|---|---|---|---|---|---|
| Qwen-VL-Chat – Detailed | $0.00 \pm 0.00$ | $0.50 \pm 0.41$ | $47.33 \pm 2.49$ | $58.00 \pm 2.83$ | $55.17 \pm 2.25$ |
| Qwen-VL-Chat – Concise | $0.00 \pm 0.00$ | $0.83 \pm 0.62$ | $48.00 \pm 2.45$ | $59.00 \pm 0.41$ | $52.50 \pm 2.68$ |
| Qwen-VL-Chat – None | $0.00 \pm 0.00$ | $6.33 \pm 0.47$ | $56.17 \pm 1.65$ | $57.67 \pm 0.85$ | $53.83 \pm 2.78$ |
| LLaVA-Next-7B – Detailed | $0.00 \pm 0.00$ | $22.17 \pm 4.03$ | $33.67 \pm 2.25$ | $0.00 \pm 0.00$ | $0.33 \pm 0.24$ |
| LLaVA-Next-7B – Concise | $0.00 \pm 0.00$ | $24.00 \pm 0.71$ | $34.50 \pm 2.68$ | $0.00 \pm 0.00$ | $0.33 \pm 0.24$ |
| LLaVA-Next-7B – None | $0.00 \pm 0.00$ | $16.67 \pm 2.01$ | $32.00 \pm 2.55$ | $0.33 \pm 0.24$ | $0.17 \pm 0.24$ |
| OpenFlamingo-9B – Detailed | $0.00 \pm 0.00$ | $39.50 \pm 1.22$ | $58.17 \pm 3.57$ | $51.17 \pm 0.85$ | $54.50 \pm 5.66$ |
| OpenFlamingo-9B – Concise | $0.00 \pm 0.00$ | $36.50 \pm 0.41$ | $51.67 \pm 2.78$ | $52.17 \pm 0.62$ | $49.33 \pm 1.25$ |
| OpenFlamingo-9B – None | $0.00 \pm 0.00$ | $38.17 \pm 1.03$ | $52.17 \pm 2.46$ | $49.17 \pm 0.85$ | $49.33 \pm 1.25$ |
| InternLM-X2 – Detailed | $0.00 \pm 0.00$ | $14.83 \pm 1.03$ | $38.00 \pm 1.78$ | $49.00 \pm 1.78$ | $50.33 \pm 3.86$ |
| InternLM-X2 – Concise | $0.00 \pm 0.00$ | $19.50 \pm 1.47$ | $40.33 \pm 1.89$ | $48.83 \pm 0.85$ | $49.17 \pm 1.93$ |
| InternLM-X2 – None | $0.00 \pm 0.00$ | $22.00 \pm 2.04$ | $43.00 \pm 2.16$ | $46.33 \pm 3.06$ | $48.17 \pm 0.62$ |
| IDEFICS-9B – Detailed | $0.00 \pm 0.00$ | $22.00 \pm 0.41$ | $52.00 \pm 2.94$ | $53.83 \pm 0.94$ | $59.17 \pm 6.20$ |
| IDEFICS-9B – Concise | $0.00 \pm 0.00$ | $28.50 \pm 1.78$ | $53.83 \pm 4.09$ | $53.83 \pm 0.94$ | $55.67 \pm 2.09$ |
| IDEFICS-9B – None | $0.00 \pm 0.00$ | $37.17 \pm 4.29$ | $52.17 \pm 4.48$ | $53.17 \pm 1.25$ | $55.50 \pm 1.47$ |

Table 58: Results of different models on CLEVR Count Induction dataset using different levels of task description (Accuracy %).

| Model | 0-Shot | 1-Shot | 2-Shot | 4-Shot | 8-Shot |
|---|---|---|---|---|---|
| Qwen-VL-Chat – Detailed | $0.00 \pm 0.00$ | $29.83 \pm 4.55$ | $25.33 \pm 3.47$ | $26.83 \pm 3.06$ | $30.17 \pm 2.95$ |
| Qwen-VL-Chat – Concise | $0.00 \pm 0.00$ | $24.67 \pm 2.32$ | $25.67 \pm 0.85$ | $25.33 \pm 1.65$ | $24.83 \pm 2.32$ |
| Qwen-VL-Chat – None | $1.00 \pm 0.00$ | $25.17 \pm 2.72$ | $24.33 \pm 1.31$ | $24.83 \pm 1.31$ | $24.67 \pm 2.36$ |
| LLaVA-Next-7B – Detailed | $0.00 \pm 0.00$ | $25.17 \pm 6.64$ | $24.83 \pm 4.90$ | $17.83 \pm 4.59$ | $0.17 \pm 0.24$ |
| LLaVA-Next-7B – Concise | $0.00 \pm 0.00$ | $25.00 \pm 3.49$ | $27.00 \pm 3.89$ | $20.00 \pm 2.48$ | $0.00 \pm 0.00$ |
| LLaVA-Next-7B – None | $0.00 \pm 0.00$ | $15.50 \pm 2.12$ | $23.83 \pm 2.87$ | $12.83 \pm 1.70$ | $0.17 \pm 0.24$ |
| OpenFlamingo-9B – Detailed | $0.00 \pm 0.00$ | $17.83 \pm 2.25$ | $17.00 \pm 2.27$ | $18.83 \pm 1.03$ | $16.33 \pm 1.43$ |
| OpenFlamingo-9B – Concise | $0.00 \pm 0.00$ | $15.33 \pm 2.39$ | $19.00 \pm 2.27$ | $20.00 \pm 0.71$ | $18.33 \pm 3.09$ |
| OpenFlamingo-9B – None | $0.00 \pm 0.00$ | $15.33 \pm 0.94$ | $18.17 \pm 1.03$ | $21.33 \pm 1.89$ | $19.33 \pm 2.78$ |
| InternLM-X2 – Detailed | $1.83 \pm 0.24$ | $26.00 \pm 1.63$ | $24.67 \pm 5.25$ | $20.00 \pm 2.94$ | $22.83 \pm 0.85$ |
| InternLM-X2 – Concise | $1.00 \pm 0.00$ | $19.33 \pm 2.25$ | $20.17 \pm 1.31$ | $9.50 \pm 1.41$ | $12.33 \pm 2.32$ |
| InternLM-X2 – None | $1.50 \pm 0.00$ | $26.67 \pm 2.09$ | $24.67 \pm 2.01$ | $25.17 \pm 1.18$ | $23.17 \pm 2.25$ |
| IDEFICS-9B – Detailed | $0.00 \pm 0.00$ | $30.33 \pm 2.25$ | $29.50 \pm 1.47$ | $27.67 \pm 2.05$ | $27.17 \pm 2.87$ |
| IDEFICS-9B – Concise | $1.00 \pm 0.00$ | $30.67 \pm 1.84$ | $31.00 \pm 3.94$ | $26.17 \pm 1.55$ | $26.83 \pm 0.62$ |
| IDEFICS-9B – None | $0.00 \pm 0.00$ | $30.83 \pm 1.43$ | $31.33 \pm 2.95$ | $28.50 \pm 1.78$ | $28.00 \pm 0.41$ |

Table 59: Results of different models on Operator Induction dataset using different levels of task description (Accuracy %).

| Model | 0-Shot | 1-Shot | 2-Shot | 4-Shot | 8-Shot |
|---|---|---|---|---|---|
| Qwen-VL-Chat – Detailed | $15.00 \pm 0.00$ | $10.00 \pm 1.36$ | $17.22 \pm 3.14$ | $18.89 \pm 1.57$ | $25.00 \pm 2.72$ |
| Qwen-VL-Chat – Concise | $15.00 \pm 0.00$ | $7.22 \pm 2.08$ | $15.56 \pm 3.42$ | $17.78 \pm 2.08$ | $27.22 \pm 0.79$ |
| Qwen-VL-Chat – None | $15.00 \pm 0.00$ | $8.33 \pm 2.36$ | $14.44 \pm 2.83$ | $18.33 \pm 2.72$ | $27.22 \pm 0.79$ |
| LLaVA-Next-7B – Detailed | $10.56 \pm 1.57$ | $6.11 \pm 1.57$ | $5.56 \pm 2.08$ | $3.33 \pm 2.72$ | $0.00 \pm 0.00$ |
| LLaVA-Next-7B – Concise | $5.00 \pm 0.00$ | $7.22 \pm 0.79$ | $5.56 \pm 2.08$ | $4.44 \pm 2.08$ | $1.11 \pm 0.79$ |
| LLaVA-Next-7B – None | $8.33 \pm 0.00$ | $6.11 \pm 0.79$ | $5.56 \pm 1.57$ | $4.44 \pm 1.57$ | $0.56 \pm 0.79$ |
| OpenFlamingo-9B – Detailed | $5.00 \pm 0.00$ | $2.22 \pm 3.14$ | $1.67 \pm 1.36$ | $2.78 \pm 0.79$ | $7.78 \pm 2.08$ |
| OpenFlamingo-9B – Concise | $6.67 \pm 0.00$ | $5.00 \pm 3.60$ | $4.44 \pm 3.14$ | $4.44 \pm 1.57$ | $9.44 \pm 1.57$ |
| OpenFlamingo-9B – None | $6.67 \pm 0.00$ | $5.00 \pm 3.60$ | $3.33 \pm 2.36$ | $4.44 \pm 2.08$ | $11.67 \pm 3.60$ |
| InternLM-X2 – Detailed | $26.11 \pm 3.14$ | $40.00 \pm 10.80$ | $40.00 \pm 4.91$ | $39.44 \pm 7.49$ | $28.89 \pm 19.83$ |
| InternLM-X2 – Concise | $18.33 \pm 0.00$ | $29.44 \pm 3.42$ | $22.78 \pm 2.83$ | $18.33 \pm 1.36$ | $16.67 \pm 2.36$ |
| InternLM-X2 – None | $18.33 \pm 0.00$ | $13.33 \pm 2.36$ | $12.78 \pm 2.83$ | $12.22 \pm 2.08$ | $16.67 \pm 2.72$ |
| IDEFICS-9B – Detailed | $11.67 \pm 0.00$ | $14.44 \pm 0.79$ | $10.56 \pm 2.08$ | $7.78 \pm 2.08$ | $11.11 \pm 1.57$ |
| IDEFICS-9B – Concise | $15.00 \pm 0.00$ | $13.89 \pm 2.83$ | $12.22 \pm 0.79$ | $8.89 \pm 0.79$ | $8.33 \pm 3.60$ |
| IDEFICS-9B – None | $15.00 \pm 0.00$ | $17.22 \pm 2.83$ | $10.56 \pm 0.79$ | $10.56 \pm 2.08$ | $7.78 \pm 3.93$ |

Table 60: Results of different models on TextOCR dataset using different levels of task description (Accuracy %).

| Model | 0-Shot | 1-Shot | 2-Shot | 4-Shot | 8-Shot |
|---|---|---|---|---|---|
| Qwen-VL-Chat – Detailed | $4.83 \pm 6.84$ | $17.17 \pm 1.43$ | $21.50 \pm 1.08$ | $22.33 \pm 1.31$ | $24.17 \pm 0.24$ |
| Qwen-VL-Chat – Concise | $0.00 \pm 0.00$ | $8.00 \pm 0.82$ | $9.50 \pm 0.41$ | $9.83 \pm 0.62$ | $9.17 \pm 0.24$ |
| Qwen-VL-Chat – None | $0.00 \pm 0.00$ | $9.67 \pm 0.62$ | $10.33 \pm 0.47$ | $10.67 \pm 0.47$ | $9.33 \pm 0.47$ |
| LLaVA-Next-7B – Detailed | $24.67 \pm 2.25$ | $0.83 \pm 0.24$ | $0.33 \pm 0.24$ | $0.00 \pm 0.00$ | $0.00 \pm 0.00$ |
| LLaVA-Next-7B – Concise | $8.50 \pm 0.00$ | $0.00 \pm 0.00$ | $0.00 \pm 0.00$ | $0.00 \pm 0.00$ | $0.00 \pm 0.00$ |
| LLaVA-Next-7B – None | $10.00 \pm 0.00$ | $0.00 \pm 0.00$ | $0.00 \pm 0.00$ | $0.00 \pm 0.00$ | $0.00 \pm 0.00$ |
| OpenFlamingo-9B – Detailed | $0.00 \pm 0.00$ | $0.00 \pm 0.00$ | $0.00 \pm 0.00$ | $0.00 \pm 0.00$ | $0.00 \pm 0.00$ |
| OpenFlamingo-9B – Concise | $0.00 \pm 0.00$ | $0.00 \pm 0.00$ | $0.00 \pm 0.00$ | $0.00 \pm 0.00$ | $0.00 \pm 0.00$ |
| OpenFlamingo-9B – None | $0.00 \pm 0.00$ | $0.00 \pm 0.00$ | $0.00 \pm 0.00$ | $0.00 \pm 0.00$ | $0.00 \pm 0.00$ |
| InternLM-X2 – Detailed | $8.67 \pm 4.01$ | $3.83 \pm 0.62$ | $10.50 \pm 0.71$ | $16.00 \pm 2.48$ | $11.83 \pm 2.95$ |
| InternLM-X2 – Concise | $0.50 \pm 0.00$ | $0.50 \pm 0.41$ | $0.83 \pm 0.47$ | $2.33 \pm 1.03$ | $0.00 \pm 0.00$ |
| InternLM-X2 – None | $0.50 \pm 0.00$ | $0.50 \pm 0.41$ | $1.33 \pm 0.47$ | $3.67 \pm 2.09$ | $0.00 \pm 0.00$ |
| IDEFICS-9B – Detailed | $16.50 \pm 0.00$ | $22.50 \pm 1.08$ | $19.83 \pm 0.62$ | $22.83 \pm 1.31$ | $28.00 \pm 1.63$ |
| IDEFICS-9B – Concise | $3.00 \pm 0.00$ | $2.50 \pm 0.41$ | $5.50 \pm 0.41$ | $5.83 \pm 0.24$ | $6.17 \pm 0.47$ |
| IDEFICS-9B – None | $4.00 \pm 0.00$ | $2.67 \pm 0.62$ | $5.33 \pm 0.47$ | $6.00 \pm 0.41$ | $6.33 \pm 0.62$ |

Table 61: Performance of LLaVA-OneVision of different model sizes on CLEVR dataset.

| Model | 0-Shot | 1-Shot | 2-Shot | 4-Shot | 8-Shot |
|---|---|---|---|---|---|
| LLaVA-OneVision-0.5B | $0.00 \pm 0.00$ | $3.83 \pm 0.24$ | $11.50 \pm 2.68$ | $12.83 \pm 0.62$ | $24.33 \pm 2.05$ |
| LLaVA-OneVision-7B | $5.50 \pm 0.00$ | $38.17 \pm 2.25$ | $33.83 \pm 0.85$ | $31.50 \pm 4.64$ | $28.33 \pm 0.24$ |
| LLaVA-OneVision-72B | $0.50 \pm 0.00$ | $37.17 \pm 2.01$ | $32.00 \pm 0.71$ | $31.83 \pm 1.84$ | $42.33 \pm 0.85$ |

Table 62: Performance of LLaVA-OneVision of different model sizes on TextOCR dataset.

| Model | 0-Shot | 1-Shot | 2-Shot | 4-Shot | 8-Shot |
|---|---|---|---|---|---|
| LLaVA-OneVision-0.5B | $16.50 \pm 0.00$ | $2.33 \pm 0.62$ | $3.00 \pm 0.41$ | $13.33 \pm 0.94$ | $25.50 \pm 0.71$ |
| LLaVA-OneVision-7B | $39.00 \pm 0.00$ | $35.67 \pm 1.18$ | $42.17 \pm 1.03$ | $42.17 \pm 0.24$ | $44.67 \pm 1.84$ |
| LLaVA-OneVision-72B | $48.50 \pm 0.00$ | $50.67 \pm 1.70$ | $51.17 \pm 0.24$ | $50.67 \pm 2.09$ | $51.67 \pm 0.62$ |

Table 63: Performance of LLaVA-OneVision of different model sizes on Fast Open-Ended Matching MiniImageNet dataset.

| Model | 0-Shot | 1-Shot | 2-Shot | 4-Shot | 8-Shot |
|---|---|---|---|---|---|
| LLaVA-OneVision-0.5B | $0.00 \pm 0.00$ | $20.33 \pm 2.25$ | $36.83 \pm 1.43$ | $36.83 \pm 2.01$ | $41.33 \pm 1.55$ |
| LLaVA-OneVision-7B | $0.00 \pm 0.00$ | $13.50 \pm 1.08$ | $48.17 \pm 3.57$ | $49.50 \pm 1.87$ | $49.50 \pm 1.08$ |
| LLaVA-OneVision-72B | $0.00 \pm 0.00$ | $12.67 \pm 0.94$ | $88.00 \pm 0.82$ | $98.67 \pm 0.47$ | $98.33 \pm 0.62$ |

Table 64: Performance of LLaVA-OneVision of different model sizes on Operator Induction dataset.

| Model | 0-Shot | 1-Shot | 2-Shot | 4-Shot | 8-Shot |
|---|---|---|---|---|---|
| LLaVA-OneVision-0.5B | $21.67 \pm 0.00$ | $6.67 \pm 1.36$ | $6.11 \pm 2.08$ | $6.67 \pm 2.36$ | $6.67 \pm 3.60$ |
| LLaVA-OneVision-7B | $35.00 \pm 0.00$ | $45.00 \pm 2.72$ | $41.11 \pm 3.42$ | $45.56 \pm 0.79$ | $28.33 \pm 0.00$ |
| LLaVA-OneVision-72B | $33.33 \pm 0.00$ | $75.56 \pm 3.42$ | $60.56 \pm 6.71$ | $68.89 \pm 1.57$ | $57.78 \pm 2.08$ |

Table 65: Performance of LLaVA-OneVision of different model sizes on Interleaved Operator Induction dataset.

| Model | 0-Shot | 1-Shot | 2-Shot | 4-Shot | 8-Shot |
|---|---|---|---|---|---|
| LLaVA-OneVision-0.5B | $13.33 \pm 0.00$ | $5.56 \pm 2.83$ | $4.44 \pm 2.83$ | $5.56 \pm 2.83$ | $1.67 \pm 0.00$ |
| LLaVA-OneVision-7B | $38.33 \pm 0.00$ | $15.00 \pm 1.36$ | $15.56 \pm 0.79$ | $22.22 \pm 3.43$ | $20.55 \pm 2.08$ |
| LLaVA-OneVision-72B | $38.33 \pm 0.00$ | $42.78 \pm 2.08$ | $41.11 \pm 2.08$ | $36.67 \pm 2.72$ | $47.78 \pm 10.03$ |

Table 66: Performance of LLaVA-OneVision of different model sizes on Fast Matching MiniImageNet dataset.

| Model | 0-Shot | 1-Shot | 2-Shot | 4-Shot | 5-Shot |
|---|---|---|---|---|---|
| LLaVA-OneVision-0.5B | $0.00 \pm 0.00$ | $8.92 \pm 1.04$ | $34.77 \pm 2.61$ | $49.92 \pm 0.72$ | $48.15 \pm 0.90$ |
| LLaVA-OneVision-7B | $0.00 \pm 0.00$ | $15.67 \pm 1.12$ | $49.58 \pm 0.12$ | $50.00 \pm 0.00$ | $50.00 \pm 0.00$ |
| LLaVA-OneVision-72B | $0.00 \pm 0.00$ | $28.50 \pm 2.07$ | $58.08 \pm 0.92$ | $68.83 \pm 0.31$ | $70.33 \pm 0.66$ |

Table 67: Performance comparisons of before and after context extension (SelfExtend) on CLEVR dataset.

| Model | 0-Shot | 1-Shot | 2-Shot | 4-Shot | 8-Shot |
|---|---|---|---|---|---|
| LLaVA-Next-7B (w/o SelfExtend) | $0.00 \pm 0.00$ | $25.17 \pm 6.64$ | $24.83 \pm 4.90$ | $17.83 \pm 4.59$ | $19.17 \pm 0.24$ |
| LLaVA-Next-7B (w/ SelfExtend) | $0.00 \pm 0.00$ | $28.83 \pm 1.31$ | $29.00 \pm 2.27$ | $27.17 \pm 1.89$ | $24.50 \pm 1.87$ |
| VILA-7B (w/o SelfExtend) | $3.50 \pm 0.00$ | $34.00 \pm 2.86$ | $32.67 \pm 1.43$ | $31.50 \pm 2.16$ | $34.33 \pm 2.39$ |
| VILA-7B (w/ SelfExtend) | $4.00 \pm 0.00$ | $34.83 \pm 1.89$ | $32.00 \pm 0.82$ | $34.50 \pm 3.08$ | $32.00 \pm 3.08$ |

Table 68: Performance comparisons of before and after context extension (SelfExtend) on TextOCR dataset.

| Model | 0-Shot | 1-Shot | 2-Shot | 4-Shot | 8-Shot |
|---|---|---|---|---|---|
| LLaVA-Next-7B (w/o SelfExtend) | $24.67 \pm 2.25$ | $0.83 \pm 0.24$ | $0.33 \pm 0.24$ | $0.00 \pm 0.00$ | $0.00 \pm 0.00$ |
| LLaVA-Next-7B (w/ SelfExtend) | $26.00 \pm 0.00$ | $0.83 \pm 0.24$ | $0.17 \pm 0.24$ | $1.00 \pm 0.41$ | $0.50 \pm 0.71$ |
| VILA-7B (w/o SelfExtend) | $28.00 \pm 0.00$ | $6.17 \pm 1.03$ | $22.17 \pm 0.24$ | $26.83 \pm 0.47$ | $30.17 \pm 1.03$ |
| VILA-7B (w SelfExtend) | $28.00 \pm 0.00$ | $6.83 \pm 0.85$ | $20.67 \pm 0.24$ | $25.50 \pm 0.82$ | $29.67 \pm 1.55$ |

Table 69: Performance comparisons of before and after context extension (SelfExtend) on Operator Induction dataset.

| Model | 0-Shot | 1-Shot | 2-Shot | 4-Shot | 8-Shot |
|---|---|---|---|---|---|
| LLaVA-Next-7B (w/o SelfExtend) | $10.56 \pm 1.57$ | $6.11 \pm 1.57$ | $5.56 \pm 2.08$ | $3.33 \pm 2.72$ | $0.00 \pm 0.00$ |
| LLaVA-Next-7B (w/ SelfExtend) | $11.67 \pm 0.00$ | $7.22 \pm 0.78$ | $5.00 \pm 2.72$ | $6.11 \pm 0.79$ | $3.89 \pm 0.79$ |
| VILA-7B (w/o SelfExtend) | $28.33 \pm 0.00$ | $11.11 \pm 4.37$ | $6.67 \pm 3.60$ | $7.78 \pm 0.78$ | $8.33 \pm 2.72$ |
| VILA-7B (w SelfExtend) | $28.33 \pm 0.00$ | $9.44 \pm 4.38$ | $5.55 \pm 2.08$ | $6.67 \pm 1.36$ | $6.11 \pm 1.57$ |

Table 70: Performance comparisons of before and after context extension (SelfExtend) on Fast Open-Ended MiniImageNet dataset.

| Model | 0-Shot | 1-Shot | 2-Shot | 4-Shot | 5-Shot |
|---|---|---|---|---|---|
| LLaVA-Next-7B (w/o SelfExtend) | $0.00 \pm 0.00$ | $22.17 \pm 4.03$ | $33.67 \pm 2.25$ | $37.24 \pm 1.02$ | $36.95 \pm 0.24$ |
| LLaVA-Next-7B (w/ SelfExtend) | $0.00 \pm 0.00$ | $29.33 \pm 1.43$ | $42.33 \pm 2.46$ | $51.00 \pm 2.68$ | $50.17 \pm 1.18$ |
| VILA-7B (w/o SelfExtend) | $0.00 \pm 0.00$ | $28.83 \pm 1.43$ | $38.17 \pm 2.72$ | $37.67 \pm 3.68$ | $37.33 \pm 1.18$ |
| VILA-7B (w/ SelfExtend) | $0.00 \pm 0.00$ | $27.00 \pm 2.04$ | $54.00 \pm 2.86$ | $45.67 \pm 3.12$ | $47.00 \pm 1.47$ |

