# OpenReview forum: "VL-ICL Bench: The Devil in the Details of Multimodal In-Context Learning"
_ICLR.cc/2025/Conference — ICLR 2025 Poster_

### Official Review · Reviewer_EuE8 · 2024-10-31

**Soundness:** 3
**Presentation:** 3
**Contribution:** 3
**Rating:** 8
**Confidence:** 5

**Summary:**

This paper introduces a comprehensive dataset, VL-ICL, for evaluating VLLM. Unlike previous datasets that primarily focus on tasks such as question answering, OCR, or image caption generation, this dataset takes into account the contextual learning capabilities of both image-to-text (I2T) and text-to-image (T2I) modalities. It assesses the in-context learning (ICL) abilities of a wide range of multimodal models and reflects on potential issues these models may face. Additionally, it summarizes some phenomena that might arise during multimodal contextual learning, offering valuable insights for future experiments and research in the multimodal field.

**Strengths:**

(1) Compared to previous image captioning or visual question answering tasks, this dataset introduces new tasks designed to assess multimodal contextual learning capabilities. This provides significant benefits for future model design, enhancing models' visual understanding abilities, and conducting comprehensive testing of multimodal models. Additionally, it includes detailed evaluations of few-shot language-to-image generation tasks.

(2) The paper thoroughly tests the contextual learning abilities of open-source VLLM with different architectures, backed by extensive experimentation.

(3) The paper documents and provides detailed explanations of a series of phenomena observed in vision ICL, offering insights for future research exploration.

**Weaknesses:**

(1) I believe this paper should expand on the impact of different ICD selections on ICL. There are already some papers that have demonstrated the role of ICD in ICL within VLLM.  Furthermore, in section 2.2 of the article, some phenomena related to multimodal ICL (mentioned in Section 2.2 :**For example, in captioning zero-shot VLLMs tend to produce more verbose captions than COCO ground-truth, and they learn to be more concise through ICL. Meanwhile, for VQA, there is a standard practice of evaluating based on string match between the ground-truth answer and the model-provided answer. For example, VizWiz has unanswerable questions, which some VLLMs answer with "I don’t know" which would not be string matched against a ground truth "Unanswerable". Some models thus learn about answer-formatting (e.g., preferred terminology; avoid using any preface or postface that may throw of a string match) from the context set. This is indeed a kind of ICL, but perhaps not what one expects to be learning in VQA. To validate this conjecture, we repeat the previous evaluation, but using soft matching to eliminate the impact of answer format learning.** ) have been previously discussed in several papers and thus the authors should refer to these studies.
- What Makes Good Examples for Visual In-Context Learning?
-How to Configure Good In-Context Sequence for Visual Question Answering
- Understanding and Improving In-Context Learning on Vision-language Models
- Exploring diverse in-context configurations for image captioning
If you could test the impact of these sample selections within VL-ICL and include appropriate citations, I would consider raising the score.

**Questions:**

(1) I noticed that the number of dataset samples/test samples is relatively small. Does this dataset suffer from a significant issue of imbalanced class sample distribution?

(2) In section 4.3, you mentioned an approach where language text is used to replace images for ICL inference. How exactly was this implemented? Did you directly remove the images from the ICL setup and only keep the questions and answers, or did you generate captions for the images to perform contextual learning inference? Or was another method used?

(3) Can you explore the impact of different model architectures on ICL abilities? In your experiments, there are models with cross-attention architectures similar to Flamingo, as well as models like LLAVA that embed visual features into the text embedding space. Does this have a significant impact on the ICL performance of the models?

---

> ### Author Response · Authors · 2024-11-21
> **Responses to Reviewer EuE8**
>
> Thank you very much for your review and questions. Here are our responses to your comments.
>
> ### W1. Experiments on In-context demonstration selection strategy
> Thank you for your suggestion, we agree this is a useful additional experiment. We employ a similarity-based selection strategy mentioned in all suggested papers for experiments. Specifically, for image-to-text tasks (CLEVR and TextOCR), we use CLIP image embeddings to retrieve the top-k most similar images and their corresponding text. Conversely, for text-to-image tasks (CobSAT), we use CLIP text embeddings to retrieve the top-k most similar texts and their associated images. Additionally, we experiment with three different arrangements of the selected top-k demonstrations: (1) from the most similar one to the least similar one, followed by the query; (2) from the least similar one to the most similar one, followed by the query; and (3) a random order of the top-k demonstrations. These are compared to the original setup in the paper, which used random selection.
> We select several top-performing models for our experiments, with the results presented below. In each cell, the first three numbers correspond to the three arrangements of the selected top-k demonstrations (1), (2), and (3), while the final value represents the baseline performance with random selection. For both text-to-image and image-to-text tasks, similarity-based retrieval consistently enhances performance. Specifically, similarity-based selection methods have a ~3% improvement in peak accuracy and a 3% boost in ICL efficiency compared to baseline (details in appendix Table 10). These findings demonstrate that selection strategies can enhance ICL tasks, highlighting a promising direction for future work aimed at improving performance across various tasks.
>
>
> #### Table 1. Results on CLEVR dataset
>
> | Model                 | 1-shot              | 2-shot              | 4-shot              | 8-shot              |
> |-----------------------|---------------------|---------------------|---------------------|---------------------|
> | LLaVA-OneVision -7B   | 48.0 / 48.0 / 48.0 / 38.2 | 42.5 / 37.0 / 39.7 / 33.8 | 33.5 / 34.5 / 35.0 / 31.5 | 33.0 / 33.5 / 34.0 / 28.3 |
> | Phi3-Vision           | 42.0 / 42.0 / 42.0 / 34.5 | 31.5 / 29.5 / 27.5 / 25.5 | 29.5 / 28.5 / 32.5 / 17.7 | 36.5 / 36.5 / 33.7 / 18.7 |
> | Mantis-Idefics2       | 36.0 / 36.0 / 36.0 / 40.2 | 36.5 / 35.5 / 35.3 / 32.0 | 28.5 / 29.5 / 28.5 / 24.5 | 22.5 / 21.5 / 26.3 / 21.7 |
> | InternLM-X2D5         | 62.0 / 62.0 / 62.0 / 63.2 | 62.0 / 59.5 / 61.3 / 58.5 | 57.5 / 55.5 / 55.8 / 56.0 | 53.0 / 51.5 / 51.7 / 53.2 |
>
> #### Table 2. Results on TextOCR dataset
>
> | Model                 | 1-shot              | 2-shot              | 4-shot              | 8-shot              |
> |-----------------------|---------------------|---------------------|---------------------|---------------------|
> | LLaVA-OneVision -7B   | 40.0 / 40.0 / 40.0 / 35.7 | 42.0 / 42.0 / 42.2 / 42.2 | 43.0 / 44.0 / 43.5 / 42.2 | 45.0 / 46.0 / 44.8 / 44.7 |
> | Phi3-Vision           | 32.5 / 32.5 / 32.5 / 32.2 | 40.0 / 39.5 / 39.0 / 38.2 | 42.5 / 42.0 / 42.7 / 39.7 | 45.0 / 43.0 / 44.7 / 41.5 |
> | Mantis-Idefics2       | 12.5 / 12.5 / 12.5 / 13.8 | 24.5 / 23.5 / 24.7 / 25.2 | 27.5 / 28.0 / 29.7 / 27.8 | 29.0 / 28.0 / 27.8 / 27.7 |
> | InternLM-X2D5         | 38.0 / 38.0 / 38.0 / 36.5 | 38.0 / 42.0 / 39.8 / 37.0 | 42.0 / 42.5 / 42.5 / 42.5 | 27.5 / 26.5 / 27.5 / 27.0 |
>
> #### Table 3. Results on CobSAT dataset
>
> | Model                 | 1-shot              | 2-shot              | 4-shot              | 8-shot              |
> |-----------------------|---------------------|---------------------|---------------------|---------------------|
> | Seed-LLama-8B         | 24.0 / 24.0 / 23.7 / 15.8 | 35.0 / 34.0 / 33.5 / 21.8 | 36.0 / 35.5 / 37.2 / 27.8 | 34.5 / 34.0 / 33.7 / 33.7 |
> | Seed-LLama-14B        | 27.0 / 27.0 / 27.0 / 23.0 | 37.0 / 34.0 / 36.5 / 33.3 | 43.5 / 41.5 / 42.2 / 40.8 | 42.0 / 43.0 / 47.8 / 43.8 |
> | Emu2-Gen              | 27.5 / 27.5 / 27.5 / 23.0 | 39.0 / 34.0 / 36.0 / 28.7 | 28.5 / 38.5 / 29.5 / 27.3 | 23.0 / 38.5 / 21.5 / 20.8 |
>
>
> We have included the new experiments and relevant references in the revised paper.
>
> Re: **format learning in VQA and captioning**, to our knowledge, it has not been widely discussed, besides brief mention in the concurrent work [1]. And as far as we know, we are the first to quantitatively evaluate and demonstrate this effect.  If you know more references related to this issue, we are happy to include them.
>
> [1] Towards Multimodal In-Context Learning for Vision & Language Models. arXiv 2024.

---

> > ### Author Response · Authors · 2024-11-21
> > **Responses to Reviewer EuE8 (continued)**
> >
> > ### Q1. Clarification on the number of data samples
> > We evaluate each task separately and when computing the overall ratings we give each task the same weight, so we believe there are no issues with imbalance class sample distribution. Generally we aimed to keep the benchmark compact to make it easily accessible.
> >
> > ### Q2. Text version of images
> > We generated captions corresponding to the images and then used caption-question-answer triple as input for in-context demonstrations. We were able to generate these algorithmically as e.g. in operator induction we control what digits we generate and in CLEVR we have access to the full scene description.
> >
> > ### Q3. Impact of different model architectures
> > Thanks for the suggestion, we have now added a new section to the appendix B.11 where we explore the impact of different model architectures. In our analysis, we split the models into three categories: cross-attention based, LLaVA-like and GPT4V for which the architecture is unknown. To give more context, we also include time in our analysis. **Figure 17** in the appendix indicates that the best accuracies and ICL efficiencies are obtained by LLaVA-like models (LLaVA-OneVision-72B, InternLM-X2d5, Phi3-Vision). However, more broadly there is no clear trend between cross-attention and LLaVA-like models, and both categories can obtain comparable performance. The first models were typically cross-attention based, but researchers have continued developing them also more recently.

---

> > > ### Comment · Reviewer_EuE8 · 2024-11-24
> > >
> > > Thank you for conducting the additional experiments!
> > > However, I still have some issues of textual descriptions in place of images. Can reliance on image captions serve as an accurate substitute for the original images? If alternative models used, such as BLIP or GPT-4V, are employed to generate these captions, there is a risk of hallucination. In such cases, can these captions still be considered reliable replacements? Additionally, if the images are removed from the ICD and only the query and answer are retained, what would be the result of these queries?

---

> > > > ### Author Response · Authors · 2024-11-25
> > > >
> > > > Many thanks for getting back to us and for the additional questions! We provide a further response to each question:
> > > >
> > > > ### Q1. Textual descriptions in place of images
> > > > For the results in Fig 6 there is no risk of hallucination when generating the captions because we do not use a model to generate them. For our experiments with textual descriptions we use CLEVR Count Induction, Operator Induction and Interleaved Operator Induction datasets, and in all of them we have access to the details that we need to generate fully accurate captions using a simple script. The CLEVR dataset includes a detailed description of objects present in each scene (metadata) so we simply use this to create a sentence that enumerates the objects present in the scene. The operator induction datasets are generated by us so we can create the captions in the same way as we generate the images.
> > > >
> > > > An example question (with caption/description) for each:
> > > >
> > > > > CLEVR:\
> > > > > The scene contains the following objects: a large gray metal sphere, a small gray rubber cube, a large red rubber sphere, a large cyan metal cylinder, a large yellow metal sphere. Question: \"material: rubber\""
> > > >
> > > > > Operator Induction:\
> > > > > What is the result of the following mathematical expression? 0 ? 1
> > > >
> > > > > Interleaved Operator Induction:\
> > > > > What is the result of the following mathematical expression? First number: 0, second number: 1
> > > >
> > > > Thus our conclusions with regard to this issue are not affected by hallucinations.
> > > >
> > > > ### Q2. Results when images are removed from ICD
> > > >
> > > > Thank you for your question—it’s an interesting experiment to explore. To address this, we conducted experiments on Fast Open-ended MiniImageNet and Operator Induction, after removing images from ICD. The results, shown below, include two values per cell: the first represents performance without images, and the second is the original performance for comparison. For both datasets, the results after removing images from ICD are below the random chance level, showing the challenges in learning without images in ICD. It is because e.g., the models have to see both image and text from Fast Open-ended MiniImageNet to link the concept (pseudo names) to the image category. This further shows the multimodal nature of the proposed benchmark.
> > > >
> > > > #### Table 4. Fast Open-Ended MiniImageNet
> > > > | Model                  | 1-shot       | 2-shot       | 4-shot       | 5-shot       |
> > > > |------------------------|--------------|--------------|--------------|--------------|
> > > > | LLaVA-OneVision-7B     | 7.00 / 13.50   | 20.17 / 48.17| 24.50 / 49.50  | 23.50 / 49.50 |
> > > > | Phi3-Vision            | 19.17 / 12.67| 31.50 / 50.00 | 32.5 / 39.00 | 30.67 / 40.17|
> > > > | Mantis-Idefics2        | 3.00 / 25.00    | 8.50 / 72.83  | 8.00 / 84.33  | 5.50 / 76.00  |
> > > > | InternLM-X2d5          | 6.83 / 12.00  | 13.67 / 82.50| 13.00 / 88.17 | 8.50 / 91.00  |
> > > >
> > > >
> > > > #### Table 5. Operator Induction
> > > > | Model                  | 1-shot         | 2-shot         | 4-shot         | 8-shot         |
> > > > |------------------------|----------------|----------------|----------------|----------------|
> > > > | LLaVA-OneVision-7B     | 29.44 / 45.0  | 21.67 / 41.11  | 31.67 / 45.56  | 27.22 / 28.33  |
> > > > | Phi3-Vision            | 31.11 / 42.22 | 23.33 / 45.56  | 19.45 / 51.67  | 15.0 / 54.44   |
> > > > | Mantis-Idefics2        | 18.33 / 12.78 | 12.78 / 8.89   | 12.78 / 14.44  | 13.33 / 16.11  |
> > > > | InternLM-X2d5          | 21.11 / 40.00 | 21.67 / 41.67  | 31.67 / 35.00     | 30.89 / 35.00     |

---

### Official Review · Reviewer_vjqe · 2024-11-02

**Soundness:** 2
**Presentation:** 2
**Contribution:** 2
**Rating:** 6
**Confidence:** 3

**Summary:**

This article introduces a comprehensive benchmark named VL-ICL Bench for assessing the capabilities of multimodal in-context learning (ICL). The suite includes a range of tasks involving both image and text inputs and outputs. The authors evaluate the performance of state-of-the-art vision large language models (VLLMs) on this benchmark, revealing their strengths and weaknesses across various tasks, and note that even the most advanced models, such as GPT-4, find these tasks challenging. The article hopes that the dataset will inspire future work on enhancing the in-context learning capabilities of VLLMs and inspire new applications that leverage VLLM ICL.

**Strengths:**

1. Point out the limitations of the common practice of quantitatively evaluating VLLM ICL through VQA and image captioning.
2. Propose a comprehensive benchmark suite of ICL tasks covering diverse challenges, including perception, reasoning, and so on.
3. It rigorously evaluates a range of state-of-the-art VLLMs on the benchmark suite and highlights their diverse strengths and weaknesses.

**Weaknesses:**

1. The evaluation seems too weak. For example, for the ICL tasks of image generation, the community might focus more on generating images based on complex instructions. For example, researchers in such fields prefer using VLMs to evaluate the generated images given complex instructions as described in [1].
2. The possible usage of this model is still unclear. There are two situations when we want to evaluate multi-modal tasks, but I think the proposed benchmark is not suitable for any situation.

In the first situation, we evaluate the generation and understanding abilities separately, which is more reasonable nowadays because existing VLMs are usually good at one type of task even for the recent ones like [2], thus they prefer to be evaluated on benchmarks for specific targets.

In the second situation, for the models that could generate and understand in the same framework, it is better to evaluate them in more complex in-context-learning settings. For example, the input and output both contain images and texts. Only in the way the upper bound of the abilities of such models could be measured. However, this setting is not contained in this paper.

[1] Lin, Zhiqiu, et al. "Evaluating text-to-visual generation with image-to-text generation." European Conference on Computer Vision. Springer, Cham, 2025.
[2] Xiao, Shitao, et al. "Omnigen: Unified image generation." arXiv preprint arXiv:2409.11340 (2024).

**Questions:**

see weakness

---

> ### Author Response · Authors · 2024-11-21
> **Responses to Reviewer vjqe**
>
> Thank you very much for your review and questions. Here are our responses to your comments.
>
> ### W1. Clarification on image generation evaluation
> Thank you for your comments. We would like to argue that while generating images under complex instructions is an important task, it is orthogonal to our learning of image generation with ICL. The goal of our task is to assess whether VLMs have the capabilities to learn the concepts, attributes, rules etc., from the in-context demonstrations and then generate the desired image. To our knowledge, complex instruction based generation such as [1] are zero-shot rather than ICL based. Therefore we have a different problem setting and use-case.
>
> ### W2. Clarification on possible usage of the benchmark
> Thank you for your comments. We do evaluate the generation and understanding capabilities separately for different models as you mentioned in the first situation. For example, as shown in Table 2 and 3, we evaluate different sets of models for image-to-text tasks and text-to-image tasks. With our evaluation, we find that vision-language ICL is challenging in both types of tasks and thus require a comprehensive benchmark to rigorously measure the performance and encourage future development. Therefore, we are providing such a comprehensive ICL suite for both types of models and we believe it will contribute to the research community.

---

> > ### Author Response · Authors · 2024-11-28
> >
> > Dear Reviewer vjqe,
> >
> > We want to thank you here, again, for the constructive comments. Could you please kindly check our responses to see if your concerns are solved? We would like to hear if you have any further questions before the discussion window is over. And if no more questions, please could you consider updating the score?
> >
> > Sincerely, \
> > Authors

---

> > ### Comment · Reviewer_vjqe · 2024-12-02
> > **Official Comments by Reviewer vjqe**
> >
> > Thanks for your clarification. I will raise my score.

---

### Official Review · Reviewer_vd4e · 2024-11-03

**Soundness:** 3
**Presentation:** 3
**Contribution:** 2
**Rating:** 6
**Confidence:** 3

**Summary:**

This paper reveals the limitations inherent in the common practice of quantitatively evaluating VLLM in-context learning (ICL) via VQA and captioning, and then introduces a comprehensive benchmark (i.e., VL-ICL Bench) for multimodal in-context learning. The introduced VL-ICL Bench incorporates both image-to-text (captioning and VQA) and text-to-image tasks, and evaluated various facets of VLLMs including fine-grained perception, rule-induction, reasoning, image interleaving, fast concept binding, long context, and shot scaling. The authors benchmarks over 20 VLLMs and highlight their strengths and limitations.

**Strengths:**

+ The overall paper is well organized and easy to follow.
+ The research problem (i.e., benchmarking the multimodal ICL capabilities of VLLMs) is quite valuable in VLLM communities. Comprehensive analysis experiments are provided to show the limitations of existing benchmarks.
+ The introduced benchmark covers multiple practical ICL tasks and assesses numerous VLLMs. Multiple discussions are provided to show the promising direction for future research.

**Weaknesses:**

- Some texts are not consistent with the figures. For instance, in Line 159-160 and 224-225, the authors claim that the ICL exhibit more significant improvement on text-to-text benchmarks compared to image-to-text benchmarks. However, the differences between Figures 3a and 4 are marginal. These line charts show similar trends.
- Some design choices are not clear. The authors want to show different trends of VLLMs on multimodal and LLM benchmarks in Figure 3 and 4. I am wondering why different models are evaluated?
- The data contribution of the proposed benchmark is somewhat weak, as all the data and annotations are collected from existing datasets. In addition, there is no dataset statics of the introduced VL-ICL Bench. It would be better to show the distribution of the data as well as the tested capabilities.

**Questions:**

- The few-shot multimodal ICL typically helps to improve the performance out-domain or out-of-distribution tasks. I am wondering why this paper only focuses on VQA and captioning?
- Why the peak accuracy is used rather than average accuracy over all shots？
- In Lines 361-363, the authors claim that ‘we attribute this to difficulty of dealing with the larger number of images and tokens confusing the model and overwhelming the value of additional training data’. It there any evidence to support such statement?

---

> ### Author Response · Authors · 2024-11-21
> **Responses to Reviewer vd4e**
>
> Thank you very much for your review and questions. Here are our responses to your comments.
>
> ### W1. Different trends between Figure 3(a) and 4
> Thanks for asking about this part. We agree Figure 3a shows a certain amount of learning, and we have revised the associated text accordingly, in particular the claims in the caption that may have been previously too strong.
> Note that as we explain further in Figure 3, the learning is primarily about how to format the answer, as shown by our analysis there. When we compare Figure 4 with Figure 3b that aims to remove the impact of formatting or style, there is a very clear difference in behavior. Moreover, while both Figures 3a and 4 show curves trending upwards, the trend is stronger in Figure 4, highlighting the scale of ICL present in text-only settings. To further support this, we have computed the ICL efficiency for the different cases and these show the following:
> - Average ICL efficiency across the datasets and models tested in Figure 3a (VLLMs): 7.48
> - Average ICL efficiency across the datasets and models tested in Figure 4 (LLMs): 14.75
> As a result the benefit of ICL is indeed significantly stronger in text-only settings than in VQA/captioning.
>
> ### W2. Design choices of using different models in Figure 3 and 4
> Thank you for your question. Figure 3 highlights a general issue: popular VLLMs do not benefit significantly from in-context learning (ICL) on tasks like image captioning and VQA. In contrast, we select a subset of models in Figure 4 to demonstrate that VLLMs do inherit the ICL capabilities of their base LLMs. The selected models are stronger and more representative, excluding relatively weaker models like OpenFlamingo and Otter. (OpenFlamingo and Otter VLLMs also do not accept text-only input).
>
> ### W3. Clarification on data contribution
> Thank you for your comments. The primary contributions of our benchmark are to construct a comprehensive evaluation suite that allows researchers to assess their models’ multimodal ICL capabilities, and a series of insights from evaluating existing models on that suite.
>
> In constructing the evaluation suite, we collated 10 diverse tasks in one place, which required re-designing the structure and annotations of source datasets in common format for ICL. There is plenty of precedent for this kind of contribution (eg: Triantafillou’s highly cited meta-dataset in ICLR20). Creating the task suite was itself non-trivial. The operator induction datasets and tasks are fully created by us. While the other tasks use existing images, we had to repurpose them in novel ways, e.g. by stacking the images on top of each other to create composed images. While the matching tasks use existing images, the learning task is newly created by us and required deriving new annotation, etc. In summary, we do not simply re-use existing images and annotations directly - they all had to be repurposed in novel ways to design tasks that facilitate thorough testing of ICL.
>
> Re dataset statistics: we do have the data statistics and tested capabilities of each subset in Table 1.
>
> ### Q1. Clarification on VQA and captioning
> We discuss ICL in VQA and captioning in the initial part of our paper as these are the main applications where multimodal ICL is commonly assessed (Bai et al., 2023; Awadalla et al., 2023; Sun et al., 2023a; Laurencon et al., 2023). However, as we discuss in the paper, VQA and captioning see limited benefits from ICL and so we design a benchmark that studies various different tasks, e.g. concept binding, induction or reasoning, to assess multimodal ICL more rigorously.
>
> ### Q2. Why peak accuracy is used instead of average accuracy
>
> We use peak accuracy as one of the metrics in addition to zero-shot accuracy and ICL efficiency in order to study the models from diverse perspectives. ICL efficiency already captures behavior across various numbers of shots and so is very similar to average accuracy over all shots, except that it discounts the zero-shot baseline. Peak accuracy is an interesting metric to report as it tells us about how well a model can perform when given an “ideal” number of shots. It has also been used in literature, in particular (Jiang et al., 2024b).
>
> ### Q3. Evidence on ICL difficulty of dealing with larger number of shots
> Thank you for your question. Some models obtain negative impact from more shots because the total input sequence length exceeds the models’ context length. E.g., in LLaVA, one image translates to 576 tokens, causing an 8-shot setting to exceed its 4k context window. We provide a summary of the context window of the evaluated VLLMs in Table 6.

---

> > ### Author Response · Authors · 2024-11-28
> >
> > Dear Reviewer vd4e,
> >
> > We want to thank you here, again, for the constructive comments. Could you please kindly check our responses to see if your concerns are solved? We would like to hear if you have any further questions before the discussion window is over. And if no more questions, please could you consider updating the score?
> >
> > Sincerely, \
> > Authors

---

> > > ### Comment · Reviewer_vd4e · 2024-11-29
> > > **Response to The authors rebuttal**
> > >
> > > Dear Authors,
> > >
> > > Thank you for your reply. Your response well addressed my concerns. I would increase my rating towards acceptance.

---

### Official Review · Reviewer_vZo4 · 2024-11-05

**Soundness:** 3
**Presentation:** 3
**Contribution:** 3
**Rating:** 6
**Confidence:** 4

**Summary:**

This paper investigates an important property for vision large language models (VLLMs): evaluting the in-context (ICL) ability in multi-modal scenes. The authors first point out current VQA and captioning benchmarks are not ideal for evaluting multi-modal ICL via quantitative results. To bridge the research gap, the authors proposes the first ICL benchmark, VL-ICL bench, which encompasses 10 tasks and is used evaluate both the image-to-text and text-to-image models. The authors provide the comprehensive evaluation for a range of VLLMs on the proposed VL-ICL bench.

Overall, I appreciate the motivation to evaluate the ICL capabilities of multi-modal models, as this is a significant yet under-explored area. The experiments are extensive and yield useful conclusions.

**Strengths:**

1. The paper starts from a good motivation for evaluating the ICL ability of current multimodal models.
2. This paper proposes the VL-ICL bench convering 10 tasks to evaluate the diverse capacibilities such as perception, reasoning, rule-induction.
3. The authors conduct extensive and thorough experiments with the current multimodal models on the proposed benchmark

**Weaknesses:**

1. Some details about the construction of VL-ICL should be clarified. For the datasets used in Table 1, do you use all the samples from the original sources? Do you perform some filtering strategies?
2. Could the authors give more explanations about the metric ICL efficiency? Why the ICL efficiency has negative numbers in Table 2?
3. Based on the curve figures (e.g., Figure 5 and Figure 6), it appears that the multimodal models do not significantly benefit from additional ICL examples, with performance gains primarily saturating at the 1-shot level. What do you think might be the reason for this? Additionally, why do you believe that the proposed VL-ICL offers a better evaluation of ICL capability compared to traditional VQA benchmarks, as the Figure 3(a) also reveals the same trend?

I would like to hear the response from authors, and then make my final score.

**Questions:**

1. Some tables do not bolden the best performance number, such as Table 5.

---

> ### Author Response · Authors · 2024-11-21
>
> Thank you very much for your review and questions. Here are our responses to your comments.
>
> ### W1. Dataset sampling and filtering strategy
> How many samples from the original sources we use is dataset-dependent. In cases such as operator induction we use all possible options. In others such as CLEVR or TextOCR we use a subset as using all samples from the set would make our benchmark unnecessarily large. When we use only a subset, we do filtering in some cases depending on if there are data-quality issues. In particular in TextOCR we did filtering to ensure we use only valid texts that are not marked as rotated. After the filtering we use random subsampling to ensure compactness of our benchmark. We now provide the details of sampling and filtering (if applicable) for each dataset in Appendix A.
>
> ### W2. Explanation on ICL efficiency metric
> We define ICL efficiency as the performance improvement as more examples are seen. Specifically, given an experimental result of accuracy vs number of shots (Fig 3, 4, 5, etc), we calculate a scalar efficiency value as the proportion of the area between the few-shot curve and the zero-shot baseline. Thus the maximum efficiency of 100% area between curves occurs if zero-shot baseline achieves 0% accuracy, and 1-shot and higher performance performance is 100% accuracy. Efficiency is 0% if performance doesn’t improve with more shots and for any number of shots is always equal to the zero-shot baseline (ie: a model that completely ignores its context set). Efficiency is negative if the few-shot performance becomes worse than the zero-shot performance. This can happen if a model fails to properly exploit the context set  and simply becomes confused by the larger number of tokens – for example VILA-7B suffers from this. The goal of ICL is to exploit the context set to improve performance, and this efficiency metric quantifies exactly this effect.
>
> ### W3-1. Reason of saturated ICL performance
> Several multi-models are restricted by context length or perception capabilities:
>
> Context-length limits: the models fail to benefit from more tokens after a certain point (typically related to the length of contexts models have been trained with). This point can be reached in fewer shots for a multi-modal than text model since the images provide more tokens.
>
> Perception limits: the model may be able to induce the rule from in-context examples, but imperfect perception still limits the ability to solve the test examples if the digits/symbols are not always recognised correctly. Perception bottlenecks impact multi-modal models more than text models.
>
> Nevertheless, please note that we do have tasks where there are some models that do not saturate in 1-shot. Figure 8 in the appendix shows that some strong models, e.g. GPT4V and LLaVA-OneVision-72B, can really learn from the in-context demonstrations and show a sharply growing trend w.r.t. shots in some tasks such as Fast Open-Ended MiniImageNet and operator induction.
>
> ### W3-2 Why our benchmark offers a better evaluation of ICL capability
> Some of them indeed have visually similar trends, however we argue that the main trends are different. In Fig 3b, we show that captioning and VQA **start with high zero-shot scores**, indicating that VLLMs can do these tasks well without using the context set, and performance **does not increase with more shots**. Compare this to Fig 6 (and Fig 8 in Appendix for an even stronger contrast) where **performance at zero-shot is low**, indicating a context set is necessary for solving these tasks, and for at least some models **performance increases with shots**.  This difference between zero-shot and ICL performance is summarized by our ICL efficiency metric discussed above, and reported in Tab 2. Several models and tasks show a good ICL efficiency in Tab 2, but ICL efficiency for Fig 3b is approximately zero (more specifically 0.65% on average, compared to average of 14.53% across all models and datasets in our benchmark).
>
> In summary, with captioning and VQA, performance generally does not improve above the zero-shot level, thus not demonstrating any learning from in-context examples; while in our VL-ICL performance starts at chance level and only increases above chance-level after seeing the support set. This makes it a better benchmark to assess ICL capabilities.
>
> ### Q1. Bolden best-performing numbers in Tables
> Thank you for your suggestion. We have bolden the best performance numbers in Table 5 in the revised paper.

---

> > ### Comment · Reviewer_vZo4 · 2024-11-26
> > **Thanks for the authors' response**
> >
> > Thanks the authors' for the detailed response. My concerns are all addressed regarding the dataset construction, the evaluation metric (ICL efficiency) and the superior of VL-ICL over the traditional VQA benchmarks. The authors have added the corresponding revision in the paper. Therefore, I hold the positive attitude towards the submission and keep my rating as 6.

---

### Author Response · Authors · 2024-11-21
**General Response**

Dear AC and reviewers,

We sincerely thank AC and all reviewers for their time and efforts in reviewing our paper. The constructive suggestions have helped us to improve our paper further. We appreciate that reviewers find our paper introducing the VL-ICL Bench well-motivated, studying an important topic, useful for evaluating future models and with valuable insights and extensive experiments.

We have conducted additional experiments and updated the manuscript according to the reviewers' comments and suggestions. Here is a summary of our updates:
- **[Impact of in-context demonstration selection]** We employ similarity-based retrieval for  in-context demonstrations (ICD) selection and show its effectiveness on final performance (updated in appendix B.10)
- **[Analysis of the impact of different model architectures]** We compare the accuracy and ICL efficiency of cross-attention based models and LLaVA-like models across released dates. (updated in appendix B.11)
- **[Clarification of datasets details and writing]** We have further explained the details that reviewers are interested in or feel unclear about. The corresponding contents have been added or modified in our revised version.

---

### Meta-Review · Area_Chair_FBgF · 2024-12-18

**Metareview:**

Summary:
The paper introduces VL-ICL Bench, a comprehensive multimodal benchmark designed to evaluate in-context learning (ICL) abilities in Vision-Language Models (VLLMs). It addresses the limitations of existing VQA and captioning benchmarks by covering 10 diverse tasks (perception, reasoning, rule-induction, etc.), testing both image-to-text and text-to-image settings. The authors rigorously benchmark 20+ VLLMs, highlighting their strengths, weaknesses, and the challenges of scaling ICL.

Strengths:
1) Novel and well-motivated benchmark that fills a significant gap in evaluating multimodal ICL.
2) Thorough experiments with state-of-the-art VLLMs, revealing new insights and limitations.
3) Analysis of task diversity, model architectures, and demonstration selection strategies.

Weaknesses:
1) Some unclear descriptions about dataset construction and demonstration selection initially.
2) Limited focus on complex in-context tasks involving mixed image-text outputs.
3) Evaluation of ICL performance sometimes lacks deeper theoretical analysis.

Overall, the paper proposes a well-constructed and novel benchmark with significant practical utility. The extensive experiments and thorough rebuttal addressing concerns demonstrate strong contributions, advancing multimodal ICL evaluation.

**Additional Comments On Reviewer Discussion:**

The reviewers highlighted concerns regarding dataset construction, metric clarity, model architecture comparisons, and the effectiveness of in-context learning (ICL). The authors addressed these by adding detailed clarifications, new experiments on ICD selection, and updates on dataset statistics. These improvements resolved major concerns, leading to the decision to recommend acceptance.

---

### Decision · Program_Chairs · 2025-01-22

Accept (Poster)